

# Aircraft-based observation of meteoric material in lower stratospheric aerosol particles between 15 and 68°N

Johannes Schneider[1], Ralf Weigel[2], Thomas Klimach[1], Antonis Dragoneas[1,2], Oliver Appel[1,2], Andreas Hünig[1,2], Sergej Molleker[1,2], Franziska Köllner[1,2], Hans-Christian Clemen[1], Oliver Eppers[1,2], Peter Hoppe[1], Peter Hoor[2], Christoph Mahnke[2,1,3], Martina Krämer[3,2], Christian Rolf[3], Jens-Uwe Grooß[3], Andreas Zahn[4], Florian Obersteiner[4], Fabrizio Ravegnani[5], Alexey Ulanovsky[6], Hans Schlager[7], Monika Scheibe[7], Glenn S. Diskin[8], Joshua P. DiGangi[8], John B. Nowak[8], Martin Zöger[9], Stephan Borrmann[2,1]

[1]Particle Chemistry Department, Max Planck Institute for Chemistry, Mainz, 55128, Germany
[2]Institute for Physics of the Atmosphere, Johannes Gutenberg University, Mainz, 55128, Germany
[3]Forschungszentrum Jülich, Institute of Energy and Climate Research, Jülich, 52425, Germany
[4]Institute for Meteorology and Climate Research, Karlsruhe Institute of Technology (KIT), Eggenstein-Leopoldshafen, 76344, Germany
[5]Institute of Atmospheric Sciences and Climate, ISAC-CNR, Bologna, 40129, Italy
[6]Central Aerological Observatory, Dolgoprudny, Moscow Region, 141700, Russia
[7]Institute of Atmospheric Physics, German Aerospace Center (DLR) Oberpfaffenhofen, Wessling, 82234, Germany
[8]NASA Langley Research Center, MS 483, Hampton, VA, 23681, USA
[9]Flight Experiments Department, German Aerospace Center (DLR) Oberpfaffenhofen, Wessling, 82234, Germany

*Correspondence to*: Johannes Schneider (johannes.schneider@mpic.de)

**Abstract.** In this paper we analyze aerosol particle composition measurements from five research missions conducted between 2014 and 2018 sampling the upper troposphere and lower stratosphere (UTLS), to assess the meridional extent of particles containing meteoric material. Additional data sets from a ground based study and from a low altitude aircraft mission are used to confirm the existence of meteoric material in lower tropospheric particles. Single particle laser ablation techniques with bipolar ion detection were used to measure the chemical composition of particles in a size range of approximately 150 nm to 3 μm. The five UTLS aircraft missions cover a latitude range from 15 to 68°N, altitudes up to 21 km, and a potential temperature range from 280 to 480 K. In total, 338 363 single particles were analyzed, of which 147 338 particles were measured in the stratosphere. Of these particles, 50 688 were characterized by high abundances of magnesium, iron, and rare iron oxide compounds, together with sulfuric acid. This particle type was found almost exclusively in the stratosphere (48 610 particles) and is interpreted as meteoric material immersed or dissolved within stratospheric sulfuric acid particles. Below the tropopause, the fraction of this particle type decreases sharply. However, small fractional abundances were observed below 3000 m a.s.l. in the Canadian Arctic and also at the Jungfraujoch high altitude station (3600 m a.s.l.). Thus, the removal pathway by sedimentation and/or mixing into the troposphere is confirmed. Our data show that in the tropics, the lower stratosphere contains only a small fraction (< 10%) of particles containing meteoric material. In contrast, in the extratropics the fraction of meteoric particles reaches 20-40% directly above the tropopause. At potential temperature levels of more than 40 K above the thermal tropopause, particles containing meteoric





material were present in higher relative abundances than near the tropopause and, at these altitudes, occurring at similar abundance-fraction across all latitudes and seasons measured. Above 440 K, the fraction of meteoric particle ranges between 60 and 80% at latitudes between 20 and 42°N. This finding suggests that the meteoric material is transported from the mesosphere into the stratosphere within the winter polar vortex, and is efficiently distributed towards low latitudes by

isentropic mixing above 440 K potential temperature. This process can explain that meteoric material is found in particles of the stratospheric aerosol layer at all latitudes.

## 1 Introduction

Aerosol particles in the upper troposphere/lower stratosphere (UTLS) play an important role in the Earth's radiative budget: Firstly, by direct scattering of sunlight back to space, secondly, by influencing homogeneous and heterogeneous cirrus cloud

formation in the upper troposphere (UT). The radiative forcing via aerosol-cloud interaction of ice clouds in the atmosphere was estimated in a model study to be around + 0.27 W m$^{-2}$ (Gettelman et al., 2012). Furthermore, especially under volcanically influenced conditions, aerosol particles in the UTLS provide surfaces for heterogeneous chemical reactions to occur, thereby influencing ozone chemistry (Pitari et al., 2002; von Hobe et al., 2011), and influence the stratospheric circulation due to heating of the stratosphere (Robock, 2000; Kremser et al., 2016). Generally, the dominating sources of

aerosol particles and their precursor gases are in the troposphere. Primary emitted particles are for example dust, sea spray, black carbon, and biomass burning particles. Secondary aerosol particles are formed from precursor gases such as organic and sulfur-containing compounds. The sources for both primary and secondary particles can be natural or anthropogenic. In contrast, an exclusively natural source of a certain fraction of atmospheric particles is located outside the Earth's atmosphere, causing an ambling but continuous particle import of cosmic origin (Pruppacher and Klett, 1997). The magnitude of cosmic

material entering the Earth's atmosphere is currently estimated to be about 43 ± 14 t d$^{-1}$ (tons per day) (Plane, 2012; Carrillo-Sánchez et al., 2016). Besides oxygen, major elements of meteoric material are iron (Fe), magnesium (Mg), and silicon (Si), which are found with roughly equal proportions in chondritic meteorites; the most abundant minor elements are carbon (C), sulfur (S), aluminum (Al), sodium (Na), calcium (Ca), and nickel (Ni) (Lodders and Fegley Jr., 1998; Hoppe, 2009; Plane et al., 2015). First detection of magnesium emission lines in the night sky spectrum and the conclusion that at least part of

atmospheric magnesium is of meteoric origin were reported by Hicks et al. (1972).

About 8 t d$^{-1}$ of the cosmic dust particles (with diameters between ~ 1 µm and ~300 µm) are completely ablated during entry in the Earth's atmosphere (Plane, 2003; Carrillo-Sánchez et al., 2016). Quenching of evaporated compounds is expected to cause their rapid re-nucleation in the mesosphere to form new particles of the size of a few nanometers, which are commonly referred to as meteoric smoke particles (MSP) (Saunders et al., 2012; Plane et al., 2015; Hervig et al., 2017). Recent remote-

sensing and in-situ measurements in the mesosphere indicated that Fe and Mg are the main constituents of MSP (Hervig et al., 2012; Rapp et al., 2012). MSP have been identified to act as ice nuclei for noctilucent clouds in the mesopause region (e.g., Alpers et al., 2001; Gumbel and Megner, 2009; Megner and Gumbel, 2009; Rapp et al., 2010) and therefore, it is



suggested that MSP have an impact on polar mesospheric summer echoes (Rapp and Lübken, 2004; Megner et al., 2006). As MSP are too small to sediment gravitationally, it is widely assumed that MSP are drained from the mesosphere into the

stratosphere most efficiently due to the air mass subsidence within the polar winter vortex, on a timescale of months (Plumb et al., 2002; Curtius et al., 2005; Megner et al., 2008; Plane, 2012; Saunders et al., 2012; Weigel et al., 2014; Plane et al., 2015; Kremser et al., 2016). The aerosol particles in the stratospheric aerosol layer (Junge et al., 1961; Junge and Manson, 1961; Kremser et al., 2016) consist mainly of sulfuric acid-water ($H_2SO_4$-$H_2O$) droplets (Lazrus et al., 1971; Rosen, 1971; Lazrus and Gandrud, 1974, 1977; Sedlacek et al., 1983; Gandrud et al., 1989; Arnold et al., 1998). The typical size range of

these droplets ranges between 100 and 200 nm (Plane et al., 2015; Kremser et al., 2016). It is thought that MSPs partially dissolve in the droplets (Murphy et al., 1998; Cziczo et al., 2001; Saunders et al., 2012; Murphy et al., 2014), such that a dilute solution of highly soluble ferrous/ferric sulfate and hydrated magnesium sulfate and silicic acid is formed (Saunders et al., 2012). It was further suggested that silicon and aluminum are present as undissolved granular cores within the droplets (Murphy et al., 2014) which would explain the observations of refractory particles in the Arctic lower stratosphere (Curtius

et al., 2005; Weigel et al., 2014).

As has recently been shown by Subasinghe et al. (2016), about 95 % of cosmic bodies of sizes greater than 1 mm in diameter undergo fragmentation upon entering the Earth's atmosphere, thereby forming unablated meteoric fragments (MF) of presumably submicron size. If such fragments were formed, these particles may sediment directly into the lower stratosphere. It has been suggested that MF may play a role in polar stratospheric cloud (PSC) formation, thereby influencing

polar ozone destruction (Voigt et al., 2005; James et al., 2018). Satellite-based observations of PSCs in the Arctic were reproduced by model simulations using CLaMS (Chemical Lagrangian Model of the Stratosphere), but only if heterogeneous nucleation of NAT (nitric acid trihydrate, Grooß et al., 2014) and ice particles (Tritscher et al., 2019) on foreign nuclei were included in the model parameterization. A potential source of the foreign nuclei is meteoric dust (James et al., 2018).

Additionally, certain amounts of cosmic particulate material enter the Earth's atmosphere as Interplanetary Dust Particles

(IDP) which, if smaller than 1 µm in diameter, are too small to experience any ablative altering during atmospheric entry. The origin of IDP is mainly attributed to collisions of asteroids, sublimation of comets and long-decayed cometary trails (Plane, 2003, 2012). In terms of the size-segregated mass influx of cosmic particles (Plane, 2003, 2012), the contribution of submicrometer sized IDP to the total atmospheric aerosol particle load is estimated to be small with estimates of about 150 t per year. However, the import of IDP is likely a continuous process compared to sporadic events of meteoric entries that

produce by far more MSP per event than a single IDP. Therefore, an ambling and persistent import of cosmic aerosol particles (by number) should be considered in relationship to the infrequent but then excessively effective ablation/fragmentation events releasing large amounts of MSP and MF in the atmosphere.

The existence of particles containing meteoric material in the lower stratosphere has been shown by direct in-situ observations. Mossop (1965) reported on insoluble inclusions found in stratospheric particles sampled at 20 km by the U-2

aircraft and suggested a meteoric origin of these particles. Shedlovsky and Paisley (1966) analyzed particles sampled by the U-2 aircraft and detected sulfur, iron, sodium, copper, and chromium. Later, aircraft-based in-situ aerosol mass spectrometry





in the tropical and mid-latitude lower stratosphere at altitudes up to 19 km showed a significant fraction of particles containing meteoric material and sulfuric acid (Murphy et al., 1998; Cziczo et al., 2001; Froyd et al., 2009; Murphy et al., 2014). Indirect evidence for the existence of meteoric aerosol particle material in the Arctic lower stratosphere up to 20 km

altitude was reported by Curtius et al. (2005) and Weigel et al. (2014). They measured non-volatile particles that were thermally stable on exposure to 250°C and had diameters of 10 nm to a few micrometers. The fraction of these non-volatile particles increased with altitude up to 70% at potential temperature levels between 430 and 500 K. Ebert et al. (2016) report on submicrometer particles collected with a cascade impactor in the Arctic stratosphere during the winters 2010 and 2011 that were analyzed for their chemical composition and morphology. They found Fe-rich particles, Ca-rich particles, silicates,

silicate/carbon mixed particles and mixed metal particles from different sources, such as meteoric material, space debris and to lower extent terrestrial sources.

Here we report findings from aircraft measurements of aerosol particle composition in the lower stratosphere at different altitudes, latitudes and seasons:

- Western Europe, spring (March-April 2014) and summer (August-September 2018),

- Mediterranean, summer (August-September 2016),

- Tropics/subtropics, summer (July-August 2017 and August 2018),

- North America/Northern Atlantic, winter (January-February 2018),

In all data sets we observed a distinct particle composition type in the lower stratosphere that can be interpreted as particles containing meteoric material, dissolved in or coated by sulfuric acid. We discuss mass spectral composition, size

distribution, vertical profiles, latitudinal distribution, and cross-tropopause transport of particles containing meteoric material.

## 2 Measurements and Methods

### 2.1 Field measurements

This study includes stratospheric and upper tropospheric data obtained during five aircraft-based research campaigns,

together with two data sets from altitudes below 3600 m a.s.l. The individual campaigns are described briefly in the following. The flight tracks of all UTLS research flights included here as well as the locations of the low altitude measurements are depicted in Fig. 1. Overview about the five UTLS aircraft campaigns is provided in Table 1.

### 2.1.1 ML-CIRRUS

The field campaign ML-CIRRUS (Mid-Latitude Cirrus) was conducted in March and April 2014 out of Oberpfaffenhofen,

Germany, using the research aircraft HALO (High Altitude and Long Range Research Aircraft). The objective of ML-CIRRUS was to study cirrus clouds by in-situ and remote sensing methods. Including test flights, a total of 16 flights were





carried out. Most of the flight time (in total 88 hours) was spent in the upper troposphere and lower stratosphere. Laser ablation aerosol mass spectrometer data were recorded during 15 flights which are included in this study. A detailed overview on the mission is given by Voigt et al. (2017).

**2.1.2 StratoClim**

Two aircraft-based research campaigns were conducted within StratoClim (Stratospheric and upper tropospheric processes for better climate predictions) which is a collaborative research project funded by the European Commission. The first StratoClim campaign took place at Kalamata airport, Greece, in August and September 2016. The aim of the mission was to study atmospheric composition in the Eastern Mediterranean region, including the remote influence of the Asian monsoon

anticyclone (AMA) outflow. Three research flights were conducted. The second StratoClim campaign was a dedicated field activity to investigate the impact of the AMA on the UTLS and took place at the Tribhuvan International Airport of Kathmandu, Nepal, in July and August 2017 (e.g., Höpfner et al., 2019). Eight scientific flights were carried out over Nepal, India and Bangladesh. The flight paths spanned latitudes from 21° N to 27° N and longitudes from 79° E to 90° E (see Fig. 1).

**2.1.3 ND-MAX/ECLIF-2**

The ND-MAX/ECLIF-2 (NASA/DLR-Multidisciplinary Airborne eXperiments/Emission and CLimate Impact of alternative Fuel) mission aimed for the characterization of gaseous and particulate aircraft emissions with a dedicated aircraft chasing field experiment over South-West Germany. For this mission, the installation of instrumentation into the NASA DC-8 aircraft took place at Palmdale, CA, USA. Measurements taken during the ferry flights from Palmdale to Germany on

January 13, 2018 and back on February 3 and 4, 2018, were used in this study. These flights reached latitudes up to 68°N (see Fig. 1), longitudes as far as 120° W, and penetrated deep into the winter stratosphere at around 11 km altitude.

**2.1.4 CAFE-Africa**

CAFE-Africa (Chemistry of the Atmosphere Field Experiment in Africa) was conducted with HALO in August 2018 out of Sal on the Cape Verde Islands. The main objective was to study the African monsoon outflow in the upper troposphere over

the Atlantic Ocean. This study includes only data which were obtained during three research flights reaching the stratosphere. These flights took place on August 15, August 24, and September 07, 2018, the latter being the ferry flight back to Germany. The flight tracks of these three flights are included in Fig. 1.

**2.1.5 Additional low altitude data sets**

To investigate the possible occurrence of meteoric particles in the lower troposphere, we used two data sets from low

altitudes: One data set was obtained during NETCARE (Network on Climate and Aerosols: Addressing Key Uncertainties in Remote Canadian Environments (Abbatt et al., 2019)), conducted in the Arctic out of Resolute Bay (Nunavut, Canada) in

July 2014. A single particle mass spectrometer was operated on board the Polar 6 aircraft (Alfred Wegener Institut –
Helmholtz Zentrum für Polar- und Meeresforschung) and measured at altitudes between 0 and 3 km. Details of the campaign
and the mass spectrometer data are given in Köllner et al. (2017). During the INUIT-JFJ (Ice Nucleation Research Unit
Jungfraujoch) campaign in January and February 2017, a single particle mass spectrometer was operated for ground based
measurements on the High Alpine Research Station Jungfraujoch (3600 m a.s.l.). The mass spectrometer data are still
unpublished, but details on the campaign can be found in Eriksen Hammer et al. (2018) and Gute et al. (2019).

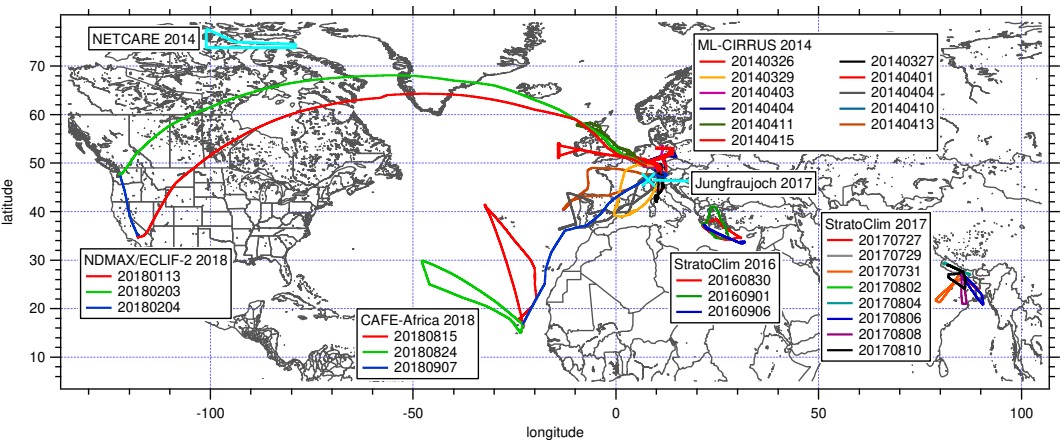

**Figure 1: Map of the flight tracks of all UTLS research flights used in this study. Additionally the locations of the low altitude
measurements are indicated: Jungfraujoch (3600 m a.s.l.) and operation range of the NETCARE flights (0 – 3000 m a.s.l.).**

**2.2 Instrumentation and data analysis**

**2.2.1 Aerosol mass spectrometer operated during ML-CIRRUS and CAFE-Africa**

The aircraft-based laser ablation aerosol mass spectrometer (ALABAMA) has been described in detail in Brands et al.
(2011), Köllner et al. (2017), and Clemen et al. (2020 (in review)). Briefly, the ALABAMA is a bipolar ion single particle
analysis instrument that samples aerosol particles from ambient air through a constant pressure inlet (CPI, Molleker et al.,
2020 (accepted)) and an aerodynamic lens. The sampled particle size range (vacuum aerodynamic diameter $d_{va}$, DeCarlo et
al. (2004)) was between about 200 and 1000 nm during ML-CIRRUS and between 200 nm and 3000 nm during CAFE-
180    Africa. Having passed the aerodynamic lens, the particles are accelerated into the vacuum chamber. The particles are
detected by two 405 nm laser diodes and their velocity information is used to determine their vacuum aerodynamic diameter
and to trigger a laser shot of the ablation laser (quadrupled Nd:YAG, 266 nm) that hits the particles in the ionization region



of the bipolar ion time-of-flight mass spectrometer. During ML-CIRRUS and CAFE-Africa, aerosol particles were sampled through the HALO aerosol submicrometer inlet (HASI, Andreae et al., 2018). The inlet was mounted on the upper side of the fuselage of the aircraft. Inside the aircraft, the sampled aerosol particles were guided through a 2.9 m long stainless steel sampling line with an inner diameter of 5 mm to the ALABAMA. The calculated transmission efficiency of this sampling line is shown in the supplement (Fig. S10). During ML-CIRRUS, the ALABAMA was operative during 15 flights and analyzed more than 24000 ambient aerosol particles (see Table 1). From CAFE-Africa, we include here a subset of three flights where HALO reached the stratosphere. In these three flights the ALABAMA sampled and analyzed more than 65000 particles. The higher efficiency and higher upper size cut-off (see above) of the ALABAMA in CAFE-Africa compared to ML-CIRRUS are due to several instrumental improvements such as a new aerodynamic lens system and delayed ion extraction (Clemen et al., 2020 (in review)). The ALABAMA was also used in the above-mentioned low altitude field campaigns NETCARE and INUIT-JFJ.

### 2.2.2 Aerosol mass spectrometer operated during StratoClim and ND-MAX

The newly developed mass spectrometer ERICA (ERC Instrument for Chemical composition of Aerosols) combines single particle laser ablation and flash vaporization/ionization techniques. It was designed for fully automated operation on the high altitude research aircraft M-55 "Geophysica" during the StratoClim project and was later re-configured to be operated on the NASA DC-8 during the ND-MAX/ECLIF-2 mission. Here we use only data obtained using the laser ablation part of the ERICA (ERICA-LAMS). The basic design is similar to that of the ALABAMA, but since this is a newly developed instrument, it is briefly reviewed here. The aerosol particles are sampled via a CPI (Molleker et al., 2020 (accepted)) and an aerodynamic lens designed for PM2.5 (Peck et al., 2016). In the vacuum chamber, the particles are detected by two laser diodes (405 nm) and ablated by a pulsed quadrupled Nd:YAG laser (Quantel Ultra, LUMIBIRD SA, Roubaix, France) emitting at 266 nm. This Nd:YAG laser is operated without a wavelength separator in the laser head and thus emits also a small fraction of the energy in form of the first and second harmonic (1064 and 532 nm). The generated ions are analyzed in a bipolar ion time-of-flight mass spectrometer (Tofwerk AG, Thun, Switzerland). The size range of the ERICA-LAMS is approximately 100 – 5000 nm ($d_{va}$). Particle size was calibrated using PSL (polystyrene latex) particles with diameters between 80 nm to 5000 nm. The particle detection efficiency at the laser diodes reaches a maximum of about 75% at 400 nm and decreases towards lower and higher diameters. The hit rate, defined as the ratio of recorded particle mass spectra to laser shots on particles was during the StratoClim measurements about 40 % at diameters around 500 nm, between 5 and 10 % below 200 nm, 20 % at 1000 nm, and below 5 % above 2000 nm. During StratoClim, the ERICA was operated on 11 research flights (three in 2016 and eight in 2017), and the ERICA-LAMS analyzed about 150 000 single particles (see Table 1). During the three ferry flights conducted in the ND-MAX/ECLIF-2 project that are used here, the ERICA-LAMS recorded more than 98 000 single particle mass spectra.



215

Table 1. Overview on the UTLS data sets used in this study.

| Project | ML-CIRRUS | StratoClim 2016 | StratoClim 2017 | ND-MAX | CAFE-Africa |
|---|---|---|---|---|---|
| Time | Mar – Apr 2014 | Aug – Sep 2016 | Jul – Aug 2017 | Jan - Feb 2018 | Aug – Sep 2018 |
| Measurement region | Western Europe | Eastern Mediterranean | South Asia | U.S. to Europe | Atlantic Ocean |
| Aircraft | HALO (G550) | M-55 Geophysica | M-55 Geophysica | NASA DC-8 | HALO (G550) |
| Instrument | ALABAMA | ERICA | ERICA | ERICA | ALABAMA |
| No. of flights used in this study | 15 | 3 | 8 | 3 | 3 |
| Altitude range (km) | up to 13.8 km | up to 20.2 km | up to 20.5 km | up to 11 km | up to 14.5 km |
| Theta range (K) | 276 - 387 | 295 - 490 | 310 - 480 | 276 – 340 | 295 - 380 |
| Latitude range (° N) | 36.3 – 57.5 | 33.4 – 41.0 | 20.8 - 29.5 | 34.6 – 68.1 | 15.0 – 48.2 |
| PV range (PVU) | 0 - 10 | 0 – 24 | 0 - 22 | 0 - 8 | 0 - 10 |
| Total number of single particle mass spectra | 24833 | 11709 | 138119 | 98598 | 65104 |
| In stratosphere (PV > 4 PVU) | 6509 | 5092 | 51599 | 73367 | 10771 |
| Number of mass spectra dominated by magnesium and iron | 3140 | 2412 | 18688 | 23138 | 3310 |
| In stratosphere (PV > 4 PVU) | 2986 | 2271 | 18421 | 22050 | 2882 |



### 2.2.3 Single particle mass spectrometer data analysis

The aircraft data sets from all campaigns were analyzed using a consistent procedure to ensure comparability of the results. First, all data measured during one campaign were merged into one data set per campaign. This resulted in data sets containing individual spectra information of 11 709 particles (StratoClim 2016) to up to 138 119 particles (StratoClim 2017) as given in Table 1. These data sets were clustered separately using a fuzzy c-means algorithm (for a general description see Bezdek et al. (1984) and Hinz et al. (1999); for an ALABAMA-specific description see Roth et al. (2016)), with a pre-selected number of 20 clusters per campaign. Only cations were considered for the clustering algorithm for two reasons: First, during ML-CIRRUS many anion mass spectra were too noisy. Second, the particle type of interest was found to be mainly characterized by the cation mass spectrum, containing magnesium and iron, as explained in the next section. Further clustering details are given in the supplement (Section S.1 and Table S1). For quality assurance and uncertainty estimation, the clustering was repeated using different starting conditions and also different algorithms. The results showed only small deviations in the type of clusters and in the numbers of mass spectra attributed to the clusters (supplement, Section S.1 and Table S2). Mean mass spectra (anions and cations) were calculated for each cluster and were used for the interpretation of the particle type associated with this cluster. Histograms of relative particle abundance were calculated for each cluster as function of altitude, potential temperature ($\Theta$), and potential vorticity (PV).

The data sets from low altitudes (NETCARE, Jungfraujoch) were treated differently: Here we searched specifically for mass spectra using selected marker ions that were found in the high altitude data. This is explained later in detail (Section 3.7).

### 2.2.4 Auxiliary data

The relation of water vapor ($H_2O$) and ozone ($O_3$) can be used to investigate the potential tropospheric influence of a stratospheric air mass. To study the transport of the particles across the tropopause, we use $O_3$ as stratospheric tracer and $H_2O$ as a tropospheric tracer. We use independent particle number concentration and particle size measurements to convert the mass spectrometer data to number concentrations. These measurements are briefly explained here.

Water vapor was measured during ML-CIRRUS and StratoClim by the airborne Fast In-situ Stratospheric Hygrometer (FISH). This instrument uses Lyman-alpha photofragment fluorescence and is described in detail by Zöger et al. (1999). The detection limit is reported to be below 0.4 ppmv, the uncertainty was determined to be about 8 – 30% for low $H_2O$ mixing ratios (1 – 4 ppmv) and 6 – 8% between 4 and 1000 ppmv (Meyer et al., 2015). During ML-CIRRUS, FISH sampled the air through a forward facing inlet mounted on the upper fuselage of the HALO aircraft, whereas during StratoClim, the forward facing FISH inlet was mounted on the side of the fuselage of the Geophysica aircraft (Afchine et al., 2018). The forward facing inlet also samples cloud droplets and ice crystals which evaporate in the inlet, such that the FISH measurements refer to total water. We therefore restricted the data set to non-cloud conditions, by removing the data points where the $H_2O$ saturation ratio was greater than 0.8. During ND-MAX/ECLIF-2, water vapor was measured using the Diode Laser Hygrometer (DLH) of NASA/LaRC (Diskin et al., 2002), which has an uncertainty of 5%. During CAFE-Africa, water





vapor was measured by SHARC (Sophisticated Hygrometer for Atmospheric ResearCh) based on direct absorption measurement by a tunable diode laser (TDL) system. The uncertainty of SHARC is 5% or ± 1 ppmv.

Ozone was measured during ML-CIRRUS and CAFE-Africa by the Fast Airborne Ozone Monitor (FAIRO), whereas during
StratoClim, $O_3$ was measured by the Fast Ozone Analyzer (FOZAN-II). FAIRO combines a dry chemiluminescence detector (CI-D) with a 2-channel UV photometer. The total uncertainty is 1.5% at 8 Hz or 1.5 ppb, whatever is higher, see Zahn et al. (2012). FOZAN-II is likewise based on a CI-D and has a uncertainty of less than 10%, see Yushkov et al. (1999) and Ulanovsky et al. (2001). During ND-MAX, $O_3$ was measured by the UV photometric Ozone analyzer TE49 (Thermo Scientific) with an uncertainty of 5%.

Aerosol particle size distributions were measured during both HALO missions using an optical particle spectrometer of type Grimm 1.129 "Sky-OPC" which was installed next to the ALABAMA. The Sky-OPC measured the total particle number concentration and size distribution for particles larger than 250 nm (manufacturer calibration) in diameter with a reproducibility of 3%, given by the manufacturer. During the StratoClim campaigns, we used a modified Ultrahigh Sensitive Aerosol Spectrometer (UHSAS-A), with a particle diameter range from 65 nm to 1000 nm. The modifications allowed for an
airborne application range up to the extreme conditions in the stratosphere at a height of 20 km. The measurement uncertainties of the UHSAS were determined to 10% (Mahnke, 2018).

Basic meteorological parameters such as pressure, temperature, as well as aircraft position and altitude were obtained during ML-CIRRUS and CAFE-Africa from the Basic HALO Measurement and Sensor System (BAHAMAS), during StratoClim from the Unit for Connection with the Scientific Equipment (UCSE), and during ND-MAX/ECLIF-2 from the NASA DC-8
facility instrumentation.

### 2.3 Meteorological reanalysis

Meteorological parameters were derived using ERA-Interim reanalysis (Dee et al., 2011) from the European Centre of Medium Range Weather forecast (ECMWF). For meridional characterization we use equivalent latitude (Lary et al., 1995) from ERA-Interim. For vertical coordinate, we use potential vorticity from ERA-Interim and potential temperature derived
from observed pressure and temperature data. The location of the thermal tropopause (first lapse rate tropopause) in potential temperature coordinates was taken from ERA-Interim.





# 3 Results

## 3.1 Distinct particle type containing magnesium and iron ions

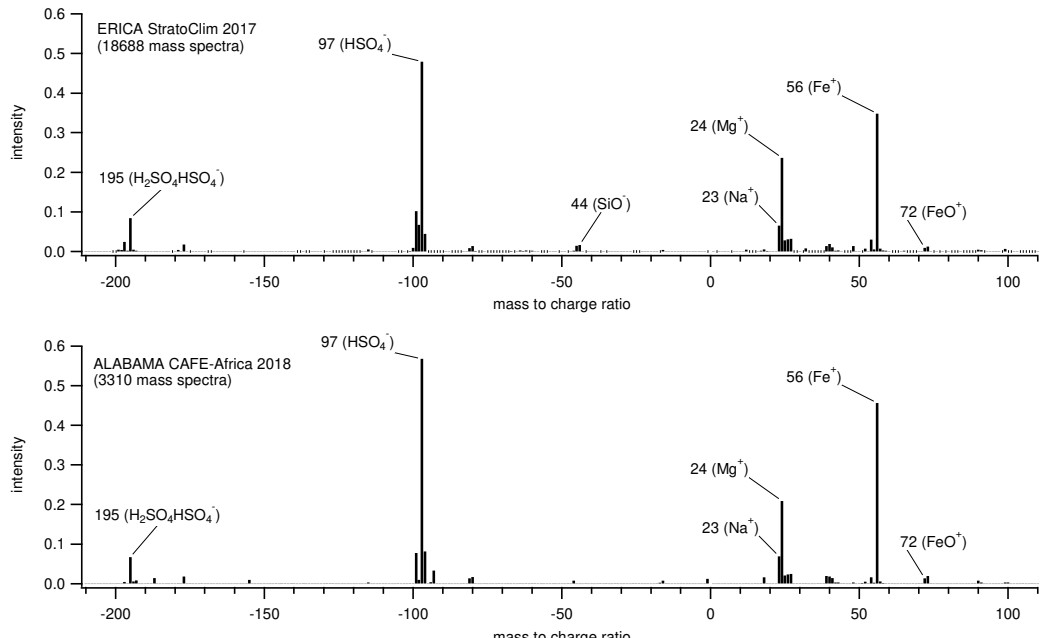

**Figure 2: Mean mass spectra of clusters containing particles of which the positive mass spectra are dominated by iron (Fe⁺, FeO⁺) and magnesium (Mg⁺). The upper panel shows the average over 18688 mass spectra recorded by the ERICA during StratoClim 2017, the lower panel shows an average over 3310 ALABAMA mass spectra recorded during CAFE-Africa 2018. The two anion as well as the two cation mass spectra correlate between the instruments with $r^2 = 0.97$. The only difference is the detection of SiO⁻ (m/z 44) by the ERICA.**

In all five upper tropospheric and lower stratospheric aircraft data sets, the clustering algorithm yielded a type of mass spectra with a mean cation mass spectrum characterized by high abundance of magnesium (Mg⁺, m/z 24 for the major isotope, m/z 25 and 26 for the minor isotopes) and iron (Fe⁺, m/z 56 for the major isotope, m/z 54 for the most abundant minor isotope). Also oxides of Fe (FeO⁺, m/z 72; FeOH⁺, m/z 73) were clearly detected. Further cations include sodium (Na⁺, m/z 23), aluminum (Al⁺, m/z 27), as well as minor signals of potassium (K⁺, m/z 39 and 41) and calcium (Ca⁺, m/z 40). The mean anion mass spectrum contains almost exclusively sulfuric acid ions, as HSO₄⁻ (m/z 97) and H₂SO₄HSO₄⁻ (m/z 195). Figure 2 shows the averaged bipolar ion spectra of this particle type from two aircraft missions, namely



StratoClim 2017 and CAFE-Africa 2018. The mean mass spectra obtained from 18668 measurements made during the StratoClim 2017 campaign look remarkably similar compared to the mean mass spectra obtained from 3310 measurements made during the CAFE-Africa 2018 campaign. A linear correlation between the mean mass spectra measured during CAFE-Africa and StratoClim yielded an $r^2$ of 0.97 for both the anions and the cations. The only difference is the detection of $SiO^-$ (m/z 44) by ERICA during StratoClim 2017 that was missing from the CAFE-Africa observations. This might be due to the additional emission of 1064 and 532 nm light of the ERICA laser in contrast to the ALABAMA laser, such that the ionization probability of Si-containing compounds is higher in the ERICA than in the ALABAMA.

Figure 3 shows the fractional abundance of the iron and magnesium particle type binned by altitude, potential temperature, potential vorticity, and ozone. For each bin (e.g., an altitude interval), the number of iron and magnesium type mass spectra recorded in this bin was divided by the total number of mass spectra recorded in this bin. The bin sizes are 500 m for altitude, 4 K for potential temperature, 0.5 PVU (potential vorticity units, 1 PVU = $10^{-6}$ m$^2$ s$^{-1}$ K kg$^{-1}$) for potential vorticity, and 50 ppbv for ozone. In total, we detected 3140 particles of this type during ML-CIRRUS, 2412 during StratoClim 2016, 18688 during StratoClim 2017, 23138 during ND-MAX 2018, and 3310 during CAFE-Africa 2018 (see also Table 1). It has to be emphasized here that this fractional abundance refers to the total number of analyzed particles by the ERICA and the ALABAMA. Both instruments use a 266 nm laser for ablation and ionization. Pure sulfuric acid particles are not ablated and ionized at this wavelength, as was previously reported (Thomson et al., 1997; Murphy, 2007) and validated by laboratory measurements with the ERICA. Thus the fraction of the iron and magnesium particle type given here represents an upper limit, because pure sulfuric acid particles are not taken into account. This is discussed in more detail in Section 3.5.

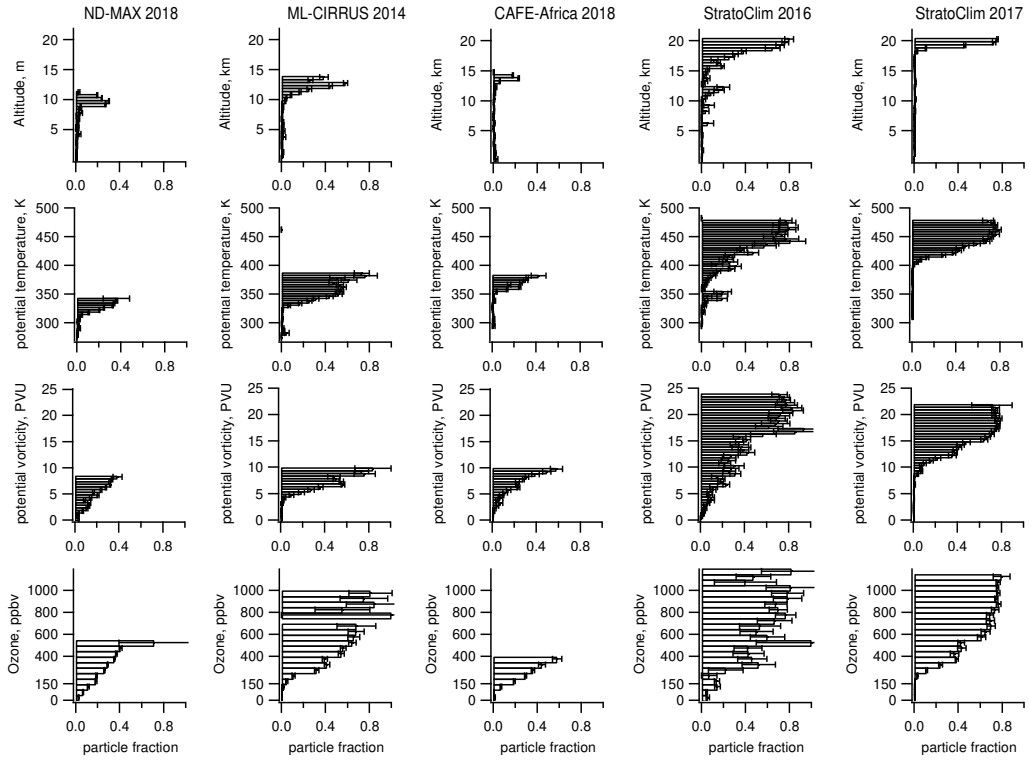


**Figure 3: Fractional abundance of particles with cation spectra dominated by magnesium and iron ions. Upper row: as function of geometric altitude; second row: as function of potential temperature; third row: as function of potential vorticity (PV); forth row: as function of ozone mixing ratio. The missions are not sorted in chronological order but from low potential temperature (leftmost column) to high potential temperature range (rightmost column). Error bars were calculated from Poisson counting statistics and**

**propagation of uncertainty.**

During these five aircraft campaigns, the number of these particles was largest at high altitudes (upper row in Fig. 3), where

a fractional abundance of 0.8 was calculated. Similar values (up to 0.6) were reported by Murphy et al. (2014) for particles with the same ion signals in the mid-latitude stratosphere. The potential temperature and potential vorticity graphs (second and third rows) show that the high fractional abundance also corresponds to high values of potential temperature and potential vorticity, indicating that the measurements showing the high fractional abundance of this particle type were taken in the stratosphere.



The tropopause as the boundary between tropopause and stratosphere is defined via the temperature lapse rate, but potential vorticity has been found to be a good indicator for the dynamical tropopause in the extratropics (Hoskins et al., 1985; Gettelman et al., 2011). The threshold value used to separate the stratosphere from the troposphere in the extratropics is typically 2 PVU (e.g., Holton et al., 1995), whereas this threshold value increases up to 5 PVU in the subtropics (Kunz et al., 2011). Here we find that the increase of this particle type fraction occurs at about 2 PVU during ND-MAX/ECLIF-2,

StratoClim 2016 and CAFE-Africa, at 4 PVU during ML-CIRRUS, and at 8 PVU during StratoClim 2017. In general, PV is not well suited to define the tropopause level in the tropics and therefore, a potential temperature of 380 K is typically used instead of PV to define the tropical tropopause (Holton et al., 1995). Notably during Stratoclim 2017, which took place over the AMA, the increase of the iron and magnesium containing particle fraction is found at 400 K, which is consistent with the high tropopause over the AMA. In the lowest row of Fig. 3, ozone is used as the vertical coordinate. Here, the increase of the

particle fraction starts above an ozone mixing ratio of about 150 ppbv, indicating the chemical tropopause (Hoor et al., 2002; Zahn and Brenninkmeijer, 2003; Pan et al., 2004). The different tropopause altitudes observed during the individual missions are due to the fact that the height of the tropopause is a function of latitude. The tropical tropopause corresponds to an isentropic surface at a potential temperature level of about 380 K (Holton et al., 1995), corresponding to a geometric altitude of about 17 km (Fueglistaler et al., 2009). In the extratropics, the isentropes are crossing the dynamical tropopause that lies

here between 2 and 5 PVU. At polar latitudes the tropopause height is typically around 8 km (Wilcox et al., 2012).

### 3.2 Latitudinal distribution

To combine all data from the five aircraft campaigns, we binned all particles (in total 338 354) by latitude and potential temperature, using 3° bins for latitude and 20 K bins for potential temperature. The same was done for the iron and magnesium-dominated particle type (in total 50688). Then we calculated the particle fraction of the magnesium-dominated

particle type for each bin. Only bins containing more than 10 particles were considered. The result is shown in Fig. 4 a) (separated graphs for the individual missions are given in the supplement in Fig. S6). We also inserted the thermal tropopause from the ECMWF data set, binned into 4-degree latitude bins. The median thermal tropopause is given by the thick dashed line and the 25% and 75% quartiles by the gray shaded area. Additionally, a 2 PVU and a 5 PVU surface are shown by the thin dashed lines, indicating the dynamical tropopause (2 PVU at mid latitudes and 5 PVU in the subtropics).

For this, we took all potential temperatures where the potential vorticity ranged between 1.5 and 2.5 PVU (4.5 and 5.5 PVU, respectively) and binned these values into 4-degree latitude bins.





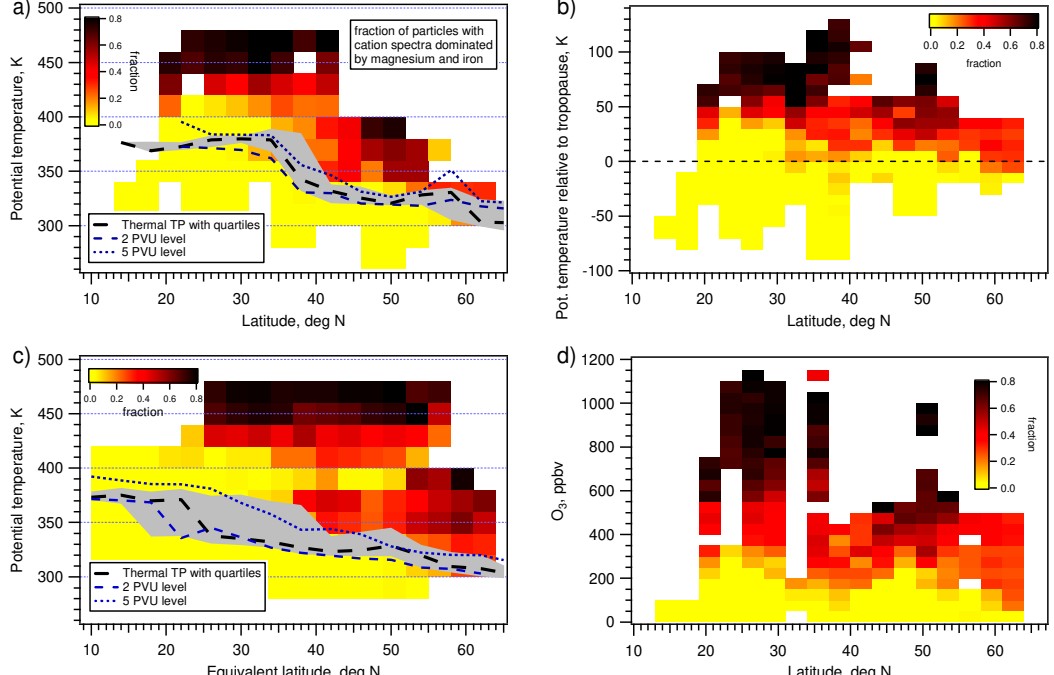

**Figure 4: Fractional abundance of particles with cation mass spectra dominated by magnesium and iron ions as a function of**
**potential temperature and latitude (a), as a function of distance to the tropopause and latitude (b), as a function of potential**
**temperature and equivalent latitude (c), and as a function of ozone and latitude (d). The data of all five UTLS aircraft missions**
**have been merged for this figure (in total 338 354 analyzed particles). Also shown in a) and c) is the median thermal tropopause**
**(from ECMWF) along with quartiles and two dynamical tropopause levels (2 PVU and 5 PVU).**


The same procedure was used for the other panels in Fig. 4. In Fig. 4 b) the potential temperature relative to the tropopause is

used as vertical coordinate with 10 K bins. Figure 4 c) uses equivalent latitude as horizontal coordinate. The equivalent

latitude of an air parcel is calculated by transforming the contour having the same potential vorticity and potential

temperature into a circle centered at the pole. The latitude enclosing this circle is then defined as the equivalent latitude.

Since potential vorticity is conserved under adiabatic processes, equivalent latitude can be used to account for reversible

adiabatic tracer transport by e.g. planetary waves (Hegglin et al., 2006; Hoor et al., 2010; Krause et al., 2018). Figure 4 d)

uses ozone as vertical coordinate, with 50 ppb ozone bins.

In geographical latitude space (Fig. 4 a), the thermal tropopause reaches the 380 K level at 34°N and remains between 370

and 380 K south of 34°N. At mid-latitudes, the tropopause height decreases until it reaches 300 K at 60°N. In equivalent





latitude space (Fig. 4c), the thermal tropopause shows more variation (larger interquartile range), especially between 20 and
40°N.

All sub-panels in Fig. 4 show that the fraction of the iron and magnesium-dominated particles increases in high and middle latitudes very close to the position of the tropopause, but not in the tropics. In theta-latitude space (Fig. 4 a), the particle fraction remains as low as in the troposphere between the tropopause (around 370 - 380 K) and 400 K at latitudes south of

30°N. Normalizing the potential temperature to the thermal tropopause (Figure 4b) confirms this observation. In theta-equivalent latitude space (Fig. 4 c), this effect is even more pronounced: South of 35°N equivalent latitude, the area between the tropopause and 420 K shows a very low fraction of the iron and magnesium-dominated particles. This corresponds to the PV profile of the StratoClim 2017 data from Figure 3, because the stratospheric tropical data in Fig. 4 are dominated by the StratoClim 2017 data set. In the AMA, which dominated the geographical region of StratoClim 2017 during the time of the

campaign, the air masses are transported upwards between about 360 K and 460 K (Ploeger et al., 2017; Vogel et al., 2019). The observation that the fraction of the iron and magnesium-dominated particle type increases only above the extratropical tropopause layer or mixing layer (Hoor et al., 2002; Hoor et al., 2004; Pan et al., 2004), i.e. 30 K above the tropopause (Fig. 4 b), indicates that the source for this particle type must be above the tropopause, because otherwise, the upwelling air masses in the AMA would contain this particle type also at lower potential temperatures. In the stratosphere, the widespread

occurrence of high fractions of this particle type over a broad range of equivalent latitudes above 440 K (Fig. 4 c) indicates that this particle type is very homogeneously distributed in the stratosphere. The large equivalent latitude range is consistent with potential transport between high and low latitudes. From Fig. 4 b) it can be seen that at above a distance of about 40 K to the tropopause the proportion of the iron and magnesium-dominated particles does not change substantially with latitude. In Fig. 4 d) similar behavior is observed for ozone levels larger than 300 ppbv.

**3.3 Interpretation as meteoric particles**

From the previous discussion we concluded that the source of this particle type is likely found above the tropopause. The capacity to record bipolar ion spectra of single particles allows us to show that each particle whose cation mass spectrum is dominated by Mg and Fe contains sulfuric acid but no other frequently observed anions like $NO^-$, $NO_2^-$, $CN^-$, or $CNO^-$. We therefore conclude that the particles we observe consist of meteoric material dissolved in sulfuric acid. This interpretation is

fully consistent with the argumentation by Murphy et al. (1998) and Cziczo et al. (2001) who measured stratospheric particle composition using a similar laser ionization mass spectrometer (PALMS) on board the WB-57F high altitude research airplane between 5 and 19 km altitude. Additional PALMS measurements from other campaigns (Cziczo et al., 2001; Cziczo et al., 2004; Murphy et al., 2007; Murphy et al., 2014) as well as laboratory measurements with reference meteoric samples and artificial meteorite particles supported the conclusion that the stratospheric particles with mass spectra dominated by Mg

and Fe consist of meteoric material dissolved in sulfuric acid (Cziczo et al., 2001). Our cation mass spectra (Fig. 2) show a very similar ion signature as the cation mass spectra from stratospheric particles, dissolved meteorites and artificial meteorite particles presented in Cziczo et al. (2001). The finding that Si is observed to a much lesser degree than expected from

meteoric composition (roughly equal amounts of Fe, Mg, and Si) was explained by Cziczo et al. (2001) and Murphy et al.

(2014) by the low solubility of $SiO_2$ in $H_2SO_4$. Thus, Si is assumed to be present as a solid inclusion in the particles and is

thereby less efficiently ionized compared to the other metals that are dissolved in $H_2SO_4$.

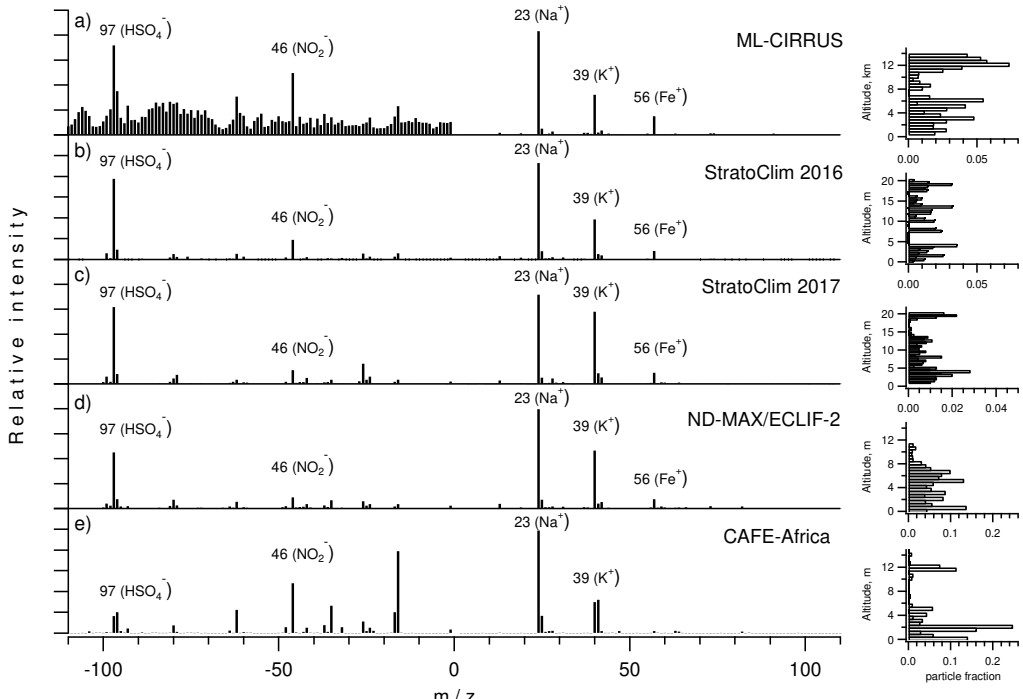

**Figure 5.** Mean mass spectra and vertical profiles of a particle type containing Na, K, and Fe, with smaller amounts of Mg and Ca.
**This type, which was observed in all five high altitude aircraft missions, does not belong to the meteoric particles, although it was**
**sometimes observed at higher altitudes. It can be interpreted as mineral dust, internally mixed with sea spray and secondary**
**inorganic compounds (nitrate, sulfate). Note that during ML-CIRRUS the anion spectra were noisy due to problems with the high**
**voltage supply of the ALABAMA.**

Other sources for this particle type, like aircraft or rocket exhaust, uplifting of particles (e.g. desert dust) from the Earth's

surface and volcanic injection, can be ruled out. The majority of aircraft traffic does not occur at such high altitudes at which

the meteoric particles were observed during the StratoClim campaigns. Rocket exhaust can be ruled out because the

dominating metal in rocket exhaust particles is expected to be aluminium (Voigt et al., 2013). Single particle mass

spectrometric measurements of rocket exhaust plumes showed ions of chlorine and oxygen, of metals like Al, Fe, Ca, Na,





and K, but not of magnesium (Cziczo et al., 2002). Furthermore, rocket exhaust plumes would hardly lead to the observed

uniform and wide geographical distribution of the particle fraction. Volcanic aerosol particles have been measured in the tropopause region and lowermost stratosphere after eruptions of Kasatochi and Sarychev (Andersson et al., 2013). These data show that volcanic aerosol particles contain a larger weight percentage of carbonaceous material than of ash, which is not reflected in our measurements. Furthermore, volcanic ash particles indeed contain a number of elements that are abundant in meteorites, like Fe, Si, Ca, K, but additionally also elements that are characteristic for crustal material like

titanium which was not observed in our mass spectra. As crustal material that can occur as particles in the troposphere (like soil dust or desert dust) contains the same elements like the stratospheric particles we observed (e.g., Na, Mg, Al, K, Fe), interferences with dust particles in the troposphere might be possible, although the ions $FeO^+$ and $FeOH^+$ (m/z 72, 73) have not been observed in single particle spectra of mineral dust (Gallavardin et al., 2008). In the tropical regions, uplifting of particles from the troposphere into the stratosphere occurs especially in the AMA (Randel et al., 2010; Pan et al., 2016; Yu et

al., 2017) and might also carry dust particles into the stratosphere. However, to explain the stratospheric abundance-fraction of the observed Fe- and Mg-rich particle type, this particle type would need to be found already during the upward transport in the AMA, which is clearly not the case (Fig. 4). The mean mass spectra and the vertical profiles of another prominent particle type containing Fe, K, Na, as well as smaller signals of Mg and Ca, is shown in Fig. 5. This particle type was occasionally observed in the stratosphere (ML-CIRRUS, StratoClim 2017, CAFE-Africa), but in general occurred mainly in

the troposphere. We interpret this particle type as an internal mixture of mineral dust, sea spray, sulfate, and nitrate, due to $Na^+$, $K^+$, and $Fe^+$ cations and chlorine ($^{35}Cl^-$, $^{37}Cl^-$), nitrate ($NO^-$, $NO_2^-$), and sulfate ($SO^-$, $SO_2^-$, $SO_3^-$, $HSO_4^-$) anions. It was therefore not included in the meteoric data set discussed in this paper. The reason why such particles were found in the stratosphere during ML-CIRRUS is presumably an outbreak of Saharan dust and its transport towards Europe during the time of the campaign (Weger et al., 2018). During StratoClim 2017 and CAFE-Africa, vertical uplifting of such particles of

tropospheric origin into the stratosphere can most likely be explained by the Asian and African monsoon systems.



### 3.4 Size-resolved fraction of meteoric particles

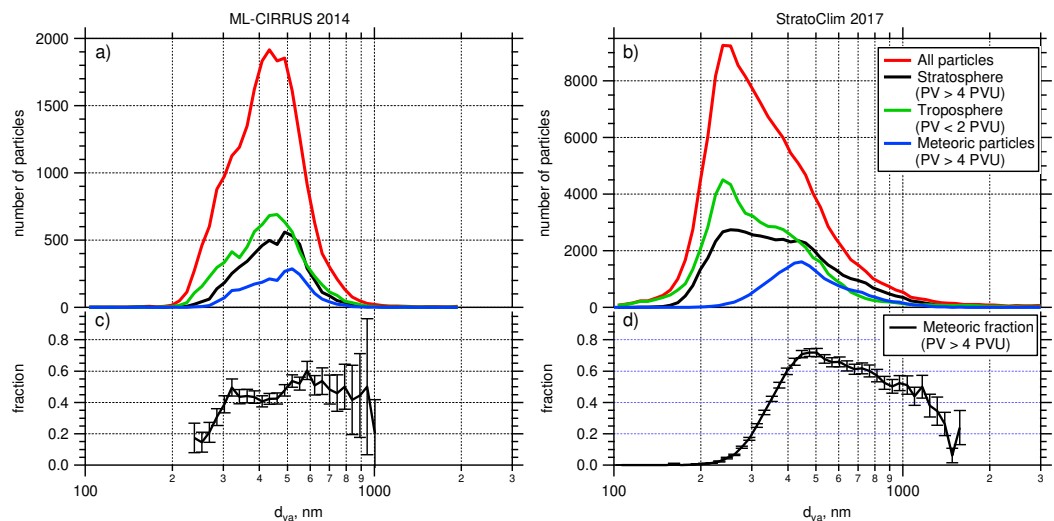

**Figure 6. Number of analyzed single particles as a function of particle size (vacuum aerodynamic diameter, dva) measured during ML-CIRRUS 2014 (a, c) and StratoClim 2017 (b, d). Panels a) and b) show absolute number of counted particles per size bin; panels c) and d) show the fraction of particles with meteoric composition signature in the stratosphere (PV > 4 PVU). Error bars were calculated from Poisson counting statistics (number of particles per size bin) and were propagated for the particle fraction.**

Both particle mass spectrometers used here (ALABAMA and ERICA) determine the particle velocity in the vacuum chamber which by laboratory calibration can be converted into the vacuum aerodynamic diameter ($d_{va}$, DeCarlo et al. (2004)) of each individual particle. To obtain the size distributions shown in Fig. 6 a) and b), we used logarithmically equidistant size bins between 100 and 2000 nm (ML-CIRRUS) and 100 and 5000 nm (StratoClim). These size distributions represent the product of instrument efficiency (inlet transmission, particle detection and ablation rate) and the ambient particle size distribution. Differences between the measurements with the ALABAMA during ML-CIRRUS and the ERICA during StratoClim 2017 are therefore mainly due to the aforementioned differences in instrumental performance. The particle sizes were separated between tropospheric (PV < 2 PVU) and stratospheric (PV > 4 PVU) conditions.

In both data sets, the tropospheric particles (green lines) tend to be smaller than the stratospheric particles (black lines). Panels a) and b) in Fig. 6 also depict the size distribution of the meteoric particles, and panels c) and d) show the ratio between the meteoric particles (also selected for stratospheric conditions) and all stratospheric particles. It turns out that the fraction of meteoric particles is lowest in the smaller-particle size range for both campaigns: In the ML-CIRRUS data set, the fractional contribution increases from about 0.2 at 250 nm to about 0.5 at 300 nm and remains almost constant at 0.5 up to



1000 nm. The StratoClim data set extends both to smaller and larger sizes and contains a larger number of particles. Here it can clearly be seen that the fraction of meteoric particles is zero at 200 nm, although stratospheric particles are detected even

below 200 nm. The meteoric fraction rises to 0.7 at 450 nm and decreases above that size, down to 0.2 at about 1600 nm. Above that size, only one meteoric particle was detected, although in total 253 stratospheric particles were measured between 1600 and 4400 nm. Thus, the meteoric fraction appears to decrease down to zero above $d_{va} \approx 1600$ nm. This finding is similar to the data shown by Murphy et al. (2014) who found a maximum of meteoric particles at diameters of around 600 – 700 nm and a decrease down to zero above $d_{va} = 1$ µm. However, the fraction of meteoric particles below 600 nm is

markedly higher in our data set compared to the study of (Murphy et al., 2014). In their data set, the fraction of meteoric particles decreases from 0.2 at 600 nm to zero at 500 nm, and no meteoric particles were detected below 500 nm diameter. The observed size range of the meteoric particles between about 250 and 1500 nm indicates that their sedimentation may play an important role for the downward transport of meteoric material through the stratosphere (see Section 4). Once the meteoric aerosol particle material has reached altitude levels near the tropopause, its rapid removal out of the stratosphere

due to cross-tropopause exchange and cloud formation processes is likely.

### 3.5 Particle number concentration

It is difficult to estimate an absolute number concentration of particles containing meteoric material from the measured particle fraction with a laser ablation mass spectrometer. The main reason is that pure sulfuric acid particles are not ablated and ionized by a laser with a wavelength of 266 nm, because sulfuric acid has a very low absorption cross section for

wavelengths larger than about 190 nm up to visible light (Thomson et al., 1997; Burkholder et al., 2000; Murphy, 2007). Thus, the fraction of particles containing meteoritic material will be overestimated if pure sulfuric acid aerosol particles existed in the air.

The hit rate of the mass spectrometer, which is defined here as the number of acquired mass spectra per time unit divided by the number of laser shots per time unit, can be used to estimate the number of missed particles. Our data show that the hit

rate in the stratosphere is generally lower than in the lower troposphere. Two examples (for ML-CIRRUS and CAFE-Africa) are shown in Fig. S8 in the supplement. The maximum achieved hit rate in the troposphere was about 0.8 during CAFE-Africa, whereas the averaged hit rate in the stratosphere was about 0.2, thus, lower by a factor of 4. A similar decrease, albeit at lower absolute values of the hit rate, was observed during ML-CIRRUS. As a conservative approach, we assume here that the decrease of the hit rate in the stratosphere is only due to the abundance of pure sulfuric acid particles that are not ablated.

Then we can estimate the absolute number of meteoric particles in two steps: First, by dividing the fraction measured by the mass spectrometer by a factor of 4 (to account for the hit rate decrease) and second, by multiplying by the total particle number concentration measured using an independent absolute particle counting (and sizing) instrument. The second step is similar to previous approaches (Qin et al., 2006; Gunsch et al., 2018; Froyd et al., 2019).



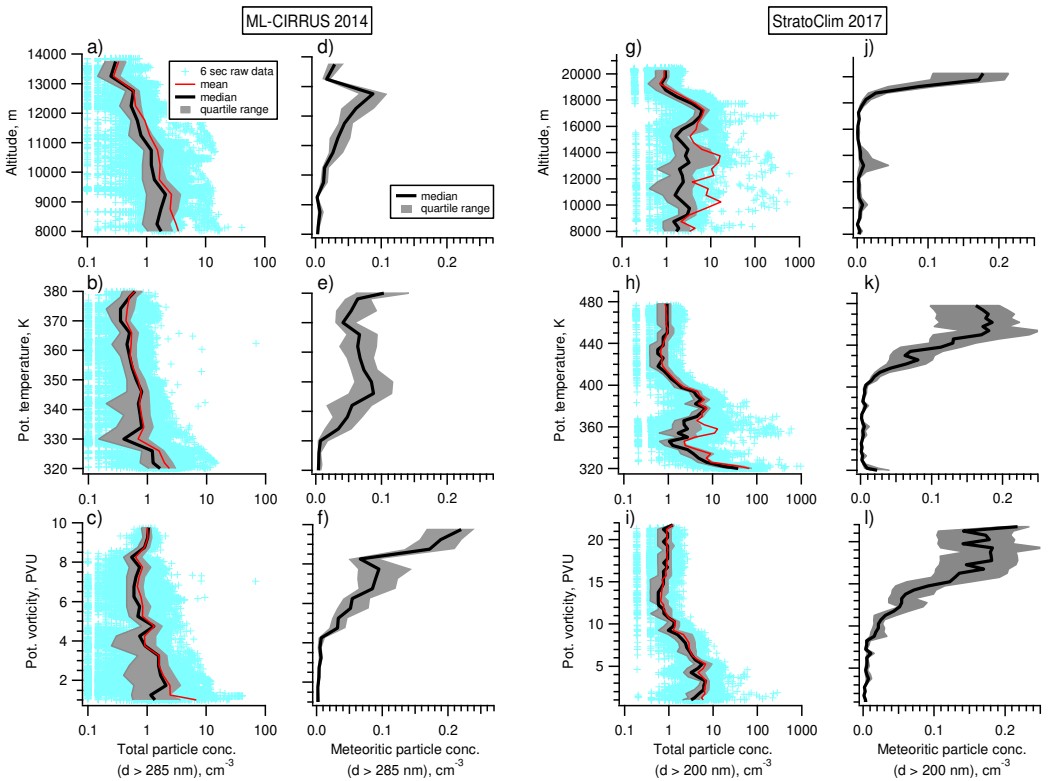


**Figure 7.** Total particle number concentrations measured during ML-CIRRUS 2014 ($d_{ve}$ > 285 nm, a) to c)) and StratoClim 2017 ($d_{ve}$ > 200 nm, g) to i)) along with calculated number concentrations of particles containing meteoric material (ML-CIRRUS 2014: d) to f); StratoClim 2017: j) to l)). Data are shown for the upper troposphere and lower stratosphere (Altitude > 8 km, potential temperature > 320 K, PV > 1 PVU). Light blue markers: 6 second raw data; red line: mean; black line: median; grey area:
quartiles (25% and 75%).

For ML-CIRRUS, we used the optical particle spectrometer "Sky-OPC" (Grimm 1.129). The nominal lower cut-off diameter
(manufacturer calibrated with PSL particles) is 250 nm. To account for the refractive index of stratospheric particles, we
performed Mie calculations for refractive indices between 1.43 and 1.45 (Yue et al., 1994). This resulted in a lower cut-off
diameter for stratospheric aerosol particles of 285 nm in diameter (supplement, Fig. S11). The size distributions in Fig. 6
show that for ML-CIRRUS the meteoric fraction is approximately constant between vacuum aerodynamic diameters greater
than 300 nm. This value translates into a volume equivalent diameter ($d_{ve}$) of about 180 nm, assuming a density of the lower
stratospheric particles of 1.7 g cm$^{-3}$ (Yue et al., 1994). We also note that the size distribution showed that 99.8 % of all
particles counted by the OPC in the stratosphere are below 1000 nm. Thus, we can assume a constant fraction of meteoric





particles for the particles counted by the OPC and therefore multiplying the binned meteoric particle fraction (divided by 4) from Fig. 3 with the binned number concentration measured by the OPC should give an estimation of the absolute concentration of meteoric particles larger than 280 nm for the mid-latitude data set from ML-CIRRUS 2014.

For StratoClim 2017, we used data recorded by the UHSAS (DMT Inc.). According to the size distribution of meteoric particles in Fig. 6, the meteoric particles fraction reaches about 50% of its maximum fraction at 340 nm ($d_{va}$). This translates

into a volume equivalent diameter ($d_{ve}$) of 200 nm (assuming the same density for stratospheric aerosol particles as above). Mie calculations using the refractive index range from 1.43 to 1.45 (Fig. S11) yield that a lower size limit of 180 nm (PSL calibration) corresponds to a $d_{ve}$ of 200 nm for stratospheric aerosol particles. We therefore used the integrated particle number concentration between 180 nm and 1000 nm (PSL calibration), multiplied this with the fraction of meteoric particles from Fig. 3 and divided by 4 to account for the hit rate. This procedure gives an estimation of the absolute concentration of

meteoric particles larger than 200 nm for the tropical data set from StratoClim 2017.

Figure 7 shows the total particle concentrations for the two missions named above as a function of altitude, potential temperature and potential vorticity for the upper troposphere and lower stratosphere. The 6 second raw data are shown along with binned mean, median, and quartiles. The calculated meteoric particle concentrations are shown as binned median values with quartiles. The highest absolute number concentrations of meteoric particle range around 0.2 cm$^{-3}$ (referring to ambient

pressure and temperature). During StratoClim 2017 these values are reached above 20 km, potential temperature = 50 K, and 17 PVU. During ML-CIRRUS, values of 0.2 cm$^{-3}$ are only reached at PV > 9 PVU, whereas in altitude and potential temperature coordinates the concentrations reach only 0.1 cm$^{-3}$. Nevertheless, the absolute range of meteoric particle concentration is very similar for both data sets, although the calculation of the meteoric particle concentration relies on different size ranges of the optical instruments and is based on several assumptions, as detailed above.

**3.6 Transport mechanism for cross-tropopause exchange**

To investigate the downward transport of meteoric particles through the tropopause into the troposphere, we use the tracer-tracer correlation of ozone as a stratospheric tracer and water vapor as a tropospheric tracer. Tracer-tracer scatter plots have been widely used to identify mixing between troposphere and stratosphere (Fischer et al., 2000; Hoor et al., 2002; Pan et al., 2004; Marcy et al., 2007; Gettelman et al., 2011; Krause et al., 2018). In such scatter plots irreversible tracer exchange

shows up as lines connecting the respective mixing ratios of the initial unmixed reservoir air parcels and are termed mixing lines (Hoor et al., 2002). Occurrence of mixing lines is a clear indication for mixing between stratospheric and tropospheric air. It is more common to use carbon monoxide as a tropospheric tracer, but because it was not measured during ML-CIRRUS, we use water vapor for which the applicability to serve as a tropospheric tracer in tracer-tracer correlations has been shown by Gettelman et al. (2011), Pan et al. (2014) and Heller et al. (2017). High ozone values indicate stratospheric

air (vertical branch), high $H_2O$ values tropospheric air (horizontal branch).

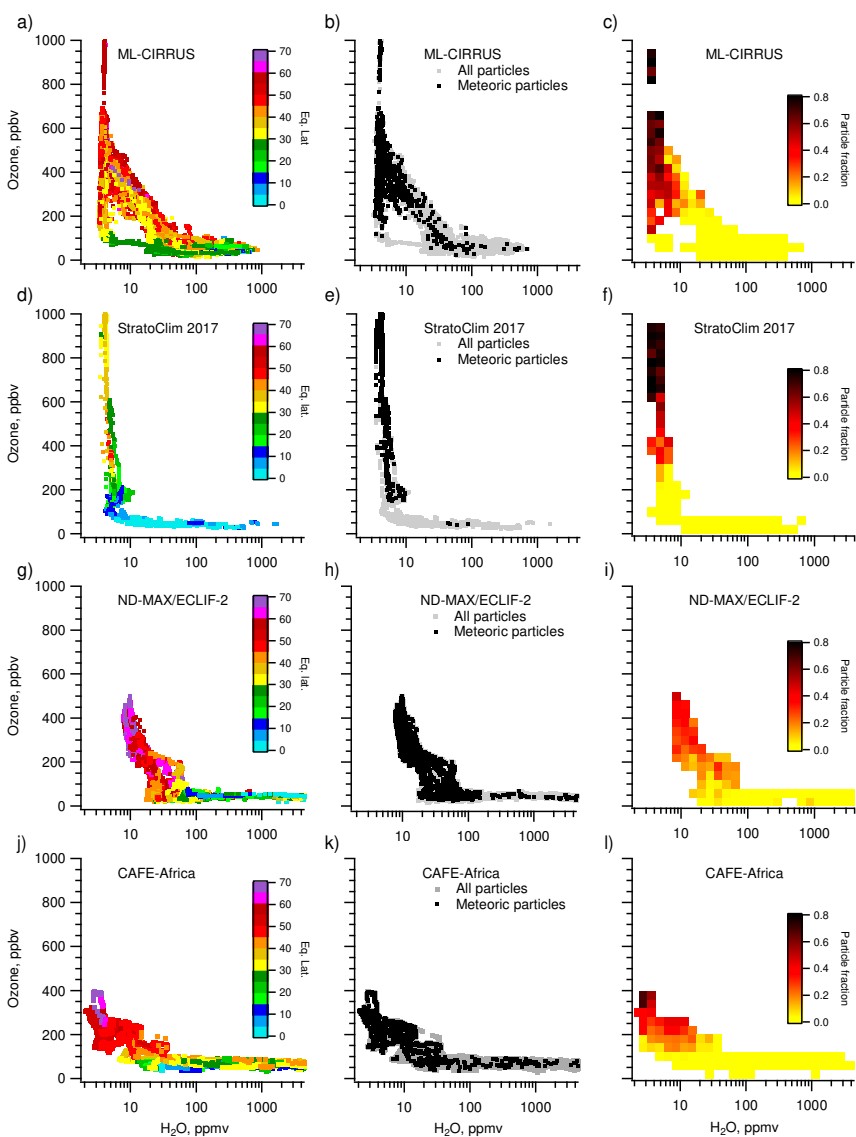

**Figure 8. Ozone mixing ratio as stratospheric tracer versus water vapor mixing ratio as tropospheric tracer color-coded with**
**equivalent latitude, for ML-CIRRUS (a), StratoClim 2017 (d), ND-MAX/ECLIF-2 (g), and CAFE-Africa (j). Panels b), e), h), and**
**k) show $O_3$ and $H_2O$ for the times when particle mass spectra were recorded (gray: all particles, black: meteoric particles). Panels**
**c), f), i), and l) show the proportion of meteoric particles within a $O_3$-$H_2O$ grid.**



Figure 8 shows the tracer-tracer correlations between ozone and $H_2O$ for ML-CIRRUS, StratoClim 2017, ND-MAX/ECLIF-2, and CAFE-Africa. The data coverage of $O_3$ and $H_2O$ during StratoClim 2016 was not sufficient (see supplement, Fig S10). The left panels show all data from the trace gas measurements, color coded by equivalent latitude.


The mid-latitude data from ML-CIRRUS (Fig. 8 a) show a clear separation between air masses of mid latitude and tropical origin. The mixing lines, indicating irreversible mixing between the troposphere and the stratosphere have equivalent latitudes > 30°N, whereas the green colored data points that correspond to tropical air masses (equivalent latitude < 30°N) do not show such mixing. Fig. 8 b) shows the $H_2O$ and $O_3$ data for all sampled particles (gray) and for all meteoric particles (black). As expected, the density of black data points is highest in the stratospheric branch of the tracer-tracer correlation.


Figure 8 c) shows the fraction of meteoric particles within an ozone-water-vapor grid. Mixing between extratropical stratospheric and tropospheric air is indicated by mixing lines with equivalent latitudes > 30°N, connecting regions of elevated extratropical and low stratospheric $H_2O$ values. Isentropic mixing between dry air which passed the Lagrangian cold point (and therefore exhibits $H_2O$ mixing ratios < 6 ppmv) and higher latitudes is indicated by the vertical branch starting at $O_3$ mixing ratios < 150 ppbv, connecting the dry upper tropical troposphere with the stratosphere.


In the StratoClim 2017 data set (Fig. 8 c), d), e)) no mixing lines were observed. Only very few meteoric particles are observed in the tropospheric branch of the $O_3$-$H_2O$ plot (below 100 ppbv $O_3$), showing that downward mixing of meteoric particles from the stratosphere does not occur in the upwelling tropical air masses of the AMA (see also Fig. 4).

The data sets of ND-MAX/ECLIF-2 (Fig. 8 f), g), h)) and CAFE-Africa (Fig. 8 i), j), k)) appear similar in this tracer-tracer correlation, although the geographic latitudes and seasons of the two campaigns were very different. In both missions, highest observed $O_3$ values are 400 – 500 ppbv, and the equivalent latitudes reach up to 60 – 70°N in the stratosphere. Both data sets show a high degree of stratosphere-troposphere mixing, as can been seen from the higher $H_2O$ mixing ratios at $O_3$ levels between 100 and 200 ppbv, corresponding to the mixing lines observed during ML-CIRRUS. Along these mixing lines, meteoric particles are frequently observed, even at tropospheric altitudes where water vapor mixing ratios of > 1000 ppmv are reached.



**3.7 Detection of particles containing meteoric material at low altitudes**

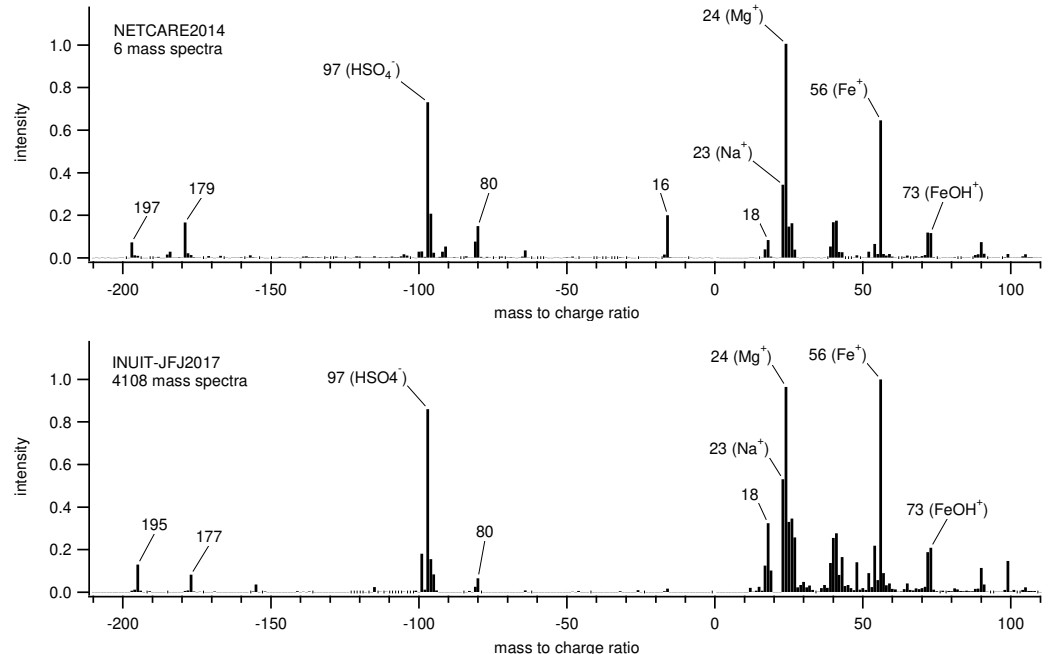

**Figure 9: Mass spectra from low altitudes (NETCARE, Canadian Arctic, summer 2014, up to 3 km; Jungfraujoch, Switzerland, winter 2017, 3600 m) showing meteoric signatures.**


Figure 8 showed that particle containing meteoric material are transported downwards through stratosphere-troposphere exchange and are therefore also present in the troposphere, albeit at low concentrations and low number fractions. We used two data sets from lower altitudes to estimate the occurrence of this particle type in the middle and lower troposphere. These are the abovementioned NETCARE data set (Canadian Arctic, summer 2014) that contains aircraft-based ALABAMA data up to 3 km altitude (Köllner et al., 2017) and a mountain-top data set from the Jungfraujoch station at 3600 m altitude in

winter 2017. In both data sets the relative number of meteoric particles was very low, such that an automated cluster algorithm would not find this particle type unless the prescribed number of cluster would be set to very high values. Thus,





we used the most prominent mass spectral features from this particle type as observed in the stratosphere (Fig. 2) and scanned the two low-altitude data sets for these marker peaks. The criteria included the presence of $^{24}$Mg, $^{25}$Mg, $^{26}$Mg, $^{54}$Fe, $^{56}$Fe, the absence of Cl (to exclude sea spray) and signal intensity of m/z 39 smaller than that of m/z 41. The latter was

introduced to minimize the influence of potassium from other sources, especially dust. By varying these search criteria, different numbers of mass spectra with similar average mass spectra were obtained, such that the absolute amount of meteoric particle at low altitude is highly uncertain. Figure 9 shows the averaged mass spectra matching the criteria given above. The spectra correspond very well to the spectra sampled in the stratosphere (Fig. 2). A higher contribution of m/z 18 ($NH_4^+$), especially in the Jungfraujoch spectra indicates a higher degree of neutralization of the sulfuric acid by ammonia in

the troposphere than in the stratosphere. During NETCARE, six out of about 10000 particle mass spectra matched the criteria. By changing the criterion for potassium to an absolute upper intensity threshold, the number of spectra was reduced to three. Thus, the fraction of meteoric particles found in the summer Arctic lower troposphere can be estimated to be around 0.025 – 0.05%. In the free tropospheric data set obtained in winter at the Jungfraujoch, about 4100 spectra (out of more than 765000) matched the criteria, corresponding to 0.5%. Also here, by varying the criteria the percentage varies between 0.2

and 1 %. This range is clearly larger than that in the Arctic summer, but it has to be kept in mind that the Jungfraujoch data set was obtained at 3600 m altitude, whereas the measurements during NETCARE only reached up to 3000 m. In winter the Jungfraujoch station is mainly located in the free troposphere (over 60% of the time, see Bukowiecki et al. (2016)), such that the influence of boundary layer particles is low. In contrast, the aerosol particles in the summer Arctic during NETCARE were to a large degree influenced by particles from marine biogenic origin (Köllner et al., 2017). Backtrajectory calculations

for the Jungfraujoch data set showed that the fraction of detected meteoric particles was higher during times when the air masses experienced higher altitudes and higher latitudes during the 5 days before the measurements (supplement, Section 9 and Fig. S12). Additionally, the fraction of meteoric particles followed the time trend of the ozone mixing ratio (Fig. S12), confirming the stratospheric origin. Overall, this shows that the meteoric material immersed in stratospheric sulfuric acid aerosol particles reaches the lower troposphere from where it will be removed by wet removal (rain-out, wash-out), thereby

finally reaching the Earth's surface. This is confirmed by a number of studies that reported the detection of meteoric material in ice cores samples from Greenland (Gabrielli et al., 2004; Lanci et al., 2012) and Antarctica (Gabrielli et al., 2006).

**4 Discussion and conclusion**

In this study we present stratospheric single particle mass spectrometer data from five aircraft-based campaigns, covering a wide range of northern hemispheric latitudes (15°N – 68°N) and seasons (winter, spring, summer). In all data sets a distinct

particle type characterized by iron and magnesium was observed in the stratosphere. The observed distribution as function of potential temperature and potential vorticity suggests that the source of this particle type is above the tropopause. From previous stratospheric data (Mossop, 1965; Murphy et al., 1998; Cziczo et al., 2001; Murphy et al., 2014), meteoric composition data (Lodders and Fegley Jr., 1998; Rapp et al., 2012; Plane et al., 2015), and theory of meteoric ablation and





fragmentation (Plane, 2003; Carrillo-Sánchez et al., 2016; Subasinghe et al., 2016), it was concluded that this particle type

represents meteoric material partially dissolved within sulfuric acid solution droplets.

Downward transport of MSP particles from the mesosphere into the lowermost stratosphere occurs efficiently in the polar vortex (Curtius et al., 2005; Weigel et al., 2014). These papers show that high altitude aircraft measurements demonstrate there is a higher proportion of refractory particles (60-70%) within the wintertime polar vortex, and one would therefore expect to find a higher abundance of meteoric particles in the lower stratosphere at high latitudes during late winter and early

spring than at lower extra-tropical latitudes and than in other seasons. This is not confirmed by our observations: Although two mid-latitude campaigns (ML-CIRRUS and ND-MAX/ECLIF-2) were conducted between January and April, we observe the same fraction of meteoric particles at the same ozone levels in the lower stratosphere (Fig. 3) during all campaigns, regardless of latitude and season.

Satellite observations with the CALIOP lidar instrument have shown that the lower edge of the stratospheric aerosol layer

lies between 450 and 500 K potential temperature for latitudes between 20 and 50°N (Vernier et al., 2009). It is important to note however, that Aitken-sized particles (a few tens of nanometers, e.g. Cadle and Kiang, 1977) are present throughout the upper troposphere and lower stratosphere, so this altitude range from the CALIOP measurements refers only to the optically-interacting aerosol population. Thus, we conclude that the particles containing meteoric material that we observed in the lower stratosphere (with diameters greater than 200 nm) originate from the stratospheric aerosol layer. Our measurements

between 20 and 40°N reach up to the lower edge of the stratospheric aerosol layer (see Fig. 4), whereas at higher latitudes and lower altitudes we assume that we observed particles that settled gravitationally from the stratospheric aerosol layer. This in turn means that the stratospheric aerosol layer particles contain meteoric material at all latitudes. We therefore conclude that meteoric material is indeed carried within the winter polar vortex from the mesosphere to the stratosphere, but that isentropic mixing above 440 K potential temperature (see Fig. 4) distributes the meteoric material over all latitudes.

Isentropic mixing in the stratosphere within the extratropics, but also between the tropics and extratropics, has been described previously (Neu and Plumb, 1999; Garny et al., 2014). From our data we can infer that at the lower edge of the stratospheric aerosol layer (above 440 K potential temperature) the meteoric material is equally distributed throughout the latitude range of about 20 to 60°N. Our observations do not give a clear indication whether the detected particles containing meteoric material originate from meteor smoke particles (MSP) dissolved in stratospheric aerosol layer particles or from

meteoric fragments (MF) or unablated interplanetary dust particles (IDP) that are coated by sulfuric acid. However, the high $H_2SO_4$ content of all detected meteoric particles and the uniform mass spectra suggest that MSP dissolved in sulfuric acid are the most likely particle source.

We calculated the terminal settling velocity for particles of different sizes and densities (pure $H_2SO_4$, $\rho$ = 1.83 g cm$^{-3}$, and pure olivine as a surrogate for meteoric composition, $\rho$ = 3.30 g cm$^{-3}$) as a function of altitude (for details see supplement).

Between 16 and 18 km, the settling velocity ranges between 1 and 12 m/day for particles between 100 and 500 nm having the densities given above. In the AMA, air masses are transported upwards between about 360 K and 460 K with about 1.5 K/day (Ploeger et al., 2017; Vogel et al., 2019), corresponding to about 35 – 40 m / day. This is larger than the above





calculated range, thus sedimentation is not fast enough to overcome the Asian monsoon upward motion. This explains that in the tropics we observe the increased fraction of particle containing meteoric material only 30 K above the thermal

tropopause (see Fig. 4) whereas in the extratropics, where little upward motion occurs, we see these particles directly above the tropopause.

Our data further show that all meteoric particles contained $H_2SO_4$, but no other anions like nitrate or organic material. Thus, from our simultaneous cation and anion measurements we can confirm previous assumptions that Mg and Fe are dissolved in $H_2SO_4$ (Murphy et al., 1998; Cziczo et al., 2001; Murphy et al., 2014). This suggests that these particles act similar as pure

$H_2SO_4$ droplets in the UT with respect to cirrus formation and also in the polar stratosphere with respect to PSC formation. In general, this particle type represents a good tracer for stratospheric aerosol particles. Downward transport along mixing lines at mid-latitudes was clearly identified, but only for equivalent latitudes above 30°N. In data sets acquired in the lower troposphere the meteoric composition-signature particles were detected as well, albeit only in very minor fractional abundance. Their size and composition (larger than 200 nm, composed mainly of $H_2SO_4$, most likely neutralized by

ammonia in the troposphere) makes them ideal CCN, such that they will be efficiently removed from the atmosphere by nucleation scavenging and wet removal and the meteoric material is by these processes transported to the Earth's surface. We re-iterate that our findings of relatively invariant meteoric-particle-fraction are for particles larger than 200 nm. The data from the Arctic campaigns (Curtius et al., 2005; Weigel et al., 2014) are for particles larger than 10 nm, including also stratospheric Aitken-sized particles, which will have a different abundance-fraction of particles containing meteoric material.

The analysis presented here, combining data obtained with two different laser ablation mass spectrometers during five aircraft missions conducted at different seasons, latitudes, and altitudes, as well as two low altitude data sets, has confirmed the widespread occurrence of meteoric material in stratospheric aerosol particles. Using the particles containing meteoric material as a tracer for stratospheric transport, our observations confirm the upward motion of air masses over the Asian monsoon anticyclone and the associated transport into the stratosphere, the exchange between stratosphere and troposphere

in the extratropics as well as the efficient isentropic mixing between high and low latitudes above 440 K potential temperature.

#### Data availability

The data shown in this study are available at Edmond – the Open Access Data Repository of the Max Planck Society, under the following permanent link: https://dx.doi.org/10.17617/3.38

#### Author contribution

JS evaluated the data, compiled the figures, and drafted the manuscript with contributions by RW. PHOO contributed significantly to the discussion on cross-tropopause mixing. Single particle mass spectrometer data were provided by OA,





AH, AD, SM, SB (ERICA) and JS, TK, FK, OE, HCC (ALABAMA). JUG provided meteorological re-analyses. CM provided UHSAS data. AZ, FO, HS, MS, FR, AU provided ozone data. MK, CR, MZ, GSD, JPD, JBN provided water vapor
data. PHOP provided information on meteoric composition. The manuscript was critically reviewed by RW, SB, OA, FK, HCC, AH, OE, PHOP, PHOO, MK, CR, JUG.

**Competing interests**

The authors declare no competing interests

**Acknowledgements**

This work was funded by the DFG Priority Program SPP 1294, grant SCHN1138/1-2 (ML-CIRRUS), by the European Research Council (ERC), EU FP7/2007–2013, Projects No. 603557 (StratoClim), and No. 321040 (EXCATRO). StratoClim was also supported by BMBF under the joint ROMIC-project SPITFIRE (01LG1205A). CAFE-Africa was funded by the Max Planck Society. ND-MAX was funded by the NASA Advanced Air Vehicles Program (program manager J. Dryer). Funding for NETCARE was provided by the Natural Sciences and Engineering Research Council of Canada through the
NETCARE project of the Climate Change and Atmospheric Research Program. The measurements at Jungfraujoch were funded by the DFG grant SCHN1138/2-2 (FOR1525 "INUIT") and by the EU Horizon 2020 research and innovation programme, grant No. 654109 (ACTRIS-2). We also acknowledge that the International Foundation High Altitude Research Stations Jungfraujoch and Gornergrat (HFSJG), 3012 Bern, Switzerland, made it possible for us to carry out our experiment at the High Altitude Research Station at Jungfraujoch. The Swiss Federal Laboratories for Materials Science and Technology
(EMPA) is acknowledged for providing ozone data for the Jungfraujoch through the EBAS/EMEP data base. We acknowledge the NOAA Air Resources Laboratory (ARL) for the provision of the HYSPLIT transport and dispersion model and/or READY website (http://www.ready.noaa.gov) used in this publication. We thank the technical and scientific coordinators of the aircraft campaigns: C. Voigt, A. Minikin, U. Schumann (ML-CIRRUS), J. Lelieveld, H. Fischer, J. Williams, M. Dorf (CAFE-Africa), B. Anderson, T. Moes (ND-MAX/ECLIF-2), F. Stroh, M. Rex, F. Cairo (StratoClim), A.
Herber, J. Abbatt, R. Leaitch (NETCARE). We also thank all aircraft crews, campaign teams and hangar staff (especially the staff of 120 ATW (Air Training Wing) at Kalamata), as well as the technical staff at MPIC for support during instrument development and operation during field campaigns.



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
