# Peer review of "Aircraft-based observation of meteoric material in lower stratospheric aerosol particles between 15 and 68°N"

_Atmospheric Chemistry and Physics, 2020_

## Referee Comment (RC1) · Anonymous Referee #1 · 7 Sep 2020

The manuscript submitted presents measurements of meteoric material, identified by its elemental composition, in atmospheric aerosol in the lower stratosphere and in some locations in the troposphere. Observations are presented from a variety of campaigns at a range of latitudes, altitudes and seasons.

Scientific significance:

Measurements of this type in the lower stratosphere are not entirely new, as acknowledged by the authors. However, the statistical analysis made possible by the size of this dataset leads to conclusions regarding trends in the atmospheric abundance of these aerosol which is a new and valuable contribution to the literature. In addition the

observation of these particles in the troposphere provides evidence of the occurrence of transport processes which have been previously speculated. I feel that by neglecting aspects of the literature the authors have underestimated the value of their work, and hope to assist in my suggestions below.

Scientific quality:

The scientific approach seems sound and appropriate to the stated aims. In my comments below I suggest several further details which might be usefully discussed.

Presentation quality:

On the whole I find the presentation to be of a standard suitable for publication. I do have some suggestions to improve the readability and effectiveness of the figures. The written English is understandable to me as a native English speaker, though it does use some non-standard (German) sentence structure. I have suggested only typographical language changes.

General comments:

It is my opinion that addressing the following issues will improve the manuscript as presented. I believe these to be minor changes, but acknowledge that some may be more complex than they seem to me. I advise the editor to accept reasonable explanations of why some of my recommendations may not be practical. The most significant change I believe is required is to broaden and better support the scope of the study by including aspects of the literature on meteoric smoke and fragmentation which have been overlooked. This has implications at various points in the manuscript.

Additional literature to discuss:

Bardeen et al. (2008) remains the clearest description of the agglomeration of MSP primary particles in the mesosphere and transport to the stratosphere. This study shows that MSP are formed at a relatively constant rate in the mesosphere, remain too small to sediment and are instead transported into the stratosphere by the downward

motion of the polar vortex. This means that it is misleading to state (page 3 line 95) that more MSP are produced from sporadic meteor events than from the constant IDP flux (which dominates the ablated material by mass). In fact both sporadic and constant fluxes feed into the same neutral metal layers which then form MSP. There is therefore a seasonal input of MSP to the polar upper stratosphere, which is then transported to lower latitude.

Brooke et al. (2017) improved on this work by including interactions of MSP particles with atmospheric sulfate. This study focussed on the difficult task of reproducing measurements of meteoric metals in ice cores, as referenced in the current manuscript. Brooke et al. (2017) concluded that additional input of meteoric material to the high latitude troposphere was needed, since only a crude treatment of transport in sedimenting large PSC aerosol was able to approach the values measured in the ice cores. The present study, particularly the tropospheric results, represents a valuable data set for future modelling studies to compare to.

Brooke et al. (2017) also tracked the likely size of MSP agglomerates through the lower atmosphere to surface deposition (figure S5). They showed, in agreement with Bardeen et al. (2008) at higher altitude, that the concentration of MSP particles above 70 nm radius is rather low. This suggests that the particles detected in the present study, with a lower limit of 200 nm diameter, are too large to be MSPs. The size and concentration of fragmented meteor particles is at present unconstrained, however recent publications have suggested that interplanetary dust particles smaller than several hundred nm are rather robust (Mannel et al., 2019), so it is likely that meteoric fragments are large enough to be detected here.

Dhomse et al. (2013) showed that the residence time of meteoric material transported through the atmosphere as MSP is several years. This is counter to the author's conclusion that "one would therefore expect to find a higher abundance of meteoric particles in the lower stratosphere at high latitudes during late winter and early spring". In addition, the theory that meteoric material leads to nucleation, growth and sedimentation

of PSC particles suggests that late winter polar stratosphere may be depleted in meteoric material. MSP are likely distributed relatively evenly throughout the stratosphere, with perhaps slightly less presence at lower latitudes (Kremser et al., 2016). However, taking meteoric fragments to have sizes greater than several hundred nm as described above, they would sediment rather rapidly to the lower stratosphere and thus likely also be distributed rather independently of season or latitude. I find the author's conclusion that the meteoric material is evenly distributed to be consistent with current theory of both MSPs and fragments.

The main text of the manuscript currently presents the mass spectra of the detected particles as remarkably reproducible, with the exception that the mass 56 peak is missing in the CAFÉ-Africa campaign. However looking at the spectra presented in the supplementary material, there is significant variability between clusters identified as meteoric. The ratio of Mg to Fe, and also the presence or absence of other metals seems rather variable between several meteoric clusters. Specifically: mass 39-41 (39K+, also MgO+ and / or 40Ca+ as assigned by Cziczo et al. (2001)) and mass 27 (Al+). It would be interesting to know if this is an instrumentation issue. Carrillo-Sánchez et al. (2016) discuss the differing elemental composition of sources of interplanetary dust. Variability in the composition of the detected aerosol may also be evidence that the detected particles are variable fragments, rather than MSP, since the latter are agglomerates of many nanoscale particles and should therefore have reproducible composition.

Previous works by some of the current authors, using steady state concentration approximations, have produced some of the highest estimates of the meteoric flux to the Earth, on the order of hundreds of tons per day (Weigel et al., 2014;Curtius et al., 2005). This, in comparison to modelling of atmospheric processes comparing the ablated amount of <50 tons per day (Carrillo-Sánchez et al., 2020), suggests that aircraft in the stratosphere are able to observe a portion of the unablated input of meteoric material to the Earth's atmosphere. It would be interesting to know whether the observations presented in this work support this conclusion. If so, then based on this and

earlier comments I think the authors should review their conclusion that their detected particles could be either MSP or fragments, or both (P27, line 649). Since fragmentation is at present rather poorly constrained, it is difficult to conclusively say that the particles detected here are fragments, but it also seems unlikely that they are MSPs. If these are fragments then the dataset represents a rare constraint on the flux of this type of meteoric material.

Other comments:

The manuscript presents results using several aircraft and a large number of instruments, measured during a variety of field campaigns. Whilst the terms used are clearly defined, I feel that a reader who was not familiar with these campaigns would benefit from the inclusion of a list of abbreviations.

The manuscript states (p28. Line 665) that "all meteoric particles contained H2SO4, but no other anions like nitrate or organic material." and "This suggests that these particles act similar as pure H2SO4 droplets in the UT with respect to cirrus formation and also in the polar stratosphere with respect to PSC formation." This is unclear. Since nitric acid is only taken up under equilibrium conditions at rather low temperatures in the polar vortex (Clegg et al., 1998), one would not expect to see nitrate signal from these particles with the possible exception of the ND-MAX data, in addition they would likely undergo significant change before the formation of PSC. For upper tropospheric cloud this may be an important observation since concentrated H2SO4 tends to be extremely hygroscopic, meaning that these particles might make extremely effective CCN. On the other hand concentrated H2SO4 is rather viscous, which may limit its ability to take up water (Price et al., 2015). It is unclear to my what the authors mean by this statement, so I suspect it needs additional clarification.

minor and typographical changes:

Figure 3: Top left panel says m, should say km. Since the location of the tropopause is later taken to be a set value for each campaign, could this be indicated with a horizontal

bar on the relevant panels?

Page 14 line 330 should read "boundary between troposphere and stratosphere"

P16 line 378 whilst "theta-latitude" is a relatively standard term, I find its use here to be somewhat abrupt. This terminology should be standardised throughout the manuscript.

P21 line 511 "between" should read "above"

P22 line 540 & Fig 8. Description of mixing lines is unclear. Perhaps "lines which are not horizontal or vertical" or "data points with intermediate concentrations of both tracers"?

P25 Line 585 change to "particles containing"

Supplement:

Page S2 first paragraph. I initially understood this to be describing the method for how the cluster was formed, rather than characteristics of a cluster which resulted from the analysis. This would be clearer if relevant sections of main text were referenced, where each characteristic of the cluster are discussed.

Page S4 last paragraph, section S10 should say "latter criterion".

Are both panels in Figure S11 on the same horizontal axis?

References.

Bardeen, C. G., Toon, O. B., Jensen, E. J., Marsh, D. R., and Harvey, V. L.: Numerical simulations of the three-dimensional distribution of meteoric dust in the mesosphere and upper stratosphere, J. Geophys. Res.: Atmos., 113, D17202, 2008. Brooke, J. S. A., Feng, W., Carrillo-Sánchez, J. D., Mann, G. W., James, A. D., Bardeen, C. G., Marshall, L., Dhomse, S. S., and Plane, J. M. C.: Meteoric smoke deposition in the polar regions: A comparison of measurements with global atmospheric models, J. Geophys. Res.: Atmos., 122, 11,112-111,130, 10.1002/2017jd027143, 2017.

[Figure]

Carrillo-Sánchez, J. D., Nesvornĭ, D., Pokornĭ, P., Janches, D., and Plane, J. M. C.: Sources of cosmic dust in the Earth's atmosphere, Geophys. Res. Lett., 43, 11,979-911,986, 10.1002/2016GL071697, 2016. Carrillo-Sánchez, J. D., Gómez-Martín, J. C., Bones, D. L., Nesvornĭ, D., Pokornĭ, P., Benna, M., Flynn, G. J., and Plane, J. M. C.: Cosmic dust fluxes in the atmospheres of Earth, Mars, and Venus, Icarus, 335, 113395, https://doi.org/10.1016/j.icarus.2019.113395, 2020. Clegg, S. L., Brimblecombe, P., and Wexler, A. S.: Thermodynamic model of the system H+−NH4+−SO42-−NO3-−H2O at tropospheric temperatures, J. Phys. Chem. A, 102, 2137-2154, 10.1021/jp973042r, 1998. Curtius, J., Weigel, R., Vössing, H. J., Wernli, H., Werner, A., Volk, C. M., Konopka, P., Krebsbach, M., Schiller, C., Roiger, A., Schlager, H., Dreiling, V., and Borrmann, S.: Observations of meteoric material and implications for aerosol nucleation in the winter Arctic lower stratosphere derived from in situ particle measurements, Atmos. Chem. Phys., 5, 3053-3069, 10.5194/acp-5-3053-2005, 2005. Cziczo, D. J., Thomson, D. S., and Murphy, D. M.: Ablation, flux, and atmospheric implications of meteors inferred from stratospheric aerosol, Science, 291, 1772-1775, 10.1126/science.1057737, 2001. Dhomse, S. S., Saunders, R. W., Tian, W., Chipperfield, M. P., and Plane, J. M. C.: Plutonium-238 observations as a test of modeled transport and surface deposition of meteoric smoke particles, Geophys. Res. Lett., 40, 4454-4458, 2013. Kremser, S., Thomason, L. W., von Hobe, M., Hermann, M., Deshler, T., Timmreck, C., Toohey, M., Stenke, A., Schwarz, J. P., Weigel, R., Fueglistaler, S., Prata, F. J., Vernier, J.-P., Schlager, H., Barnes, J. E., Antuña-Marrero, J.-C., Fairlie, D., Palm, M., Mahieu, E., Notholt, J., Rex, M., Bingen, C., Vanhellemont, F., Bourassa, A., Plane, J. n. M. C., Klocke, D., Carn, S. A., Clarisse, L., Trickl, T., Neely, R., James, A. D., Rieger, L., Wilson, J. C., and Meland, B.: Stratospheric aerosol—Observations, processes, and impact on climate, Reviews of Geophysics, 54, 278-335, 10.1002/2015rg000511, 2016. Mannel, T., Bentley, M. S., Boakes, P. D., Jeszenszky, H., Ehrenfreund, P., Engrand, C., Koeberl, C., Levasseur-Regourd, A. C., Romstedt, J., Schmied, R., Torkar, K., and Weber, I.: Dust of comet 67P/Churyumov-Gerasimenko collected by Rosetta/MIDAS: classification and extension to the nanometer scale, A&A,

630, A26, 2019. Price, H. C., Mattsson, J., Zhang, Y., Bertram, A. K., Davies, J. F., Grayson, J. W., Martin, S. T., O'Sullivan, D., Reid, J. P., Rickards, A. M. J., and Murray, B. J.: Water diffusion in atmospherically relevant $\alpha$-pinene secondary organic material, Chem.l Sci., 6, 4876-4883, 10.1039/C5SC00685F, 2015. Weigel, R., Volk, C. M., Kandler, K., Hösen, E., Günther, G., Vogel, B., Grooß, J. U., Khaykin, S., Belyaev, G. V., and Borrmann, S.: Enhancements of the refractory submicron aerosol fraction in the Arctic polar vortex: feature or exception?, Atmos. Chem. Phys., 14, 12319-12342, 10.5194/acp-14-12319-2014, 2014.

---

## Referee Comment (RC2) · Anonymous Referee #2 · 21 Sep 2020

Review of manuscript by Johannes Schneider et al. for Atmos. Chem. Phys.
"Aircraft-based observation of meteoric material in lower stratospheric
aerosol particles between 15 and 68N"

This manuscript presents an analysis of high-altitude aircraft
measurements of stratospheric aerosol particle composition from
two different laser-ablation aerosol mass spectrometer instruments.

The study brings together aerosol composition measurements from
5 different field campaigns between 2014 and 2018, from three
different research aircraft (the German G550 HALO, the European M55
Geophysica and the US NASA DC-8).

The analysis focuses on assessing the prevalence of the iron and magnesium
particle spectra within the mid-latitude stratospheric aerosol layer,
the composition signature indicating the presence of meteoric material
within these particles.  This topic is of particular interest given the
similar PALMS laser-ablation aerosol composition measurements on the WB-57
high-altitude aircraft showed in 1998 this meteoric signature was highly
abundant among aerosol particles in the mid-latitude stratosphere.
Refractory particle counter measurements in the Arctic stratosphere
show elevated concentrations within the polar vortex, increasing with
altitude into the stratosphere, thereby also strongly indicating them
likely being meteoric particles transported down from the upper atmosphere.

The authors are to be congratulated for bringing together this breadth of
measurements from the different campaigns, which then enables to assess
the meridional extent of the fraction of particles with the meteoric
composition-signature, and the vertical profile of the abundance of these
particles within the lowermost stratosphere.  Measurements are shown across
the range of latitudes and altitudes sampled in 5 different aircraft
missions that sampled in the upper troposphere and lower stratosphere (UTLS).

There has been a renewed debate in the stratospheric aerosol community
about the presence of meteoric material since the PALMS measurements
in 1998 revealed ~50% of particles in the mid-latitude lowermost stratosphere
contained signatures of meteoric material.  The refractory particle measurements
in the Arctic stratosphere from 2002/3 (Curtius et al., 2005) have been
confirmed with similar enhancement in refractory particles within the polar
vortex also found in Arctic campaigns in 2009/10 and 2010/11
(see Weigel et al., 2014), consistent with the particles being transported
each winter within subsiding air masses in the polar vortex, bringing a
seasonal source of meteoric particles down into the stratosphere each winter.

The observations will also be of considerable interest within the interactive
stratospheric aerosol modelling community, with the majority of current
models tending only to simulate the homogeneously nucleated particle
population, most not including the particle formation pathway of heterogeneous
sulfuric particle formation on meteoric particles.

I should disclose that I reviewed an earlier version of this manuscript
submitted to JGR in Jan/Feb 2020, a large number of corrections and revisions
required at that time, myself and the other two reviewers each finding
independently substantial changes were required before publication.  The
authors replied to all of the reviewer comments, and a greatly improved
revised manuscript was submitted, but although I recommended publication after
minor revisions, I can only assume the consensus among the reviewers and
editor was that the manuscript was still not quite at the standard for
publication, as I was subsequently notified that the paper had been rejected.

I can see that the paper has been substantially further improved since that
time, in particular with there now being a very welcome additional
Supplementary Material section,  which presents additional background information

to enable the interpetations of the data given in the main article to be further scrutinised and better understood.

The article is certainly suitable for ACP and is generally well-written, and I recommend publication after a number of specific minor revisions are made, which I have listed below.

However, there is one major concern that the authors need to explain, and caveat the "percentage meteoric" values presented in the Abstract sufficiently such that readers understand the real values may be substantially less than this, because of undersampling of the pure sulfuric particles.

The sentence beginning on line 483 states "...pure sulfuric acid particles are not ablated and ionized by a laser with wavelength of 266nm, because sulfuric acid has a very low absorption cross-section for wavelengths larger than about 190nm up to visible light", then citing the articles by Thomson et al. (1997), Burkholder et al. (2000) and Murphy et al. (2007).

The PALMS instrument used in Murphy et al. (1998) and Murphy et al. (2007) papers uses a 193nm laser, whereas, as is explained in the article, both the ALABAMA and ERICA instruments use 266nm lasers.

The Thomson et al. (1997) article assesses and discusses the aerosol absorption from 157nm, 193nm and 248nm lasers, and reading that paper, I see indeed the issue the authors are referring to --- that there is a large difference in absorption behaviour between pure sulfuric particles and those with organics and other "small amounts of contamination".

The authors state in the paragraph on lines 481-487 that "the fractions of particles containing meteoric material will be overestimated if pure sulfuric acid aerosol particles existed in the air".

But it is far from clear how the reader should then interpret the results.

Are the authors arguing that there are actually very few particles in the lower stratosphere that have levels of impurity low enough for them not to be ablated by the 266nm laser?

If so then they need to replace that wording "if pure sulfuric aerosol particles existed" and state it in those terms.

As the text stands, this potential undersampling issue remains a potential problem, potentially rendering the proportions stated in the Abstract to potentially be substantial overestimates.

The central question is to what extent particles are "pure enough" to suffer from the effect discussed in the Thomson et al. (1997) paper. Perhaps the authors are explaining that, in reality, all particles in the stratosphere contain a sufficient level of impurity that they believe that there is not a significant undersampling problem.

I do not know the answer to that question.

Figure 3b in the Murphy et al. (2007) shows the mass spectra of the "pure sulfuric particles" (as named in the Murphy et al., 2014 article) measured by the PALMS instrument (with the 193nm laser). Those spectra without the Fe and Mg signature do have other impurities, and it is not clear to me whether or not these would be detected by the ALABAMA and ERICA laser-ablation mass spectrometer instruments.

Provided the authors can provide information to assure the reader that this is the case, or can provide sufficient caveat to explain the severity of the potential undersampling they are acknowledging may be an issue, then the article can be published accordingly.

But the article is currently far from clear about this.

A related point, is that the authors really need to give some information about the composition of the particles ablated by the laser that do not have the Fe and Mg composition signature.

Do the ALABAMA and ERICA measure, similarly to Murphy et al. (2007), that the particles without the Fe and Mg signature group into spectra that are rich in carbonaceous material (i.e. Figure 3c in Murphy et al., 2007) and those that don't (those in Figure 3b)?

Murphy et al. (2007) show that the vast majority of the particles within the meteoric signature are the organic-rich sulfuric particles.

In which case the undersampling problem is not so important, because only 10-20% of the particles are pure enough to be missed by the ALABAMA and ERICA instruments.

And those percentages with the meteoric composition-signature can be considered to be highly reliable (albeit with a slight overestimate of perhaps 10% or so).

Basically the paragraph on lines 481-487 needs to be re-written to explain much more clearly how the reader should then interpret the fraction of meteoric particles being presented.

And I recommend adding to Figure 2 (or to the Supplementary Material) equivalent panels showing this mean spectra for the particles without the meteoric signature, with (if possible) further separated into those with and without the carbonaceous signature, consistently with the categorization from Murphy et al. (2014) as "pure sulfuric" and "organic-sulfuric" particles.

It sounds like the authors are explaining that only a small proportion of the non-meteoric-signature particles are pure sulfuric particles. And that is consistent with the results from the PALMS measurements. If so then I advise that be communicated within a revised version of this manuscript.

The paper needs to provide the reader with a clear interpretation of the findings, so that an approximate %-confidence measure can be considered in relation to the 20-40% and 60-80% meteoric particle fractions/proportions that are presented in the Abstract.

The current wording of that paragraph suggests a more significant under-sampling of the pure sulfuric particles, compared to the meteoric-signature particles, and this central issue needs to be presented much more clearly to enable the results to be properly interpreted.

Since the results are presented in the Abstract without caveat, I am assuming that the bias is of only a small magnitude (which is what the PALMS measurements suggest). If that is the case then  a sentence should be added both to the conclusions and the Abstract to provide clarity on the reason why the reader can be confident that is the case.

Although the article is mostly very well written, by contrast the Abstract seemed much less well-written and requires improvement. In almost all cases however, the revisions are minor wording improvements, but are important to better communicate the study's findings.

The rest of the article is well-written, although the Supplementary Material I found also needed quite a substantial number of minor revisions. The list of minor specific revisions are mostly then for the Abstract and Supplementary Material, and aside from the major revision explained above, the majority of the manuscript is in excellent shape already (perhaps reflecting its improvement after the previous set of reviews in the other journal).

Overall, provided the authors can address this one major issue, better
communicating the magnitude of the uncertainty via a corresponding sentence
added to the Abstract and Conclusions, then I am happy to recommend publication
to ACP once the minor revisions listed below are addressed.

Minor specific revisions
* * *
1) Abstract, lines 20-22 -- This first sentence is a little clunky to read, and
the scientific aim of the analysis in the paper is better to be communicated
earlier in the sentence. Suggest to move "to assess the meridional extent of
particles containing meteoric material" to be immediately after "between 2014
and 2018", then replacing "sampling" with the word "in". Also, suggest to delete
"In this paper", beginning instead as "We analyse ..." and delete "conducted"

I.e. have the sentence be "We analyze aerosol particle composition measurements
from five research missions between 2014 and 2018 to assess the meridional extent
of particles containing meteoric material in the upper troposphere and lower
stratosphere (UTLS)".

2) Abstract, lines 22-24 -- stating "confirm the existence of" is not really
appropriate. I know what you mean, but it's more to assess whether the meteoric
signature is also present in the lower troposphere. "Confirming the existence
of" suggests there is some doubt about whether these particles exist at all,
which is not the case.

Suggest to delete "are used to" and replace "confirm the  existence of meteoric
material in" with "show that meteoric material is also present within". Also,
the Jungfraujoch site is not sampling lower tropospheric particles, but
mid-tropospheric particles, so insert "middle and" before "the lower
tropospheric particles", adding also the clarifying additional
words ", but within only a very small proportion of particles."

Also, the wording of the first half of this sentence needs to be improved.
Firstly, the word "datasets" is too general a term, better to say "measurements"
and delete "Additional" -- and the phrase "a ground based study" should communicate
the location, such as "a mountain-top site" or better still "the Jungfraujoch
mountain-top site". Also it makes the sentence easier to read to hyphenate "low
altitude" to "low-altitude", with the "from" prior to that word also able to be
deleted for better wording.

So I mean that I am suggesting that the sentence be re-worded to something like:

"Measurements from the Jungraujoch mountain-top site and a low-altitude aircraft
mission show that meteoric material is also present within the lower and mid
tropospheric aerosol, but within only a very small proportion of particles."

3) Abstract, lines 24-25 -- Again suggest slight improvement to the wording here
to better link to the previous sentences and make it clear these are the main
observational datasets in both the UTLS field campaigns, and from the Jungfraujoch
and lower-altitude aircraft flights.

This can be achieved by changing the start of the sentence from "Single particle
laser ablation..." instead to "For both the UTLS campaigns and the
lower/mid-troposphere observations, the measurements were with single particle
 laser ablation...". Also suggest to change "techniques" to "mass spectrometers"
to be more precise, and "were used to measure" to "which enabled to measure".
Please also replace "size range" with "diameter range" or add "diameter" at the
end of the sentence, so it's clear those values are diameter values.

4) Abstract, line 27 -- Delete the words "particles" (after "147,338") and
"measured" (before "in the stratosphere"), better not to state that again, it's
implicit from earlier in the sentence and easier to read without these words.

5) Abstract, lines 27-30 -- Insert "total" after "Of these", delete "and rare iron oxide compounds", (the mass spectra are detecting the ions, and the same could be said of magnesium oxide, but doesn't need to be), also replacing "together with sulfuric acid" with "together with sulfuric ions".

I strongly suggest also to merge the subsequent sentence into this sentence, shortening the 2nd sentence so be a 2nd half of this sentence, i.e. replace ". This particle type was found almost exclusively in the stratosphere (48,610 particles) and is" with ", the vast majority (48,610) in the stratosphere,", also delete the "stratospheric" before "sulfuric acid" at the end of the sentence, also deleting the last word "particles". So I mean I'm suggested to re-word to:

"Of these total particles, 50,688 were characterized by high abundances of magnesium and iron, together with sulfuric ions, the vast majority (46,610) in the stratosphere, and are interpreted as meteoric material immersed or dissolved within sulfuric acid."

6) Abstract, lines 30-32 -- suggest to replace "particle type" with either "meteoric-sulfuric type" or "meteoric particle type" (or similar). Suggest to again join up the subsequent sentence, and shorten, also providing specific values for the two tropospheric locations -- i.e. replace ". However, small fractional abundances were observed below 3000m a.s.l. in the ..." with something like ", with 0.2-1 \% abundance at Jungfraujoch, and smaller abundances  (0.0x-0.0y \%) from the lower altitude Canadian Arctic aircraft measurements.".

7) Abstract, line 32 -- this sentence is strange -- it is not a new result to confirm that the removal pathway is by sedimentation and/or mixing into the troposphere.  The fact that there is a steep gradient across the tropopause confirms that the particles originate from the stratosphere or above, but that is not the way this is reported. It's kind-of obvious that a tracer with a source in the stratosphere (or above) would have a gradient across the tropopause, and that it would be removed by mixing into the troposphere. The question is really how important sedimentation is, in additional to simply air mass exchange from the stratosphere into the troposphere -- but that's not really addressed directly here.

The size distribution of the meteoric-signature particle is however an indirect measure of how important sedimentation is, because if the signature were found only in the smallest particles (~200nm) then sedimentation might not be that important, but here the findings from Murphy et al. (2014) are confirmed, that the meteoric-signature is found mostly in sulfuric particles at around 400-500nm, with much fewer in the 200-400nm size range.  That does suggest that the sedimentation is important in addition to mixing of air into the troposphere.

The size distribution of the meteoric signature is not currently mentioned in the Abstract, and this sentence is where this could be stated. I suggest the authors replace this sentence with "The size distribution of the meteoric-sulfuric particles measured in the UTLS campaigns is consistent with that measured by the PALMS measurements, with only 5-10% fractions in the smallest particles detected (200-300nm diameter), but with substantial (> 40%) abundance-fractions for particles from 300-350nm up to 900nm in diameter, suggesting sedimentation is the primary loss mechanism." Or similar sentence to this.

8) Abstract, line 36 -- replace "present in higher" with "present at much higher".

9) Abstract, lines 38-40 -- I'm not sure this sentence is necessarily the case. In the Introduction (lines 81-84), the authors discuss how meteoric fragments may sediment directly into the stratosphere. In contrast, the sentene here suggests the particles are transported down into the mesosphere only at high latitudes. That predominantly-transport-driven seasonal source of meteoric material

is the case for meteoric smoke particles (which tend to only be a few tens of nm), but if there is also a significant source of meteoric fragment particles (in addition to the smoke particles), then there may well be a source at other latitudes too. Indeed the finding on line 36 of the Abstract, that similar abundance-fraction is seen across all latitudes and seasons measured suggests the fragments are a substantial proportion of the meteoric material in the stratosphere.

Suggest to move "This finding suggests that" to be the start of the final sentence, and have this penultimate sentence explaining this winter polar vortex mechanism is the case for meteoric smoke particles. With then the sentence after explaining that the findings here suggest that there is another source of particles, in addition to the meteoric smoke.

I mean change the start of the the sentence beginning on line 38 from "This finding suggests that the meteoric material is transported..." to instead say "Meteoric smoke particles are transported...", change "is efficiently distributed towards" with "is subsequently transported towards..." and I think the authors must mean "below 440K potential temperature" not "above 440K potential temperature", because that transport tends mostly to occur in the lower part of the polar vortex.

10) Abstract lines 40-41 -- As per comment 9), I'm suggesting to begin this final sentence "By contrast, the findings from the UTLS measurements show meteoric material is found in stratospheric aerosol particles at all latitudes and seasons, which suggests meteoric fragments may nucleate a substantial proportion of the observed meteoric-sulfuric particles." Or something like this.

11) Introduction, line 62 -- replace "in the Earth's atmosphere" with "into the Earth's atmosphere"

12) Introduction, lines 68 and 69 -- Although MSP is almost always used with the third letters' corresponding water in the plural (Particles), it makes it much easier to read to communicate the plural including the lower-case s -- as MSPs. This is similar to way polar stratospheric clouds are referred to as PSCs. So replace the instances of "MSP" on lines 68 and 69 instead with "MSPs". Also on lines 95 and 97.

13) Introduction, line 82 -- Similarly you likely have "MF" here as an abbreviation for the plural term "Meteoric Fragments" but again, it's better to say "MFs", in the same way as MSPs and PSCs. Please change "MF" to "MFs" here and on line 84. Also on line 97.

14) Introduction, line 83 -- I'm not sure why you are questioning whether meteoric fragments form here. The preceding sentence begins "As has recently been shown...", so either that sentence needs to be changed to "have suggested" or else this sentence needs to be re-worded. However, the existence of meteoric fragments has been clear since rocket-borne measurements in the early 1960s (e.g. Hemenway and Soberman, 1962), with the fragments terminology having been introduced in the 1950s (e.g. Jacchia, 1955). Suggest to re-word the start of the preceding sentence to "As was hypothesised in the 1950s (e.g. Jacchia, 1955) and shown in measurements from the 1960s (e.g. Hemenway and Soberman, 1962), recently also further established by Subasinghe et al. (2016)...". Maybe it's just to change "were formed" to "are formed" and add "at sufficient particle concentrations" afterwards.

15) Introduction, lines 90-94 -- Again, although the term IDP is being used here as the plural term, it's easier to read this making the plural clear as "IDPs". Please change "IDP" to "IDPs" in all instances here, except on line 95 when the term is used in the singular.

16) Introduction, line 101 -- Change "Later, aircraft based" to "More recently," or "Much more recently,"

17) Introduction, line 116 -- Delete "summer" from the "Tropics/sub-tropics" because this seasonal variation is not relevant here.

18) Measurements and Methods, line 124 -- replace "includes" with "analyses", insert "lower" before "stratospheric" and provide a more descriptive word than "data", also avoiding using bland terms such as "obtained" (since they don't communicate these being measurements from the field).  Suggest also to replace "data obtained during" with "aerosol composition measurements taken" and replace "research" with "field". Also insert "additional composition measurement" after "with two" and insert "the lower troposphere" before "altitudes below 3600m a.s.l.", putting that last text in brackets -- i.e. "the lower troposphere (altitudes below 3600m a.s.l.".

19) Section 2.1.1, line 133 -- suggest to insert ", the full dataset from" before "which are included".

20) Section 2.1.3, line 147 -- hypenate "aircraft chasing" to "aircraft-chasing".

21) Section 2.1.4, line 155 -- replace "data which were obtained during" with "the measurement data from the" and replace "flights reaching" with "flights that reached".

22) Section 2.1.5, line 159 -- insert "middle and" before "lower troposphere", since Jungfraujoch is (in my opinion) sampling above the lower troposphere. Also change "we used two data sets" to "we also analyse two additional aerosol composition measurement datasets" and replace "low" with "lower".

23) Section 2.1.5, line 160 -- replace "during NETCARE" with "during the NETCARE field campaign".

24) Section 2.1.5, line 164 -- Improve the start of this sentence, changing "During the..." instead to "The other lower altitude dataset is from the mountain-top Jungfraujoch site during the...".

25) Section 2.2.1, line 175 -- Replace "has been described" with "is described".

26) Section 2.2.1, line 180 -- Replace "Having passed the aerodynamic lens" with "Having passed through an aerodynamic lens", insert "then" before "accelerated" and change "the vacuum chamber" to "a vaccum chamber".

27) Section 2.2.1, line 186 -- Suggest to delete "to the ALABAMA".

28) Section 2.2.1, line 187 -- Suggest to insert "to this paper" after "supplement".

29) Section 2.2.1, line 188 -- Suggest to replace "we include here a subset of" with "we analyse only the measurements from the"

30) Section 2.2.1, line 189 -- Insert "(i.e. where ......)" after "reached the stratosphere" to clarify the criterion that was used for this.

31) Section 2.2.2, line 200 -- replace "briefly reviewed" with "also described".

32) Section 2.2.2, line 200 -- the acronym "CPI" should be spelt out here as "constant pressure inlet" since it is its first use.  Note that cloud particle imager also has the same three-letter-acronym.

33) Section 2.2.2, line 209 -- move "during the StratoClim measurements" to the end of the sentence, as this is more of a clarifying term, i.e. make the sentence instead say "... on particles was about 40\% at diameters around 500nm during the StratoClim measurements".

34) Caption to Table 1 (line 216) -- change "Overview on the UTLS data sets" to "Overview of the 5 different aerosol composition measurement datasets".

35) Table 1 -- Given the issue with these measurements all having a lower frequency (higher wavelength) laser, add a row giving the wavelength used here.  Even though these (I think) are all the same at 266nm, it's important to state these here so the reader can easily scan that Table to find that information.

36) Section 2.2.3, line 221 -- insert "5 UTLS" before "campaigns were analyzed".
   Also, since this is a European journal, please change all instances of
   "analyzed" instead to "analysed".

38) Section 2.2.3, line 233-234 -- It is really great that the analysis has done this analysis to understand the variations with these different metrics, and the rationale for doing so should be stated.  So please change the start of this sentence from "Histograms of..." to "To enable to understand the different origin of the meteoric-signature particles, meteorological re-analysis data was combined with the measurements, with histograms of...", deleting "were" before "calculated" and replace "cluster as function of" with "cluster, as a function of".

39) Section 2.2.4, line 238 -- Suggest to improve the start of this 1st para of this section, replacing "The relation of " with "The steep vertical gradients in", and add "across the tropopause, means that correlating with measurements or re-analysis of these species" before "can be used". Then also replacing "potential tropospheric influence" with "previous tropospheric influence".

40) Section 2.2.4, lines 239-240 -- the use of the word "tracer" is potentially confusing (e.g. modellers use the word tracer as abbreviation for "trace species"). I suggest with the re-wording in point 39), this sentence can actually be deleted.

41) Section 2.2.4, line 241 -- replace "These measurements are briefly..." with "These additional measurement datasets are briefly..."

42) Section 2.2.4, lines 245, 246 and 247 -- hyphenate these 3 instances of "forward facing" instead to "forward-facing".

43) Section 2.2.4, line 252 -- insert ", whose detection method is"
before "based on"

44) Section 2.2.4, line 253 -- replace "of SHARC" with "of the SHARC hygrometer".

45) Section 2.2.4, line 254 -- replace "Monitor" with "monitor"

46) Section 2.2.4, line 256 -- replace "whatever" with "whichever".

47) Section 2.2.4, line 256 -- replace "with an" with "which has an".

48) Section 2.2.4, line 265 -- delete "range up to the extreme conditions"
and change "at a height of 20km" to "up to a height of ~20km".

49) Section 2.3, line 272 -- insert "for stratifying the data (e.g. the histograms in section 2.2.3)" before "were derived" and replace "using" with "from the".

50) Section 2.3, line 275 -- replace "first lapse rate tropopause" with "lowest altitude negative lapse rate" or some other more precise term.

51) Section 3.1 -- line 279 -- Suggest to replace "Distinct particle type" with "Meteoric-signature particle type" to be more scientifically descriptive.

52) Figure 2 -- as per the main issue I am asking the authors to reply to, there is a question as to the composition of the particles whose spectra do not show any Fe and Mg peaks.  The article needs to show the equivalent mean spectra for the non-meteoric-signature particles (ideally separated also into those with carbonaceous and those without carbonaceous, as in Murphy et al., 2007). This should be shown either in additional panels of this Figure 2 or as an additional Figure in the Supplementary Material.

53) Section 3.1 -- line 291 -- Replace "Further cations include" with
"Additional minor cation peaks include"

54) Section 3.1 -- line 292 -- suggest to replace "minor signals" with
"trace signals"

55) Section 3.1, lines 294-295 -- insert "the" before "two aircraft missions",
insert "with the ERICA" after "missions" and delete "namely".

56) Section 3.1, line 295 -- the word "spectra" is plural but here the term
is referring to the mean of the spectra, which is singular, so the word
"spectrum" should be used instead of "spectra" in this instance.
Also delete "obtained" and insert "the" before "18668 measurements".

57) Section 3.1, line 296 -- replace "during the StratoClim campaign"
with simply "during StratoClim", and since the word "spectrum" is used,
then the word "look" should be replaced with "looks". The word "compared"
can also be deleted on this line and "the mean mass spectra obtained"
replaced with the word "that", also inserting "the" before "3310" and
replacing "made during the CAFE-Africa 2018 campaign" with simply
"during CAFE-Africa 2018". Those changes make the text much easier to read.

58) Section 3.1, line 302 -- use the abbreviations Fe and Mg for iron
and magnesium on this line and replace "binned" with the more scientific
term "stratified".

59) Section 3.1, lines 303-304 -- this sentence beginning "For each bin"
can be deleted -- the information there is obvious and just makes this
paragraph more difficult to read.

60) In addition to deleting that sentence on lines 303-304, the text after
that can be tacked onto the end of the first sentence in that paragraph as
", with bin sizes of ...."

61) Section 3.1, lines 307-309 -- this sentence beginning "It has to be
emphasized" can (in my opinion) be deleted -- that is obvious, and the
text already gives the total number of particles in the previous sentence,
so the reader will have those numbers in their mind already.  I think it
makes this sentence much easier to read if you simply delete this sentence
(the reader will understand that to be the case already).

62) Section 3.1, lines 311-312 --  again, use Fe and Mg abbreviations
here rather than the words iron and magnesium.   But more importantly
this sentence needs to be much clearer how much of an effect this
value is.  Since Murphy et al. (2007) PALMS measurements, which have
the lower wavelength (higher frequency, i.e. higher power) laser,
and therefore do sample the pure sulfuric particles, show that these
pure sulphuric particles represent only about 10% of the particles.
So I think you can say here that the under-sampling of the pure
sulphuric particles will not have a significant effect on the fractions
given -- and that the reader should be confident in these numbers.

63) Section 3.1, line 324 -- replace "these particles" with
"the meteoric-signature particles".

64) Section 3.1, line 330 -- you've written "tropopause" but you mean
"troposphere" here -- please correct that. Also insert "often" before
"defined via the..." and suggest to add "(known as the thermal
tropopause or cold-point tropopause)" after "lapse rate" and make
that be the end of that sentence. Then have that start the next sentence
". The potential vorticity" instead of continuing as
", but potential vorticity..."

65) Section 3.1, line 331 -- I'd suggest "better indicator" rather
than "good indicator" -- I think the dynamical tropopause would be
the preferred metric if both were available. And please also put
the words "dynamical tropopause" in inverted-commas in the manuscript,
also changing the preceding words from "indicator for the" instead
to ", representing a " so that the sentence is introducing this term.

66) Section 3.1, lines 337-338 -- reword "during StratoClim 2017
which took place over the AMA" instead to "during the StratoClim
2017 flights sampling above the AMA".

67) Section 3.2, line 351 -- replace "inserted" with "added to the Figure".

68) Section 3.2, lines 352-353 -- please state what time-interval for
the individual measurements (across which this median and quartiles
are taken).

69) Section 3.2, line 353 -- insert "range for the dynamical tropopause
is shown, from" before "a 2 PVU and a 5 PVU surface" -- deleting
the "a" and replaceing "and" with "to" -- i.e. changing that to be
"range for the dynamical tropopause from 2 PVU to 5 PVU".

70) Section 3.2, Figure 4 -- in the legends delete the text
"with quartiles" -- that can go in the caption to the Figure.
Having it in the legend obscures some of the yellow parts of
the data, and it would be better then to have the smaller box
and seeing more of the data.

71) Section 3.5, line 482 -- Insert the word "accurately" after
estimate, and replace "an absolute" with "the absolute".

72) Section 3.5, lines 486-487 -- as per my major comments at the
start of this review, this sentence needs to be changed -- it's not
appropriate to write "if pure sulfuric acid aerosol particles
existed in the air".  It's clear from the PALMS measurements that
only about 10-20 \% (at most 40%) of particles in the campaigns
analysed in Murphy et al. (2007) were of this pure sulphuric
particle nature. And you should add a sentence here stating these
percentages so that the reader can know that at least two-thirds
of the particles (probably more) are being sampled by the 266nm
laser used by the ALABAMA and ERICA instruments. That way the
reader can know that it is only a relatively small-to-moderate
fraction of the particles that might be being missed in these
measurements.

73) Section 3.5, line 490 -- insert "mid and" before "lower".

74) Section 3.6, line 590 -- give the range of percentage occurence
that you mean by "was very low".  At Jungfraujoch this is 0.2 to 1\%,
whereas in the Canadian Arctic the value is much lower.  Better to
give the corresponding values here.

75) Section 4, lines 621-622 -- Suggest to replace "From previous"
with "Consistent with " and insert "aerosol composition measurement"
after "previous stratospheric" and replace "it was concluded"
with "it is concluded".

76) Section 4, line 626 -- with this being the Discussion and
conclusions section, better here not to use the MSP acronym, instead
give the words, replacing "MSP particles" with "meteoric smoke particles".

77) Section 4, line 637 -- replace "so this altitude" with
"with this altitude", then also replacing "refers to"
instead to "referring only to".

78) Supplementary Material -- Introduction, 2nd line (1st sentence)
Insert "shown in the main article, to enable its" before "interpretation",
and insert "to be scrutinised transparently" after "interpretation.

79) Supplementary Material -- Introduction, 2nd line (2nd sentence)
Replace "It includes the clustering parameters...." with
"Firstly, the clustering methodology is explained in more
detail, with the clustering parameters...", also replacing
"evaluation and the uncertainty" with "evaluation, and an
associated uncertainty", replacing "estimation" with
"estimated".

80) Supplementary Material -- Introduction, 3rd line (3rd sentence)
The text "Individual clusters of particles are displayed (S2)" needs
to be changed because the Figures S1 to S5 show mean spectra
not individual spectra. Also, the vertical profiles of the
meteoric fraction are also shown in those Figures.
So, replace that text instead with
"Secondly (S2), the mean mass spectra and vertical profile of
the meteoric-particle abundance fractions for each of the
5 UTLS campaigns are shown in Figures S1 to S5."

81) Supplementary Material -- Introduction, 4th line (4th sentence)
Insert "each of" before "the individual" and replace
"mission" with "missions", adding "(Figure S6) after "in S3".

82) Supplementary Material -- Introduction, 4th line (5th sentence)
Insert "(Figure S7)" before "shows the O3-H2O"...

83) Supplementary Material -- Introduction, 10th line
(penultimate sentence in this section) Replace "present in"
with "presented in".

84) Supplementary Material -- Introduction, 11th line
(final sentence in this section) Replace "SectionS10"
with "Section S10", delete "the" after "explains",
insert "changing" before "the threshold" and replace
"was derived" with "affects the stratospheric proportions
presented".

85) Supplementary -- Clustering algorithm, lines 6 & 7
Replace "chose as distance metric" with "used for the
distance metric" and replace "spectra): a Pearson..."
with "spectra), with a perfect Pearson...", then
putting "r=1" in brackets, and changing "means that"
to "meaning that".

86) Supplementary -- Clustering algorithm, final sentence
Replace "stopping" with "convergence".

87) Supplementary -- Variation of clustering parameters, line 6
Replace "particles containing" with "particles identified
to contain" and correct "meteorological material" with
"meteoric material".

88) Supplementary -- Section S2 -- insert "each of" before
"the five".

89) Supplementary -- Section S3 -- insert "each of" before
"the five" and replace "mission. All data were merged to"
instead to "missions, these data merged to".

90) Supplementary -- Section S8 -- line 8 of the text.

Insert "when the refractive index for stratospheric
aerosol is used".

91) Supplementary -- Section S8 -- line 9 of the text.
Delete "size channel with" and replace "corresponds to"
with "increases to", deleting "for stratospheric
aerosol particles".

92) Supplementary -- Section S9 -- line 3 of the text.
Insert "(NCEP meteorological re-analysis, Saha et al., 2010)"
after "0.5 degree data set".

93) Supplementary -- Section S9 -- line 5 of the text.
Insert ", with" before "27 trajectories".

94) Supplementary -- Section S9 -- line 6 of the text.
Replace "binned in altitude and latitude bins and"
with "stratified into altitude and latitude bins and"

95) Supplementary -- Section S10 -- line 1 of the text.
Replace "recorded" with "measured".

96) Supplementary -- Section S10 -- line 2 of the text.
Replace "We used" with "To test the sensitivity of
the calculations, we used"

97) Supplementary -- Section S10 -- line 5 of the text.
Replace "as a threshold" with "as the threshold".

References
* * *
Burkholder, J. B., Mills, M. and McKeen, S. (2000),
"Upper limit for the UV absorption cross sections of H2SO4",
Geophys. Res. Lett., vol. 27, no. 16, pp. 2,493-2,496, 2000.

Curtius, J., Weigel, R., Voessing, H.-J. et al. (2003)
"Observations of meteoric material and implications for aerosol
nucleation in the winter Arctic lower stratosphere derived from in
situ particle measurements",
Atmos. Chem. Phys., vol. 5, pp. 3,053-3,069, 2003.

Deshler, T., Hervig, M. E., Hofmann, D. J. et al. (2003),
"Thirty years of in situ stratospheric aerosol size distribution
measurements from Laramie, Wyoming (41oN), using balloon-borne instruments"
J. Geophys. Res., vol. 108, no. D5, 4167, doi:10.1029/2002JD002514, 2003.

Jacchia, L. G.
"The physical theory of meteors. VIII. Fragmentation as
cause of the faint-meteor anomaly",
Astronomical Journal, vol. 121, pp. 521-527.

Hemenway, C. L. and Soberman, R. K. (1962)
"Studies of micrometeorites from recoverable sounding rocket",
Astronomical Journal, vol. 67, no. 5,

Murphy, D. M., Thomson, D. S. and Mahoney, M. J. (1998):
"In Situ Measurements of organics, meteoritic material, mercury, and
other elements in aerosols at 5 to 19 kilometers",
Science, vol. 282, pp. 1,664-1,669, 1998.

Murphy, D. M., Cziczo, D. J. Hudson, P. K. and Thomson, D. S. (2007):
"Carbonaceous material in aerosol particles in the lower stratosphere
and tropopause region",

J. Geophys. Res., vol. 112, D04203, doi:10.1029/2006JD007297, 2007.

Murphy, D. M., Froyd, K. D., Schwarz, J. P. and Wilson, J. C. (2014):
"Observations of the chemical composition of stratospheric aerosol particles",
Q. J. Roy. Meteorol. Soc., vol. 140, pp. 1,269-1,278, 2014.

Saha, S., Moorthi, S., Pan, H.-L. et al. (2010):
"The NCEP climate forecast system re-analysis"
Bull. Amer. Meteorol. Soc., vol. 91, no. 8, pp. 1015-1058.

Thomson et al., D. S., Middlebrook, A. M. and Murphy, D. M. (1997)
"Thresholds for laser-induced ion formation from aerosols in a vacuum
using ultraviolet and vacuum-ultraviolet laser wavelengths"
Aer. Sci. Techonol., vol. 26 no. 6, pp. 544-559, 1997.

Weigel, R., Volk, C. M., Kandler, K. et al., (2014)
"Enhancements of the refractory submicron aerosol fraction in the
Arctic polar vortex: feature or exception?"
Atmos. Chem. Phys., vol. 14, pp. 12,319-12,342, 2014.

---

## Short Comment (SC1) · 26 Sep 2020

This interesting paper presents a nice analysis of the solid material collected in the stratosphere and assumed to originate from meteorites. Nevertheless, I have some troubles with the content of the paper:

1) How can the authors be sure that the analyzed particles have a non-terrestrial origin? Solid particles originated from Earth during various process, or even produced inside the atmosphere, can have the same chemical elements. Some dynamical processes can lift these particles in the stratosphere.

2) Confusion is made by the author all along the paper between meteoritic disintegration and interplanetary dust (IDP). Some of the particles of such size (1-300 $\mu$m) could be interplanetary dust grains mainly coming from comets, not particles coming from meteorite disintegration. Also, some of these (large) particles can survive the atmospheric entry, as those found in the Antarctica ices. The author must consider the works done by the teams that collect such particles. Also, interplanetary dust and grains coming from meteorites do not have the same composition as cometary grains. The authors must consider the works done on the composition of cometary grains and interplanetary grains, and not only on the composition of meteorites.

3) Since the authors have collected a large number of such grains, they can calculate the total concentration and even the total mass-concentration, and they must verify that these values are consistent with the expected flux of solid material (coming from comets and meteorites) that entry the Earth atmosphere. The author must also consider the concentration of interplanetary dust at Earth level.

4) We have discussed the problem of the various origins of solid material in the stratosphere, and of the vertical transport of the particles, in the paper now published in "Atmosphere": J.‐B. Renard, G. Berthet, A.‐C. Levasseur‐Regourd, S. Beresnev, A. Miffre, P. Rairoux, D. Vignelles, Fabrice Jégou, Origins and Spatial Distribution of Non‐Pure Sulfate Particles (NSPs) in the Stratosphere Detected by the Balloon‐Borne Light Optical Aerosols Counter (LOAC), Atmosphere 2020, 11, 1031; doi:10.3390/atmos11101031. We discuss the IDP and meteoritic material confusion done in many papers; we present a summary of the properties of the IDP and of cometary material. The authors must be advised that we had submitted a few month ago in ACP a previous version of our paper, but it was rejected by an associated editor that is in the same laboratory as one author of this Schneider et al. paper. Obviously, we are sure this is just a coincidence. Nevertheless, we encourage the authors to consider our work and to clarify their analysis considering the various sources that can exist for the material they have identified.

---

## Author Comment (AC1) · 13 Nov 2020

Dear Dr. Renard,

Thank you for your comment to our manuscript. Please find the answers to your comments below in blue color.

Changes in the revised manuscript are printed in red.

This interesting paper presents a nice analysis of the solid material collected in the stratosphere and assumed to originate from meteorites. Nevertheless, I have some troubles with the content of the paper:

[Figure]

1) How can the authors be sure that the analyzed particles have a non-terrestrial origin? Solid particles originated from Earth during various process, or even produced inside the atmosphere, can have the same chemical elements. Some dynamical processes can lift these particles in the stratosphere.

In fact, we discussed this issue in our manuscript in detail (see Sections 3.2 and 3.3): We have three major arguments:

- The fraction of the observed Fe- and Mg containing particles increases with altitude (Fig. 3).

- The composition (cation mass spectra dominated by Mg and Fe, anion mass spectra by sulfate) excludes other (terrestrial or anthropogenic) sources (other sources show different composition, see Section 3.3).

- Upward transport of tropospheric air masses into the stratosphere occurs mainly in the tropics. However, our data show that in the tropics (here mainly the Asian Monsoon Anticyclone, AMA), the fraction of the Mg- and Fe-containing particle remains low in the lower stratosphere (Fig. 4). See end of Section 3.2: "The observation that the fraction of the iron and magnesium-dominated particle type increases only above the extratropical tropopause layer or mixing layer (Hoor et al., 2002; Hoor et al., 2004; Pan et al., 2004), i.e. 30 K above the tropopause (Fig. 4 b), indicates that the source for this particle type must be above the tropopause, because otherwise, the upwelling air masses in the AMA would contain this particle type also at lower potential temperatures".

2) Confusion is made by the author all along the paper between meteoritic disintegration and interplanetary dust (IDP). Some of the particles of such size (1-300 $\mu$m) could be interplanetary dust grains mainly coming from comets, not particles coming from meteorite disintegration. Also, some of these (large) particles can survive the atmospheric entry, as those found in the Antarctica ices. The author must consider

the works done by the teams that collect such particles. Also, interplanetary dust and grains coming from meteorites do not have the same composition as cometary grains. The authors must consider the works done on the composition of cometary grains and interplanetary grains, and not only on the composition of meteorites.

We rewrote the introduction with respect to the subject of IPD and meteoric sources, summarize the different possible contributions under the general term "meteoric material" and use this term from there on in the rest of the paper.

We emphasize that from our data we can not distinguish between the different sources of meteoric material, and clarified that again in the conclusions section of the revised version:

"Our observations of particles with signatures of meteoric material do not clearly indicate the formation history, i.e. whether the material originates from meteoric disintegration by ablation (MSP), fragmentation (MF) or from interplanetary dust particles (IDP), since the meteoric material is at least partially dissolved in sulfuric acid."

3) Since the authors have collected a large number of such grains, they can calculate the total concentration and even the total mass-concentration, and they must verify that these values are consistent with the expected flux of solid material (coming from comets and meteorites) that entry the Earth atmosphere. The author must also consider the concentration of interplanetary dust at Earth level.

Unfortunately, the single particle mass spectrometry technique is not able to provide a mass fraction of certain compounds in one single particle. Thus, we do not know how much mass of meteoric material is contained in the detected particles. For example, a particle may have a diameter of 300 nm and is mainly composed of sulfuric acid, but may contain the meteoric material that originates from one meteor smoke particle (MSP) of a few nm in diameter. It may also be that by collision and coagulation more than one MSP ended up in one stratospheric sulfuric acid-dominated particle. Thus, our method (together with the fact that our data represent only parts of the lower stratosphere) does not allow us to calculate a mass concentration of extraterrestrial material
in the atmosphere.

4) We have discussed the problem of the various origins of solid material in the strato-
sphere, and of the vertical transport of the particles, in the paper now published in "At-
mosphere": J.- B. Renard, G. Berthet, A- C. Levasseur-Regourd, S. Beresnev, A. Miffre,
P. Rairoux, D. Vignelles, F. Jégou, Origins and Spatial Distribution of Non-Pure Sul-
fate Particles (NSPs) in the Stratosphere Detected by the Balloon-Borne Light Optical
Aerosols Counter (LOAC), Atmosphere 2020, 11, 1031; doi:10.3390/atmos11101031.
We discuss the IDP and meteoritic material confusion done in many papers; we present
a summary of the properties of the IDP and of cometary material. The authors must be
advised that we had submitted a few month ago in ACP a previous version of our paper,
but it was rejected by an associated editor that is in the same laboratory as one author
of this Schneider et al. paper. Obviously, we are sure this is just a coincidence. Nev-
ertheless, we encourage the authors to consider our work and to clarify their analysis
considering the various sources that can exist for the material they have identified.

Thank you for pointing us to your recent paper in Atmosphere. We will refer to the
results you obtained from stratospheric balloon-borne measurements in the revised
version of our manuscript. One point we would like to mention is that the particles we
are describing (the Mg- and Fe- dominated particles we interpret as meteoric material
dissolved in sulfuric acid) represent only a subset of the "none-pure sulfate particles"
(NSP) that you describe in your publication. In fact, as also reviewer 2 pointed out (and
we discussed in section 3.5), our single particle mass spectrometer has a low detection
efficiency for pure sulfuric acid. That means, that almost all particles detected by laser
ablation mass spectrometry (using 266 nm ablation laser wavelength) represent NSP.
Thus, many of the particles we observed will have terrestrial origin. But, the focus
of our present paper is the stratospheric meridional distribution of particles containing
meteoric material. Further publications will analyze the nature and sources of the other
particles detected during the individual campaigns.

---

## Author Response (AR1)

**ACP-2020-660**
Aircraft-based observation of meteoric material in lower stratospheric
aerosol particles between 15 and 68°N
J. Schneider et al.

**Reply to Referee #1**

The reviewer comments are written in this font style and color.

Our answers are written in this font style and color.

Changes in the revised version of the manuscript are printed in red.

The manuscript submitted presents measurements of meteoric material, identified by its elemental composition, in atmospheric aerosol in the lower stratosphere and in some locations in the troposphere. Observations are presented from a variety of campaigns at a range of latitudes, altitudes and seasons.

Scientific significance:

Measurements of this type in the lower stratosphere are not entirely new, as acknowledged by the authors. However, the statistical analysis made possible by the size of this dataset leads to conclusions regarding trends in the atmospheric abundance of these aerosol which is a new and valuable contribution to the literature. In addition the observation of these particles in the troposphere provides evidence of the occurrence of transport processes which have been previously speculated. I feel that by neglecting aspects of the literature the authors have underestimated the value of their work, and hope to assist in my suggestions below.

Scientific quality:

The scientific approach seems sound and appropriate to the stated aims. In my comments below I suggest several further details which might be usefully discussed.

Presentation quality:

On the whole I find the presentation to be of a standard suitable for publication. I do have some suggestions to improve the readability and effectiveness of the figures. The written English is understandable to me as a native English speaker, though it does use some non-standard (German) sentence structure. I have suggested only typographical language changes.

General comments:

It is my opinion that addressing the following issues will improve the manuscript as presented. I believe these to be minor changes, but acknowledge that some may be more complex than they seem to me. I advise the editor to accept reasonable explanations of why some of my recommendations may not be practical. The most significant change I believe is required is to broaden and better support the scope of the study by including aspects of the literature on meteoric smoke and fragmentation which have been overlooked. This has implications at various points in the manuscript.

Additional literature to discuss:

Bardeen et al. (2008) remains the clearest description of the agglomeration of MSP primary particles in the mesosphere and transport to the stratosphere. This study shows that MSP are formed at a relatively constant rate in the mesosphere, remain too small to sediment and are instead transported into the stratosphere by the downward motion of the polar vortex. This means that it is misleading to state (page 3 line 95) that more MSP are produced from sporadic meteor events than from the constant IDP flux (which dominates the ablated material by mass). In fact both sporadic and constant fluxes feed into the same neutral metal layers which then form MSP. There is therefore a seasonal input of MSP to the polar upper stratosphere, which is then transported to lower latitude.

We agree with the reviewer that our statement on the sporadic events and constant IDP fluxes was incorrect.

We therefore revised this part of the introduction as follows, including also the results by Bardeen et al. (2008) and Brooke et al. (2017). In the first sentence of this part we introduce the term "meteoric material" which encompasses the contribution of IDP, sporadic events, and MSP. Throughout the rest of the manuscript we use only the term "meteoric material".

"The continuous import of submicrometer IDPs, the sporadic events of meteors' disintegration on atmospheric entry, and the meteoric fragments (with radii < 0.5 µm, Brooke et al., 2017) contribute to the atmosphere's load of meteoric material, which becomes incorporated and partially dissolved in acidic aerosols (e.g. of $HNO_3$ and/or $H_2SO_4$ at different dilutions with $H_2O$). Bardeen et al. (2008) investigated ablated meteoric material by means of coupled general circulation model and sectional microphysics model simulations. Due to a mesospheric meridional circulation, as Bardeen et al. (2008) revealed, the re-nucleated meteoric ablation material is transported towards the respective winter pole where it subsides within the polar vortex to stratospheric altitudes. According to the investigations of Dhomse et al. (2013), the nanoparticles released at upper mesospheric altitudes (corresponding to MSP, which are produced by ablation and recombination in the upper atmosphere) reside for about four years in the atmosphere until they are deposited on the surface. The same simulations (Dhomse et al., 2013) predicted the strongest deposition of meteoric ablation material at mid-latitudes with a substantially (~ 15 times) higher efficiency over Greenland than in Antarctica."

Brooke et al. (2017) improved on this work by including interactions of MSP particles with atmospheric sulfate. This study focussed on the difficult task of reproducing measurements of meteoric metals in ice cores, as referenced in the current manuscript.

Brooke et al. (2017) concluded that additional input of meteoric material to the high latitude troposphere was needed, since only a crude treatment of transport in sedimenting large PSC aerosol was able to approach the values measured in the ice cores. The present study, particularly the tropospheric results, represents a valuable data set for future modelling studies to compare to.

The changes in the introduction as given above do also take into account the results by Brooke et al. 2017.

Brooke et al. (2017) also tracked the likely size of MSP agglomerates through the lower atmosphere to surface deposition (figure S5). They showed, in agreement with Bardeen et al. (2008) at higher altitude, that the concentration of MSP particles above 70 nm radius is rather low. This suggests that the particles detected in the present study, with a lower limit of 200 nm diameter, are too large to be MSPs. The size and concentration of fragmented meteor particles is at present unconstrained, however recent publications have suggested that interplanetary dust particles smaller than several hundred nm are rather robust (Mannel et al., 2019), so it is likely that meteoric fragments are large enough to be detected here.

This is a misunderstanding. As we emphasized in our manuscript, the analysed particles consist of meteoric material dissolved in (or possibly coated by) sulfuric acid. All particles that show the meteoric signature (Mg and Fe) show a large sulfuric ($HSO_4^-$) anion signal. However, our method does not allow us to derive the mass fraction of the meteoric material in the particles. Thus, we do not know the original size of the initial MSP that is dissolved in such a $H_2SO_4$ particle of a few hundred nm in diameter.

Therefore, as already mentioned above, we prefer to use the term "meteoric material" for the detected particles by our method.

Dhomse et al. (2013) showed that the residence time of meteoric material transported through the atmosphere as MSP is several years. This is counter to the author's conclusion that "one would therefore expect to find a higher abundance of meteoric particles in the lower stratosphere at high latitudes during late winter and early spring".

But, as you mentioned above, the study by Bardeen et al. (2008) shows "…that MSP are formed at a relatively constant rate in the mesosphere, remain too small to sediment and are instead transported into the stratosphere by the downward motion of the polar vortex".

Thus, we should expect a higher abundance of meteoric material in the outflow of the polar vortex.

We clarified the paragraph in the conclusions section:

"Downward transport of meteoric smoke particles from the mesosphere into the lowermost stratosphere occurs efficiently in the polar vortex (Curtius et al., 2005; Weigel et al., 2014). These papers show that high altitude aircraft measurements demonstrate there is a higher proportion of refractory particles (60-70%) within the wintertime polar vortex, and one would therefore expect to find a higher abundance of meteoric particles in the lower stratosphere at high latitudes during late winter and early spring than at lower extra-tropical latitudes and than in other seasons. This expectation would largely agree with (1) the results by Dhomse et al. (2013), who predicted a more effective (by a factor of ~15) deposition of meteoric ablation material over Greenland than in Antarctica, and (2) the works of Bardeen et al. (2008) and Brooke et al. (2017), according to which the meteoric ablation material most effectively subsides to stratospheric altitudes within the polar winter vortex. This is not confirmed by our observations: Although two mid-latitude campaigns (ML-CIRRUS and ND-MAX/ECLIF-2) were conducted between January and April, we observe the same fraction of meteoric particles at the same ozone levels in the lower stratosphere (Fig. 3) during all campaigns, regardless of latitude and season. "

In addition, the theory that meteoric material leads to nucleation, growth and sedimentation of PSC particles suggests that late winter polar stratosphere may be depleted in meteoric material.

We disagree with this statement. PSCs form at altitudes between around 16 and 24 km, and their sedimentation leads to a re-distribution of, e.g., odd nitrogen to the lower stratosphere (12 – 14 km) where the PSC particles evaporate. For example, Hübler et al. (1990) and Fischer et al. (1997) found elevated $NO_y$ concentrations in the Arctic lower stratosphere at potential temperatures of about 350 K (around 12 km), which is well in the Arctic stratosphere. Therefore, PSC sedimentation would only lead to enhanced downward transport of meteoric material into the lower stratosphere, but not to a removal from the atmosphere by further downward transport into the troposphere.

MSP are likely distributed relatively evenly throughout the stratosphere, with perhaps slightly less presence at lower latitudes (Kremser et al., 2016). However, taking meteoric fragments to have sizes greater than several hundred nm as described above, they would sediment rather rapidly to the lower stratosphere and thus likely also be distributed rather independently of season or latitude. I find the author's conclusion that the meteoric material is evenly distributed to be consistent with current theory of both MSPs and fragments.

As we emphasized above, we can't conclude from our data what the origin of the meteoric material is. We therefore clarified in the conclusions section:

"Our observations of particles with signatures of meteoric material do not clearly indicate the formation history, i.e. whether the material originates from meteoric disintegration by ablation (MSP), fragmentation (MF) or from interplanetary dust particles (IDP), since the meteoric material is at least partially dissolved in sulfuric acid."

We removed the following sentence: "However, the high $H_2SO_4$ content of all detected meteoric particles and the uniform mass spectra suggest that MSPs dissolved in sulfuric acid are the most likely particle source."

The main text of the manuscript currently presents the mass spectra of the detected particles as remarkably reproducible, with the exception that the mass 56 peak is missing in the CAFE-Africa campaign. However looking at the spectra presented in the supplementary material, there is significant variability between clusters identified as meteoric.
The ratio of Mg to Fe, and also the presence or absence of other metals seems rather variable between several meteoric clusters. Specifically: mass 39-41 (39K+, also MgO+ and / or 40Ca+ as assigned by Cziczo et al. (2001)) and mass 27 (Al+). It would be interesting to know if this is an instrumentation issue. Carrillo-Sánchez et al. (2016) discuss the differing elemental composition of sources of interplanetary dust. Variability in the composition of the detected aerosol may also be evidence that the detected particles are variable fragments, rather than MSP, since the latter are agglomerates of many nanoscale particles and should therefore have reproducible composition. Previous works by some of the current authors, using steady state concentration approximations, have produced some of the highest estimates of the meteoric flux to the Earth, on the order of hundreds of tons per day (Weigel et al., 2014;Curtius et al., 2005). This, in comparison to modelling of atmospheric processes comparing the ablated amount of <50 tons per day (Carrillo-Sánchez et al., 2020), suggests that aircraft in the stratosphere are able to observe a portion of the unablated input of meteoric material to the Earth's atmosphere. It would be interesting to know whether the observations presented in this work support this conclusion. If so, then based on this and earlier comments I think the authors should review their conclusion that their detected particles could be either MSP or fragments, or both (P27, line 649). Since fragmentation is at present rather poorly constrained, it is difficult to conclusively say that the particles detected here are fragments, but it also seems unlikely that they are MSPs. If these are fragments then the dataset represents a rare constraint on the flux of this type of meteoric material.

We have had the same idea when analyzing the data and checked whether we could find a dependence of the ion ratios Mg/Fe, Al/Fe, K/Fe, and Na/Fe of latitude, altitude, or potential temperature. However, no significant trend effect was observed. Thus, we conclude that this is an instrumental issue. A random variation in the ion ratios due to the ablation ionization process means that the clustering algorithm will result in a certain number of clusters with different ion ratios, and the number of these resulting clusters depends on the number of prescribed clusters. Summarizing, it is not possible to distinguish MSP and fragments from our method, at least not at our current state of knowledge.

Other comments:

The manuscript presents results using several aircraft and a large number of instruments, measured during a variety of field campaigns. Whilst the terms used are clearly defined, I feel that a reader who was not familiar with these campaigns would benefit from the inclusion of a list of abbreviations.

To our opinion, it is sufficient that all acronyms are spelled out at first use. ACP requires that abbreviations "… need to be defined in the abstract and then again at the first instance in the rest of the text":
(https://www.atmospheric-chemistry-and-physics.net/submission.html#manuscriptcomposition).
A separate acronym list is not foreseen by ACP.

The manuscript states (p28. Line 665) that "all meteoric particles contained H2SO4, but no other anions like nitrate or organic material." and "This suggests that these particles act similar as pure H2SO4 droplets in the UT with respect to cirrus formation and also in the polar stratosphere with respect to PSC formation." This is unclear. Since nitric acid is only taken up under equilibrium conditions at rather low temperatures in the polar vortex (Clegg et al., 1998), one would not expect to see nitrate signal from these particles with the possible exception of the ND-MAX data, in addition they would likely undergo significant change before the formation of PSC. For upper tropospheric cloud this may be an important observation since concentrated H2SO4 tends to be extremely hygroscopic, meaning that these particles might make extremely effective CCN. On the other hand concentrated H2SO4 is rather viscous, which may limit its ability to take up water (Price et al., 2015). It is unclear to my what the authors mean by this statement, so I suspect it needs additional clarification.

For PCS formation, we are here referring to the works of Tritscher et al. (2019) and James et al. (2018) that were referenced in the introduction. These studies needed to include "foreign nuclei" into their simulations to reproduce PSC observations. Meteoric particles were suggested to be such "foreign nuclei". However, if the particles containing meteoric material "behave" like $H_2SO_4/H_2O$ droplets due to their small mass fraction of meteoric material (20 nm MSP dissolved in 200 nm $H_2SO_4/H_2O$), this information needs to be added to the model.

With respect to cirrus clouds, laboratory measurements (Saunders et al., 2010) showed that refractory particles consisting of $Fe_2O_3$ and $MgO$ nucleated ice under cirrus conditions. However, if particles of meteoric origin are not present as solid particles but as a dilute solution in $H_2SO_4/H_2O$ droplets, the freezing properties will likely change from heterogeneous to homogeneous freezing. However, we agree that dissolved meteoric material might also alter the viscosity of $H_2SO_4$ under low temperature conditions that has been described by Williams and Long (1995).

We therefore changed the statement to:

"Our data further show that all meteoric particles contained $H_2SO_4$, but no other anions like nitrate or organic material. Thus, from our simultaneous cation and anion measurements we can confirm previous assumptions that Mg and Fe are dissolved in $H_2SO_4$ (Murphy et al., 1998; Cziczo et al., 2001; Murphy et al., 2014). This suggests that these particles act similar as pure $H_2SO_4$ droplets in the UT with respect to cirrus formation, but it is conceivable that dissolved

meteoric material alters the viscosity of $H_2SO_4/H_2O$ droplets which was found to increase at low temperatures (Williams and Long, 1995). With respect to PSC formation in the polar stratosphere, the works by James et al. (2018) and Tritscher et al. (2019) showed that "foreign nuclei" are needed to be included in their simulations to reproduce PSC observations. The finding that meteoric material present as dilute solution in $H_2SO_4/H_2O$ droplets needs to be included in future simulations."

**minor and typographical changes:**

Figure 3: Top left panel says m, should say km.

Thanks for pointing out this mistake, it was corrected.

Since the location of the tropopause is later taken to be a set value for each campaign, could this be indicated with a horizontal bar on the relevant panels?

We don't think that this would be helpful. It would mean adding the 380 K line to the StratoClim 2017 plot. temperature graph, adding a shaded area 2-5 PVU for the 4 extratropical campaigns to the PV graph, and then finally a 150 ppb $O_3$ line in the lowest row to all graphs. We also see from Figure 4 that the thermal tropopause varies as a function of latitude and is therefore not a constant value for the individual campaigns.

Page 14 line 330 should read "boundary between troposphere and stratosphere"

Corrected

P16 line 378 whilst "theta-latitude" is a relatively standard term, I find its use here to be somewhat abrupt. This terminology should be standardised throughout the manuscript.

We introduced "theta-latitude space" at the beginning of section 3.2 and use "theta-latitude space" and "theta-equivalent latitude space" throughout the rest of the text.

P21 line 511 "between" should read "above"

Corrected

P22 line 540 & Fig 8. Description of mixing lines is unclear. Perhaps "lines which are not horizontal or vertical" or "data points with intermediate concentrations of both tracers"?

The definition of mixing lines as "lines connecting the respective mixing ratios of the initial unmixed reservoir air parcels" is taken from the referenced Hoor et al. (2002) publication and thus we would prefer to keep it.

P25 Line 585 change to "particles containing"

Corrected

Supplement:

Page S2 first paragraph. I initially understood this to be describing the method for how the cluster was formed, rather than characteristics of a cluster which resulted from the analysis. This would be clearer if relevant sections of main text were referenced, where each characteristic of the cluster are discussed.

The method after which the clusters were selected to contain "meteoric material" are given at the end of section S1:

"Criteria for selecting a certain cluster as "containing meteoric material" were 1) high cation signals of $Fe^+$ and $Mg^+$ (additionally allowing $Na^+$, $K^+$, $Al^+$), 2) anion signal at $HSO_4^-$ or cation signals at $S^+$, $SO_4^+$, $H_3SO_4^+$, 3) vertical profile showing increasing fractional abundance with increasing altitude, potential temperature, or potential vorticity."

We added a sentence explaining this to Section S2.

Page S4 last paragraph, section S10 should say "latter criterion".

Corrected

Are both panels in Figure S11 on the same horizontal axis?

Yes, the horizontal axis is the same (0.1 - 2 µm), but the vertical axis is different.

References.

Bardeen, C. G., Toon, O. B., Jensen, E. J., Marsh, D. R., and Harvey, V. L.: Numerical simulations of the three-dimensional distribution of meteoric dust in the mesosphere and upper stratosphere, J. Geophys. Res.-Atmos., 113, 10.1029/2007jd009515, 2008.

Brooke, J. S. A., Feng, W., Carrillo-Sánchez, J. D., Mann, G. W., James, A. D., Bardeen, C. G., Marshall, L., Dhomse, S. S., and Plane, J. M. C.: Meteoric Smoke Deposition in the Polar Regions: A Comparison of Measurements With Global Atmospheric Models, Journal of Geophysical Research: Atmospheres, 122, 11,112-111,130, 10.1002/2017JD027143, 2017.

Curtius, J., Weigel, R., Vossing, H. J., Wernli, H., Werner, A., Volk, C. M., Konopka, P., Krebsbach, M., Schiller, C., Roiger, A., Schlager, H., Dreiling, V., and Borrmann, S.: Observations of meteoric material and implications for aerosol nucleation in the winter Arctic lower stratosphere derived from in situ particle measurements, Atmospheric Chemistry and Physics, 5, 3053-3069, 2005.

Cziczo, D. J., Thomson, D. S., and Murphy, D. M.: Ablation, flux, and atmospheric implications of meteors inferred from stratospheric aerosol, Science, 291, 1772-1775, 2001.

Dhomse, S. S., Saunders, R. W., Tian, W., Chipperfield, M. P., and Plane, J. M. C.: Plutonium-238 observations as a test of modeled transport and surface deposition of meteoric smoke particles, Geophys. Res. Lett., 40, 4454-4458, 10.1002/grl.50840, 2013.

Fischer, H., Waibel, A. E., Welling, M., Wienhold, F. G., Zenker, T., Crutzen, P. J., Arnold, F., Burger, V., Schneider, J., Bregman, A., Lelieveld, J., and Siegmund, P. C.: Observations of high concentrations of total reactive nitrogen (NOy) and nitric acid (HNO3) in the lower Arctic stratosphere during the stratosphere-troposphere experiment by aircraft measurements (STREAM) II campaign in February 1995, J. Geophys. Res.-Atmos., 102, 23559-23571, 1997.

Hoor, P., Fischer, H., Lange, L., Lelieveld, J., and Brunner, D.: Seasonal variations of a mixing layer in the lowermost stratosphere as identified by the CO-O-3 correlation from in situ measurements, J. Geophys. Res.-Atmos., 107, 10.1029/2000jd000289, 2002.

Hübler, G., Fahey, D. W., Kelly, K. K., Montzka, D. D., Carroll, M. A., Tuck, A. F., Heidt, L. E., Pollock, W. H., Gregory, G. L., and Vedder, J. F.: Redistribution of reactive odd nitrogen in the lower Arctic stratosphere, Geophys. Res. Lett., 17, 453-456, 10.1029/GL017i004p00453, 1990.

James, A. D., Brooke, J. S. A., Mangan, T. P., Whale, T. F., Plane, J. M. C., and Murray, B. J.: Nucleation of nitric acid hydrates in polar stratospheric clouds by meteoric material, Atmos. Chem. Phys., 18, 4519-4531, 10.5194/acp-18-4519-2018, 2018.

Murphy, D. M., Thomson, D. S., and Mahoney, T. M. J.: In situ measurements of organics, meteoritic material, mercury, and other elements in aerosols at 5 to 19 kilometers, Science, 282, 1664-1669, 1998.

Murphy, D. M., Froyd, K. D., Schwarz, J. P., and Wilson, J. C.: Observations of the chemical composition of stratospheric aerosol particles, Quarterly Journal of the Royal Meteorological Society, 140, 1269-1278, 10.1002/qj.2213, 2014.

Saunders, R. W., Möhler, O., Schnaiter, M., Benz, S., Wagner, R., Saathoff, H., Connolly, P. J., Burgess, R., Murray, B. J., Gallagher, M., Wills, R., and Plane, J. M. C.: An aerosol chamber investigation of the heterogeneous ice nucleating potential of refractory nanoparticles, Atmos. Chem. Phys., 10, 1227-1247, 10.5194/acp-10-1227-2010, 2010.

Tritscher, I., Grooß, J. U., Spang, R., Pitts, M. C., Poole, L. R., Müller, R., and Riese, M.: Lagrangian simulation of ice particles and resulting dehydration in the polar winter stratosphere, Atmos. Chem. Phys., 19, 543-563, 10.5194/acp-19-543-2019, 2019.

Weigel, R., Volk, C. M., Kandler, K., Hösen, E., Günther, G., Vogel, B., Grooß, J. U., Khaykin, S., Belyaev, G. V., and Borrmann, S.: Enhancements of the refractory submicron aerosol fraction in the Arctic polar vortex: feature or exception?, Atmos. Chem. Phys., 14, 12319–12342, 10.5194/acpd-14-12319-2014, 2014.

Williams, L. R., and Long, F. S.: Viscosity of supercooled sulfuric acid solution, Journal of Physical Chemistry, 99, 3748-3751, 10.1021/j100011a050, 1995.

**Reply to Reviewer #2**

The reviewer comments are written in this font style and color.

Our answers are written in this font style and color.

Changes to the revised version of the manuscript are printed in red.

Review of manuscript by Johannes Schneider et al. for Atmos. Chem. Phys.
"Aircraft-based observation of meteoric material in lower stratospheric
aerosol particles between 15 and 68N"

This manuscript presents an analysis of high-altitude aircraft
measurements of stratospheric aerosol particle composition from two
different laser-ablation aerosol mass spectrometer instruments.

The study brings together aerosol composition measurements from 5
different field campaigns between 2014 and 2018, from three different
research aircraft (the German G550 HALO, the European M55 Geophysica and
the US NASA DC-8).

The analysis focuses on assessing the prevalence of the iron and magnesium
particle spectra within the mid-latitude stratospheric aerosol layer, the
composition signature indicating the presence of meteoric material within
these particles. This topic is of particular interest given the similar
PALMS laser-ablation aerosol composition measurements on the WB-57 high-
altitude aircraft showed in 1998 this meteoric signature was highly
abundant among aerosol particles in the mid-latitude stratosphere.
Refractory particle counter measurements in the Arctic stratosphere show
elevated concentrations within the polar vortex, increasing with altitude
into the stratosphere, thereby also strongly indicating them likely being
meteoric particles transported down from the upper atmosphere.

The authors are to be congratulated for bringing together this breadth of
measurements from the different campaigns, which then enables to assess
the meridional extent of the fraction of particles with the meteoric
composition-signature, and the vertical profile of the abundance of these
particles within the lowermost stratosphere. Measurements are shown
across the range of latitudes and altitudes sampled in 5 different
aircraft missions that sampled in the upper troposphere and lower
stratosphere (UTLS).

There has been a renewed debate in the stratospheric aerosol community
about the presence of meteoric material since the PALMS measurements in
1998 revealed ~50% of particles in the mid-latitude lowermost stratosphere
contained signatures of meteoric material. The refractory particle
measurements in the Arctic stratosphere from 2002/3 (Curtius et al., 2005)
have been confirmed with similar enhancement in refractory particles
within the polar vortex also found in Arctic campaigns in 2009/10 and
2010/11 (see Weigel et al., 2014), consistent with the particles being
transported each winter within subsiding air masses in the polar vortex,

bringing a seasonal source of meteoric particles down into the stratosphere each winter.

The observations will also be of considerable interest within the interactive stratospheric aerosol modelling community, with the majority of current models tending only to simulate the homogeneously nucleated particle population, most not including the particle formation pathway of heterogeneous sulfuric particle formation on meteoric particles.

I should disclose that I reviewed an earlier version of this manuscript submitted to JGR in Jan/Feb 2020, a large number of corrections and revisions required at that time, myself and the other two reviewers each finding independently substantial changes were required before publication. The authors replied to all of the reviewer comments, and a greatly improved revised manuscript was submitted, but although I recommended publication after minor revisions, I can only assume the consensus among the reviewers and editor was that the manuscript was still not quite at the standard for publication, as I was subsequently notified that the paper had been rejected.

I can see that the paper has been substantially further improved since that time, in particular with there now being a very welcome additional Supplementary Material section, which presents additional background information to enable the interpetations of the data given in the main article to be further scrutinised and better understood.

The article is certainly suitable for ACP and is generally well-written, and I recommend publication after a number of specific minor revisions are made, which I have listed below.

We thank the reviewer for this generally positive rating of our manuscript.

However, there is one major concern that the authors need to explain, and caveat the "percentage meteoric" values presented in the Abstract sufficiently such that readers understand the real values may be substantially less than this, because of undersampling of the pure sulfuric particles.

We have addressed this major concern in the revised version and explain this in the answers given below.

The sentence beginning on line 483 states "...pure sulfuric acid particles are not ablated and ionized by a laser with wavelength of 266nm, because sulfuric acid has a very low absorption cross-section for wavelengths larger than about 190nm up to visible light", then citing the articles by Thomson et al. (1997), Burkholder et al. (2000) and Murphy et al. (2007). The PALMS instrument used in Murphy et al. (1998) and Murphy et al. (2007) papers uses a 193nm laser, whereas, as is explained in the article, both the ALABAMA and ERICA instruments use 266nm lasers.
The Thomson et al. (1997) article assesses and discusses the aerosol absorption from 157nm, 193nm and 248nm lasers, and reading that paper, I see indeed the issue the authors are referring to --- that there is a large difference in absorption behaviour between pure sulfuric particles and those with organics and other "small amounts of contamination".
The authors state in the paragraph on lines 481-487 that "the fractions of particles containing meteoric material will be overestimated if pure sulfuric acid aerosol particles existed in the air".
But it is far from clear how the reader should then interpret the results.

Are the authors arguing that there are actually very few particles in the
lower stratosphere that have levels of impurity low enough for them not
to be ablated by the 266nm laser?
If so then they need to replace that wording "if pure sulfuric aerosol
particles existed" and state it in those terms.

We rewrote Section 3.5 as detailed below.

As the text stands, this potential undersampling issue remains a potential
problem, potentially rendering the proportions stated in the Abstract to
potentially be substantial overestimates.

The central question is to what extent particles are "pure enough" to
suffer from the effect discussed in the Thomson et al. (1997) paper.
Perhaps the authors are explaining that, in reality, all particles in the
stratosphere contain a sufficient level of impurity that they believe
that there is not a significant undersampling problem.
I do not know the answer to that question.
Figure 3b in the Murphy et al. (2007) shows the mass spectra of the "pure
sulfuric particles" (as named in the Murphy et al., 2014 article) measured
by the PALMS instrument (with the 193nm laser). Those spectra without the
Fe and Mg signature do have other impurities, and it is not clear to me
whether or not these would be detected by the ALABAMA and ERICA laser-
ablation mass spectrometer instruments.
Provided the authors can provide information to assure the reader that
this is the case, or can provide sufficient caveat to explain the severity
of the potential undersampling they are acknowledging may be an issue,
then the article can be published accordingly.
But the article is currently far from clear about this.

The PALMS instrument, using an ablation laser with 193 nm, is able to detect "pure sulfuric
particles", as presented in Murphy et al. (2007; 2014). This particle type reaches a fraction
between 10 and 20% at mid-latitudes, for ozone mixing ratios up to 1800 ppb (see Fig. 3a in
Murphy et al., 2014). In Murphy et al. (2007; Fig 4), the reported fractions range between 10
and 30%, up to 8 km above the tropopause, with ozone reaching up to 1200 ppb. These
campaigns were conducted out of Houston, Texas, and San Jose, Costa Rica.

We may therefore assume that in our data set (ozone never exceeding 1200 ppb), the
underestimation of the total analyzed particle number due to the presence of pure sulfuric
particles is about 20 (± 10) %.
This translates into an overestimation of the fraction of meteoric particles by the same range.

We state this underestimation in the abstract of the revised version and rewrote Section 3.5 (see
below). However, we prefer not to correct for this underestimation, as the uncertainty would be
very high. The variation of the pure sulfuric fraction with altitude, potential temperature,
latitude, and season is not known well enough to transfer the data by Murphy et al. (2007; 2014)
to our data set.

A related point, is that the authors really need to give some information
about the composition of the particles ablated by the laser that do not
have the Fe and Mg composition signature.
Do the ALABAMA and ERICA measure, similarly to Murphy et al. (2007), that
the particles without the Fe and Mg signature group into spectra that are
rich in carbonaceous material (i.e. Figure 3c in Murphy et al., 2007) and
those that don't (those in Figure 3b)?

Our results give a more complicated picture of stratospheric aerosol. The particles do not only fall into the three categories that were described by Murphy et al. 2007 and 2014, but contain also ammonia, black carbon, secondary organics, and nitrate. We present an averaged mass spectrum of all stratospheric particles that do not contain the meteoric signatures (Mg and Fe) in the new panels (b) and (d) of Figure 2.

However, a further analysis of the different types of stratospheric particles and their composition and sources is outside the scope of this paper and will be presented in upcoming publications.

```
Murphy et al. (2007) show that the vast majority of the particles within
the meteoric signature are the organic-rich sulfuric particles.
In which case the undersampling problem is not so important, because only
10-20% of the particles are pure enough to be missed by the ALABAMA and
ERICA instruments.
And those percentages with the meteoric composition-signature can be
considered to be highly reliable (albeit with a slight overestimate of
perhaps 10% or so).
Basically the paragraph on lines 481-487 needs to be re-written to explain
much more clearly how the reader should then interpret the fraction of
meteoric particles being presented.
```

We rewrote the beginning of Section 3.5 as follows:

"It is difficult to estimate accurately the absolute number concentration of particles containing meteoric material from the measured particle fraction with our laser ablation mass spectrometers. The main reason is that pure sulfuric acid particles are not ablated and ionized by a laser with a wavelength of 266 nm, because sulfuric acid has a very low absorption cross section for wavelengths larger than about 190 nm up to visible light (Thomson et al., 1997; Burkholder et al., 2000; Murphy, 2007). Thus, the fraction of particles containing meteoric material will be overestimated due to the presence of pure sulfuric acid aerosol particles in the stratosphere.

The PALMS instrument, using an ablation laser with 193 nm (Murphy et al., 1998; Cziczo et al., 2001; Murphy et al., 2007; Murphy et al., 2014) is able to detect pure sulfuric acid particles. The results presented in Murphy et al. (2007) show that the number fraction of the sulfuric particle type ranges between 10 and 30% up to 8 km above the tropopause and at ozone mixing ratios up to 1200 ppb. These data were obtained at tropical (Costa Rica) and mid latitudes (Texas). In Murphy et al. (2014), the presented number fraction of sulfuric particles measured at mid latitudes ranges between 10 and 20 %, for ozone mixing ratios up to 1800 ppb. We may therefore assume that in our data, where ozone never exceeded 1200 ppb, the underestimation of the total analyzed particle number due to the presence of pure sulfuric acid particles is about 20 %, ranging between 10 and 30 %. This translates into an overestimation of the meteoric particle fraction by the same percentage. However, the variation of the pure sulfuric acid fraction with altitude, potential temperature, latitude, and season is not known well enough to apply a correction to our data set. Thus, it must be noted that the meteoric particle number fraction as well as the following estimation of the absolute number concentration of particles containing meteoric material may be overestimated by 10 – 30 %."

```
And I recommend adding to Figure 2 (or to the Supplementary Material)
equivalent panels showing this mean spectra for the particles without the
meteoric signature, with (if possible) further separated into those with
and without the carbonaceous signature, consistently with the
categorization from Murphy et al. (2014) as "pure sulfuric" and "organic-
sulfuric" particles.
```

We added to Figure 2 the mean spectra of all stratospheric particles without the meteoric signature. A further separation into different particles types (such as with and without carbonaceous signature) will be subject of upcoming publications that focus on the composition of UTLS aerosol under the influence of Asian and African monsoon.

```
It sounds like the authors are explaining that only a small proportion of
the non-meteoric-signature particles are pure sulfuric particles. And
that is consistent with the results from the PALMS measurements. If so
then I advise that be communicated within a revised version of this
manuscript.
```

We rewrote section 3.5 (see above)

```
The paper needs to provide the reader with a clear interpretation of the
findings, so that an approximate %-confidence measure can be considered
in relation to the 20-40% and 60-80% meteoric particle
fractions/proportions that are presented in the Abstract.

The current wording of that paragraph suggests a more significant under-
sampling of the pure sulfuric particles, compared to the meteoric-
signature particles, and this central issue needs to be presented much
more clearly to enable the results to be properly interpreted.
```

We rewrote Section 3.5 (see above), removed Figure S8 and modified Figure 7 accordingly. We mention an overestimation of the meteoric particle fraction by about 10 – 30% in the abstract and in Section 3.5.

```
Since the results are presented in the Abstract without caveat, I am
assuming that the bias is of only a small magnitude (which is what the
PALMS measurements suggest). If that is the case then a sentence should
be added both to the conclusions and the Abstract to provide clarity on
the reason why the reader can be confident that is the case.
```

We added "It must be noted that the relative abundance of such meteoric particles may be overestimated by about 10 to 30% due to the presence of pure sulfuric acid particles in the stratosphere which are not detected by the instruments used here."
Furthermore, we added "observed fraction" or "of the observed particles" to all places where percentage values are mentioned in the abstract.

```
Although the article is mostly very well written, by contrast the Abstract
seemed much less well-written and requires improvement. In almost all
cases however, the revisions are minor wording improvements, but are
important to better communicate the study's findings.

The rest of the article is well-written, although the Supplementary
Material I found also needed quite a substantial number of minor
revisions. The list of minor specific revisions are mostly then for the
Abstract and Supplementary Material, and aside from the major revision
explained above, the majority of the manuscript is in excellent shape
already (perhaps reflecting its improvement after the previous set of
reviews in the other journal).
```

Overall, provided the authors can address this one major issue, better communicating the magnitude of the uncertainty via a corresponding sentence added to the Abstract and Conclusions, then I am happy to recommend publication to ACP once the minor revisions listed below are addressed.

Minor specific revisions
* * *
1) Abstract, lines 20-22 -- This first sentence is a little clunky to read, and the scientific aim of the analysis in the paper is better to be communicated earlier in the sentence. Suggest to move "to assess the meridional extent of particles containing meteoric material" to be immediately after "between 2014 and 2018", then replacing "sampling" with the word "in". Also, suggest to delete "In this paper", beginning instead as "We analyse ..." and delete "conducted" I.e. have the sentence be "We analyze aerosol particle composition measurements from five research missions between 2014 and 2018 to assess the meridional extent of particles containing meteoric material in the upper troposphere and lower stratosphere (UTLS)".

Changed as suggested

2) Abstract, lines 22-24 -- stating "confirm the existence of" is not really appropriate. I know what you mean, but it's more to assess whether the meteoric signature is also present in the lower troposphere. "Confirming the existence of" suggests there is some doubt about whether these particles exist at all, which is not the case. Suggest to delete "are used to" and replace "confirm the existence of meteoric material in" with "show that meteoric material is also present within". Also, the Jungfraujoch site is not sampling lower tropospheric particles, but mid-tropospheric particles, so insert "middle and" before "the lower tropospheric particles", adding also the clarifying additional words ", but within only a very small proportion of particles." Also, the wording of the first half of this sentence needs to be improved. Firstly, the word "datasets" is too general a term, better to say "measurements" and delete "Additional" -- and the phrase "a ground based study" should communicate the location, such as "a mountain-top site" or better still "the Jungfraujoch mountain-top site". Also it makes the sentence easier to read to hyphenate "low altitude" to "low-altitude", with the "from" prior to that word also able to be deleted for better wording. So I mean that I am suggesting that the sentence be re-worded to something like: "Measurements from the Jungraujoch mountain-top site and a low-altitude aircraft mission show that meteoric material is also present within the lower and mid tropospheric aerosol, but within only a very small proportion of particles."

Changed as suggested

3) Abstract, lines 24-25 -- Again suggest slight improvement to the wording here to better link to the previous sentences and make it clear these are the main observational datasets in both the UTLS field campaigns, and from the Jungfraujoch and lower-altitude aircraft flights. This can be achieved by changing the start of the sentence from "Single particle laser ablation..." instead to "For both the UTLS campaigns and the lower/mid-troposphere observations, the measurements were with single particle laser ablation...". Also suggest to change "techniques" to "mass spectrometers" to be more precise, and "were used to measure" to "which enabled to measure". Please also replace "size range" with "diameter

range" or add "diameter" at the end of the sentence, so it's clear those values are diameter values.

Changed as suggested

4) Abstract, line 27 -- Delete the words "particles" (after "147,338") and "measured" (before "in the stratosphere"), better not to state that again, it's implicit from earlier in the sentence and easier to read without these words.

Changed as suggested

5) Abstract, lines 27-30 -- Insert "total" after "Of these", delete "and rare iron oxide compounds", (the mass spectra are detecting the ions, and the same could be said of magnesium oxide, but doesn't need to be), also replacing "together with sulfuric acid" with "together with sulfuric ions". I strongly suggest also to merge the subsequent sentence into this sentence, shortening the 2nd sentence so be a 2nd half of this sentence, i.e. replace ". This particle type was found almost exclusively in the stratosphere (48,610 particles) and is" with ", the vast majority (48,610) in the stratosphere,", also delete the "stratospheric" before "sulfuric acid" at the end of the sentence, also deleting the last word "particles". So I mean I'm suggested to re-word to: "Of these total particles, 50,688 were characterized by high abundances of magnesium and iron, together with sulfuric ions, the vast majority (46,610) in the stratosphere, and are interpreted as meteoric material immersed or dissolved within sulfuric acid."

Changed as suggested

6) Abstract, lines 30-32 -- suggest to replace "particle type" with either "meteoric-sulfuric type" or "meteoric particle type" (or similar). Suggest to again join up the subsequent sentence, and shorten, also providing specific values for the two tropospheric locations -- i.e. replace ". However, small fractional abundances were observed below 3000m a.s.l. in the ..." with something like ", with 0.2-1 \% abundance at Jungfraujoch, and smaller abundances (0.0x-0.0y \%) from the lower altitude Canadian Arctic aircraft measurements.".

Changed to "Below the tropopause, the observed fraction of the meteoric particle type decreases sharply with 0.2 − 1 % abundance at Jungfraujoch, and even smaller abundances (0.025 − 0.05 %) observed during the lower altitude Canadian Arctic aircraft measurements."

7) Abstract, line 32 -- this sentence is strange -- it is not a new result to confirm that the removal pathway is by sedimentation and/or mixing into the troposphere. The fact that there is a steep gradient across the tropopause confirms that the particles originate from the stratosphere or above, but that is not the way this is reported. It's kind-of obvious that a tracer with a source in the stratosphere (or above) would have a gradient across the tropopause, and that it would be removed by mixing into the troposphere. The question is really how important sedimentation is, in additional to simply air mass exchange from the stratosphere into the troposphere -- but that's not really addressed directly here. The size distribution of the meteoric-signature particle is however an indirect measure of how important sedimentation is, because if the signature were found only in the smallest particles (~200nm) then

sedimentation might not be that important, but here the findings from Murphy et al. (2014) are confirmed, that the meteoric-signature is found mostly in sulfuric particles at around 400-500nm, with much fewer in the 200-400nm size range. That does suggest that the sedimentation is important in addition to mixing of air into the troposphere. The size distribution of the meteoric signature is not currently mentioned in the Abstract, and this sentence is where this could be stated. I suggest the authors replace this sentence with "The size distribution of the meteoric-sulfuric particles measured in the UTLS campaigns is consistent with that measured by the PALMS measurements, with only 5-10% fractions in the smallest particles detected (200-300nm diameter), but with substantial (> 40%) abundance-fractions for particles from 300-350nm up to 900nm in diameter, suggesting sedimentation is the primary loss mechanism." Or similar sentence to this.

Changed as suggested, but as we did not mention ALABAMA and ERICA up to now, we also would not like to mention PALMS, and instead wrote "with earlier aircraft-based mass spectrometric measurements,..."

8) Abstract, line 36 -- replace "present in higher" with "present at much higher".

Changed

9) Abstract, lines 38-40 -- I'm not sure this sentence is necessarily the case. In the Introduction (lines 81-84), the authors discuss how meteoric fragments may sediment directly into the stratosphere. In contrast, the sentene here suggests the particles are transported down into the mesosphere only at high latitudes. That predominantly-transport-driven seasonal source of meteoric material is the case for meteoric smoke particles (which tend to only be a few tens of nm), but if there is also a significant source of meteoric fragment particles (in addition to the smoke particles), then there may well be a source at other latitudes too. Indeed the finding on line 36 of the Abstract, that similar abundance-fraction is seen across all latitudes and seasons measured suggests the fragments are a substantial proportion of the meteoric material in the stratosphere. Suggest to move "This finding suggests that" to be the start of the final sentence, and have this penultimate sentence explaining this winter polar vortex mechanism is the case for meteoric smoke particles. With then the sentence after explaining that the findings here suggest that there is another source of particles, in addition to the meteoric smoke. I mean change the start of the the sentence beginning on line 38 from "This finding suggests that the meteoric material is transported..." to instead say "Meteoric smoke particles are transported...", change "is efficiently distributed towards" with "is subsequently transported towards..." and I think the authors must mean "below 440K potential temperature" not "above 440K potential temperature", because that transport tends mostly to occur in the lower part of the polar vortex.

We agree that meteoric fragments are as likely as MSP to explain our observations. But, as we can't distinguish between the two, we changed this part mostly as suggested, but with the last sentence: "By contrast, the findings from the UTLS measurements show that meteoric material is found in stratospheric aerosol particles at all latitudes and seasons, which suggests that either isentropic mixing is effective also above 440 K or that meteoric fragments may be the source of a substantial proportion of the observed meteoric material." The uniform occurrence of meteoric material above 440 K is a strong argument for an effective isentropic mixing also above 440 K.

10) Abstract lines 40-41 -- As per comment 9), I'm suggesting to begin this final sentence "By contrast, the findings from the UTLS measurements show meteoric material is found in stratospheric aerosol particles at all latitudes and seasons, which suggests meteoric fragments may nucleate a substantial proportion of the observed meteoric-sulfuric particles." Or something like this.

Changed to "By contrast, the findings from the UTLS measurements show that meteoric material is found in stratospheric aerosol particles at all latitudes and seasons, which suggests that meteoric fragments may be the source of a substantial proportion of the observed meteoric material."

11) Introduction, line 62 -- replace "in the Earth's atmosphere" with "into the Earth's atmosphere"

Done

12) Introduction, lines 68 and 69 -- Although MSP is almost always used with the third letters' corresponding water in the plural (Particles), it makes it much easier to read to communicate the plural including the lower-case s -- as MSPs. This is similar to way polar stratospheric clouds are referred to as PSCs. So replace the instances of "MSP" on lines 68 and 69 instead with "MSPs".
Also on lines 95 and 97.

Done (throughout the manuscript)

13) Introduction, line 82 -- Similarly you likely have "MF" here as an abbreviation for the plural term "Meteoric Fragments" but again, it's better to say "MFs", in the same way as MSPs and PSCs. Please change "MF" to "MFs" here and on line 84. Also on line 97.

Done

14) Introduction, line 83 -- I'm not sure why you are questioning whether meteoric fragments form here. The preceding sentence begins "As has recently been shown...", so either that sentence needs to be changed to "have suggested" or else this sentence needs to be re-worded. However, the existence of meteoric fragments has been clear since rocket-borne measurements in the early 1960s (e.g. Hemenway and Soberman, 1962), with the fragments terminology having been introduced in the 1950s (e.g. Jacchia, 1955). Suggest to re-word the start of the preceding sentence to "As was hypothesised in the 1950s (e.g. Jacchia, 1955) and shown in measurements from the 1960s (e.g. Hemenway and Soberman, 1962), recently also further established by Subasinghe et al. (2016)...". Maybe it's just to change "were formed" to "are formed" and add "at sufficient particle concentrations" afterwards.

Changed as suggested

15) Introduction, lines 90-94 -- Again, although the term IDP is being used here as the plural term, it's easier to read this making the plural clear as "IDPs". Please change "IDP" to "IDPs" in all instances here, except on line 95 when the term is used in the singular.

Changed as suggested

16) Introduction, line 101 -- Change "Later, aircraft based" to "More recently," or "Much more recently,"

Changed to "more recently"

17) Introduction, line 116 -- Delete "summer" from the "Tropics/sub-tropics" because this seasonal variation is not relevant here.

Done

18) Measurements and Methods, line 124 -- replace "includes" with "analyses", insert "lower" before "stratospheric" and provide a more descriptive word than "data", also avoiding using bland terms such as "obtained" (since they don't communicate these being measurements from the field). Suggest also to replace "data obtained during" with "aerosol composition measurements taken" and replace "research" with "field". Also insert "additional composition measurement" after "with two" and insert "the lower troposphere" before "altitudes below 3600m a.s.l.", putting that last text in brackets -- i.e. "the lower troposphere (altitudes below 3600m a.s.l.".

The sentence reads now: "This study analyses lower stratospheric and upper tropospheric aerosol composition measurements taken during five aircraft-based field campaigns, together with two additional composition measurements from the middle and lower troposphere (altitudes below 3600 m a.s.l)."

19) Section 2.1.1, line 133 -- suggest to insert ", the full dataset from" before "which are included".

Done

20) Section 2.1.3, line 147 -- hypenate "aircraft chasing" to "aircraft-chasing".

Done

21) Section 2.1.4, line 155 -- replace "data which were obtained during" with "the measurement data from the" and replace "flights reaching" with "flights that reached".

Done

22) Section 2.1.5, line 159 -- insert "middle and" before "lower troposphere", since Jungfraujoch is (in my opinion) sampling above the lower troposphere. Also change "we used two data sets" to "we also analyse two additional aerosol composition measurement datasets" and replace "low" with "lower".

Done

23) Section 2.1.5, line 160 -- replace "during NETCARE" with "during the NETCARE field campaign".

Done

24) Section 2.1.5, line 164 -- Improve the start of this sentence, changing "During the..." instead to "The other lower altitude dataset is from the mountain-top Jungfraujoch site during the...".

Changed to: "The other lower altitude dataset is from the mountain-top Jungfraujoch site (3600 m a.s.l.) where a single particle mass spectrometer was operated during the INUIT-JFJ (Ice Nucleation Research Unit Jungfraujoch) campaign in January and February 2017."

25) Section 2.2.1, line 175 -- Replace "has been described" with "is described".

Done

26) Section 2.2.1, line 180 -- Replace "Having passed the aerodynamic lens" with "Having passed through an aerodynamic lens", insert "then" before "accelerated" and change "the vacuum chamber" to "a vacuum chamber".

Done

27) Section 2.2.1, line 186 -- Suggest to delete "to the ALABAMA".

Done

28) Section 2.2.1, line 187 -- Suggest to insert "to this paper" after "supplement".

Done

29) Section 2.2.1, line 188 -- Suggest to replace "we include here a subset of" with "we analyse only the measurements from the"

Done

30) Section 2.2.1, line 189 -- Insert "(i.e. where ......)" after "reached the stratosphere" to clarify the criterion that was used for this.

This was done by inspecting the temperature profile of each flight. Only those three flights where a temperature minimum indicated that the tropopause was crossed were selected. We added ", as was inferred from the temperature profiles."

31) Section 2.2.2, line 200 -- replace "briefly reviewed" with "also described".

Done

32) Section 2.2.2, line 200 -- the acronym "CPI" should be spelt out here as "constant pressure inlet" since it is its first use. Note that cloud particle imager also has the same three-letter-acronym.

CPI is first spelled out in Section 2.2.1. We know that the cloud particle imager uses the same acronym, but the constant pressure inlet is published in the meantime by Molleker et al. (2020) in Atmospheric Measurement Techniques (we updated the reference).

```
33) Section 2.2.2, line 209 -- move "during the StratoClim measurements"
to the end of the sentence, as this is more of a clarifying term, i.e.
make the sentence instead say "... on particles was about 40\% at
diameters around 500nm during the StratoClim measurements".
```

We moved "during the StratoClim measurements" to the end of the sentence, i.e., after "… and below 5 % above 2000 nm".

```
34) Caption to Table 1 (line 216) -- change "Overview on the UTLS data
sets" to
"Overview of the 5 different aerosol composition measurement datasets".
```

Changed to "Overview of the 5 different aerosol composition measurement datasets from the UTLS used in this study", because we want to note that these are only the UTLS datasets, not the lower altitude datasets.

```
35) Table 1 -- Given the issue with these measurements all having a lower
frequency (higher wavelength) laser, add a row giving the wavelength used
here. Even though these (I think) are all the same at 266nm, it's important
to state these here so the reader can easily scan that Table to find that
information.
```

Done

```
36) Section 2.2.3, line 221 -- insert "5 UTLS" before "campaigns were
analyzed".
Also, since this is a European journal, please change all instances of
"analyzed" instead to "analysed".
```

Changed as suggested

```
38) Section 2.2.3, line 233-234 -- It is really great that the analysis
has done this analysis to understand the variations with these different
metrics, and the rationale for doing so should be stated. So please change
the start of this sentence from "Histograms of..." to "To enable to
understand the different origin of the meteoric-signature particles,
meteorological re-analysis data was combined with the measurements, with
histograms of...", deleting "were" before "calculated" and replace
"cluster as function of" with "cluster, as a function of".
```

Changed as suggested

```
39) Section 2.2.4, line 238 -- Suggest to improve the start of this 1st
para of this section, replacing "The relation of " with "The steep
vertical gradients in", and add "across the tropopause, means that
correlating with measurements or re-analysis of these species" before
"can be used". Then also replacing
"potential tropospheric influence" with "previous tropospheric
influence".
```

Changed to "The steep vertical gradients in water vapor (H2O) and ozone (O3) across the tropopause means that correlations of measurements or re-analysis of these species can be used to investigate the previous tropospheric influence of a stratospheric air mass."

```
40) Section 2.2.4, lines 239-240 -- the use of the word "tracer" is
potentially confusing (e.g. modellers use the word tracer as abbreviation
for "trace species"). I suggest with the re-wording in point 39), this
sentence can actually be deleted.
```

The sentence was deleted

```
41) Section 2.2.4, line 241 -- replace "These measurements are briefly..."
with "These additional measurement datasets are briefly..."
```

Done

```
42) Section 2.2.4, lines 245, 246 and 247 -- hyphenate these 3 instances
of "forward facing" instead to "forward-facing".
```

Done

```
43) Section 2.2.4, line 252 -- insert ", whose detection method is"
before "based on"
```

Done

```
44) Section 2.2.4, line 253 -- replace "of SHARC" with "of the SHARC
hygrometer".
```

Done

```
45) Section 2.2.4, line 254 -- replace "Monitor" with "monitor"
```

Done

```
46) Section 2.2.4, line 256 -- replace "whatever" with "whichever".
```

Done

```
47) Section 2.2.4, line 256 -- replace "with an" with "which has an".
```

Done, assuming that you meant line 259

```
48) Section 2.2.4, line 265 -- delete "range up to the extreme conditions"
and change "at a height of 20km" to "up to a height of ~20km".
```

Done

```
49) Section 2.3, line 272 -- insert "for stratifying the data (e.g. the
histograms in section 2.2.3)" before "were derived" and replace "using"
with "from the".
```

Done

50) Section 2.3, line 275 -- replace "first lapse rate tropopause" with "lowest altitude negative lapse rate" or some other more precise term.

Done

51) Section 3.1 -- line 279 -- Suggest to replace "Distinct particle type" with "Meteoric-signature particle type" to be more scientifically descriptive.

At this point in our argumentation, we did not draw the conclusion yet that these particles are of meteoric origin. This conclusion is drawn later in section 3.3. Thus we would prefer to leave the section heading as it is.

52) Figure 2 -- as per the main issue I am asking the authors to reply to, there is a question as to the composition of the particles whose spectra do not show any Fe and Mg peaks. The article needs to show the equivalent mean spectra for the non-meteoric-signature particles (ideally separated also into those with carbonaceous and those without carbonaceous, as in Murphy et al., 2007). This should be shown either in additional panels of this Figure 2 or as an additional Figure in the Supplementary Material.

We added to Figure 2 the mean spectra of all stratospheric particles without the meteoric signature. A further separation into different particles types (such as with and without carbonaceous signature) will be subject of upcoming publications that focus on the composition of UTLS aerosol under the influence of Asian and African monsoon.

53) Section 3.1 -- line 291 -- Replace "Further cations include" with "Additional minor cation peaks include"

Done

54) Section 3.1 -- line 292 -- suggest to replace "minor signals" with "trace signals"

Done

55) Section 3.1, lines 294-295 -- insert "the" before "two aircraft missions", insert "with the ERICA" after "missions" and delete "namely".

During CAFE-Africa we used the ALABAMA. Thus, we changed the sentence to "…from the aircraft mission StratoClim 2017 with the ERICA and from the aircraft mission CAFE-Africa 2018 with the ALABAMA".

56) Section 3.1, line 295 -- the word "spectra" is plural but here the term is referring to the mean of the spectra, which is singular, so the word "spectrum" should be used instead of "spectra" in this instance. Also delete "obtained" and insert "the" before "18668 measurements".

Done

57) Section 3.1, line 296 -- replace "during the StratoClim campaign" with simply "during StratoClim", and since the word "spectrum" is used, then the word "look" should be replaced with "looks". The word "compared"

can also be deleted on this line and "the mean mass spectra obtained" replaced with the word "that", also inserting "the" before "3310" and replacing "made during the CAFE-Africa 2018 campaign" with simply "during CAFE-Africa 2018". Those changes make the text much easier to read.

Done. As we also changed Fig. 2 to show only stratospheric particles, the numbers have slightly changed. The sentence now reads: "The mean mass spectrum from the 18421 measurements made during StratoClim 2017 looks remarkably similar to that from the 2882 measurements made during CAFE-Africa 2018."

58) Section 3.1, line 302 -- use the abbreviations Fe and Mg for iron and magnesium on this line and replace "binned" with the more scientific term "stratified".

Done

59) Section 3.1, lines 303-304 -- this sentence beginning "For each bin" can be deleted -- the information there is obvious and just makes this paragraph more difficult to read.

Done

60) In addition to deleting that sentence on lines 303-304, the text after that can be tacked onto the end of the first sentence in that paragraph as ", with bin sizes of ...."

Done

61) Section 3.1, lines 307-309 -- this sentence beginning "It has to be emphasized" can (in my opinion) be deleted -- that is obvious, and the text already gives the total number of particles in the previous sentence, so the reader will have those numbers in their mind already. I think it makes this sentence much easier to read if you simply delete this sentence (the reader will understand that to be the case already).

This sentence is needed to introduce the following statements on the problem with the detection of the pure sulfuric acid particles. Thus, we prefer to keep it.

62) Section 3.1, lines 311-312 -- again, use Fe and Mg abbreviations here rather than the words iron and magnesium. But more importantly this sentence needs to be much clearer how much of an effect this value is. Since Murphy et al. (2007) PALMS measurements, which have the lower wavelength (higher frequency, i.e. higher power) laser, and therefore do sample the pure sulfuric particles, show that these pure sulphuric particles represent only about 10% of the particles. So I think you can say here that the under-sampling of the pure sulphuric particles will not have a significant effect on the fractions given -- and that the reader should be confident in these numbers.

We changed to: "Thus, the fraction of the Fe and Mg particle type given here represents an upper limit and may be overestimated by about 10 – 30 %, because pure sulfuric acid particles are not taken into account. This is discussed in more detail in Section 3.5."

63) Section 3.1, line 324 -- replace "these particles" with "the meteoric-signature particles".

As stated before, the conclusion "meteoric-signature" has not been drawn yet, so we replace by "the Fe and Mg containing particles"

```
64) Section 3.1, line 330 -- you've written "tropopause" but you mean
"troposphere" here -- please correct that. Also insert "often" before
"defined via the..." and suggest to add "(known as the thermal tropopause
or cold-point tropopause)" after "lapse rate" and make that be the end of
that sentence. Then have that start the next sentence ". The potential
vorticity" instead of continuing as , but potential vorticity..."
```

Changed as suggested

```
65) Section 3.1, line 331 -- I'd suggest "better indicator" rather
than "good indicator" -- I think the dynamical tropopause would be
the preferred metric if both were available. And please also put
the words "dynamical tropopause" in inverted-commas in the manuscript,
also changing the preceding words from "indicator for the" instead
to ", representing a " so that the sentence is introducing this term.
```

We hope that this is what the reviewer means:

"The potential vorticity has been found to be a better indicator, representing a "dynamical tropopause" in the extratropics"

```
66) Section 3.1, lines 337-338 -- reword "during StratoClim 2017
which took place over the AMA" instead to "during the StratoClim
2017 flights sampling above the AMA".
```

Done

```
67) Section 3.2, line 351 -- replace "inserted" with "added to the
Figure".
```

Done

```
68) Section 3.2, lines 352-353 -- please state what time-interval for
the individual measurements (across which this median and quartiles
are taken).
```

It is not clear to us what is meant by this comment. The thermal tropopause values were taken from the ECMWF-interpolations along each flight path. Thus, time and lat-lon coordinates correspond to the flights. Otherwise, with respect to the in-situ measurements: All data were taken in their original time resolution (typically 1 second, sometimes 5 seconds) and the median and quartiles were calculated from all values in a certain latitude and temperature bin.

```
69) Section 3.2, line 353 -- insert "range for the dynamical tropopause
is shown, from" before "a 2 PVU and a 5 PVU surface" -- deleting
the "a" and replaceing "and" with "to" -- i.e. changing that to be
"range for the dynamical tropopause from 2 PVU to 5 PVU".
```

Changed as suggested

```
70) Section 3.2, Figure 4 -- in the legends delete the text
"with quartiles" -- that can go in the caption to the Figure.
Having it in the legend obscures some of the yellow parts of
the data, and it would be better then to have the smaller box
```

and seeing more of the data.

Changed as suggested

71) Section 3.5, line 482 -- Insert the word "accurately" after
estimate, and replace "an absolute" with "the absolute".

Done

72) Section 3.5, lines 486-487 -- as per my major comments at the start
of this review, this sentence needs to be changed -- it's not appropriate
to write "if pure sulfuric acid aerosol particles existed in the air".
It's clear from the PALMS measurements that only about 10-20 \% (at most
40%) of particles in the campaigns analysed in Murphy et al. (2007) were
of this pure sulphuric particle nature. And you should add a sentence
here stating these percentages so that the reader can know that at least
two-thirds of the particles (probably more) are being sampled by the 266nm
laser used by the ALABAMA and ERICA instruments. That way the reader can
know that it is only a relatively small-to-moderate fraction of the
particles that might be being missed in these measurements.

We have changed the beginning of Section 3.5 as already detailed above.

73) Section 3.5, line 490 -- insert "mid and" before "lower".

Done

74) Section 3.6, line 590 -- give the range of percentage occurrence that
you mean by "was very low". At Jungfraujoch this is 0.2 to 1\%, whereas
in the Canadian Arctic the value is much lower. Better to give the
corresponding values here.

We added "(0.0025 – 1%)" after "very low"

75) Section 4, lines 621-622 -- Suggest to replace "From previous" with
"Consistent with " and insert "aerosol composition measurement" after
"previous stratospheric" and replace "it was concluded" with "it is
concluded".

Done

76) Section 4, line 626 -- with this being the Discussion and conclusions
section, better here not to use the MSP acronym, instead give the words,
replacing "MSP particles" with "meteoric smoke particles".

Done

77) Section 4, line 637 -- replace "so this altitude" with "with this
altitude", then also replacing "refers to" instead to "referring only
to".

Done

78) Supplementary Material -- Introduction, 2nd line (1st sentence) Insert "shown in the main article, to enable its" before "interpretation", and insert "to be scrutinised transparently" after "interpretation.

Done

79) Supplementary Material -- Introduction, 2nd line (2nd sentence) Replace "It includes the clustering parameters...." with "Firstly, the clustering methodology is explained in more detail, with the clustering parameters...", also replacing "evaluation and the uncertainty" with "evaluation, and an associated uncertainty", replacing "estimation" with "estimated".

Done

80) Supplementary Material -- Introduction, 3rd line (3rd sentence) The text "Individual clusters of particles are displayed (S2)" needs to be changed because the Figures S1 to S5 show mean spectra not individual spectra. Also, the vertical profiles of the meteoric fraction are also shown in those Figures. So, replace that text instead with "Secondly (S2), the mean mass spectra and vertical profile of the meteoric-particle abundance fractions for each of the 5 UTLS campaigns are shown in Figures S1 to S5."

Done

81) Supplementary Material -- Introduction, 4th line (4th sentence) Insert "each of" before "the individual" and replace "mission" with "missions", adding "(Figure S6) after "in S3".

Done

82) Supplementary Material -- Introduction, 4th line (5th sentence) Insert "(Figure S7)" before "shows the O3-H2O"...

Done

83) Supplementary Material -- Introduction, 10th line (penultimate sentence in this section) Replace "present in" with "presented in".

Done

84) Supplementary Material -- Introduction, 11th line (final sentence in this section) Replace "SectionS10" with "Section S10", delete "the" after "explains", insert "changing" before "the threshold" and replace "was derived" with "affects the stratospheric proportions presented".

Done

85) Supplementary -- Clustering algorithm, lines 6 & 7 Replace "chose as distance metric" with "used for the distance metric" and replace "spectra): a Pearson..." with "spectra), with a perfect Pearson...", then putting "r=1" in brackets, and changing "means that" to "meaning that".

Changed to: "Linear correlation was used for the distance metric (defining "similarity" of the spectra), with a perfect Pearson correlation (r = 1) meaning that two spectra are identical."

86) Supplementary -- Clustering algorithm, final sentence Replace "stopping" with "convergence".

Done

87) Supplementary -- Variation of clustering parameters, line 6 Replace "particles containing" with "particles identified to contain" and correct "meteorological material" with "meteoric material".

Done

88) Supplementary -- Section S2 -- insert "each of" before "the five".

Done

89) Supplementary -- Section S3 -- insert "each of" before "the five" and replace "mission. All data were merged to" instead to "missions, these data merged to".

Done

90) Supplementary -- Section S8 -- line 8 of the text. Insert "when the refractive index for stratospheric aerosol is used".

Done

91) Supplementary -- Section S8 -- line 9 of the text. Delete "size channel with" and replace "corresponds to" with "increases to", deleting "for stratospheric aerosol particles".

Done

92) Supplementary -- Section S9 -- line 3 of the text. Insert "(NCEP meteorological re-analysis, Saha et al., 2010)" after "0.5 degree data set".

Done

93) Supplementary -- Section S9 -- line 5 of the text. Insert ", with" before "27 trajectories".

Done

94) Supplementary -- Section S9 -- line 6 of the text. Replace "binned in altitude and latitude bins and" with "stratified into altitude and latitude bins and"

Done

95) Supplementary -- Section S10 -- line 1 of the text. Replace "recorded" with "measured".

Done

```
96) Supplementary -- Section S10 -- line 2 of the text. Replace "We used"
with "To test the sensitivity of the calculations, we used"
```

Done

```
97) Supplementary -- Section S10 -- line 5 of the text. Replace "as a
threshold" with "as the threshold".
```

Done

We discuss the IDP and meteoritic material confusion done in many papers; we present a summary of the properties of the IDP and of cometary material. The authors must be advised that we had submitted a few month ago in ACP a previous version of our paper, but it was rejected by an associated editor that is in the same laboratory as one author of this Schneider et al. paper. Obviously, we are sure this is just a coincidence. Nevertheless, we encourage the authors to consider our work and to clarify their analysis considering the various sources that can exist for the material they have identified.

Thank you for pointing us to your recent paper in Atmosphere. We will refer to the results you obtained from stratospheric balloon-borne measurements in the revised version of our manuscript.

One point we would like to mention is that the particles we are describing (the Mg- and Fe-dominated particles we interpret as meteoric material dissolved in sulfuric acid) represent only a subset of the "none-pure sulfate particles" (NSP) that you describe in your publication. In fact, as also reviewer #2 pointed out (and we discussed in section 3.5), our single particle mass spectrometer has a low detection efficiency for pure sulfuric acid. That means, that

almost all particles detected by laser ablation mass spectrometry (using 266 nm ablation laser wavelength) represent NSP. Thus, many of the particles we observed will have terrestrial origin. But, the focus of our present paper is the stratospheric meridional distribution of particles containing meteoric material. Further publications will analyze the nature and sources of the other particles detected during the individual campaigns.

[revised manuscript text omitted]

**Introduction**

This document contains supplementary information, with the objective to give more background information on the data shown in the main article, to enable its interpretation to be scrutinised transparently. Firstly, the clustering methodology is explained in more detail, with the  clustering parameters of the single particle data evaluation, and an associated  uncertainty  estimated (S1). Secondly (S2), the mean mass spectra and vertical profiles of the meteoric-particle abundance fractions for each of the five UTLS campaigns are shown in Figures S1 to S5. The data displayed in Figure 4 of the main text are shown for each of the individual missions in S3 (Figure S6). Section S4 (Figure S7) shows the O$_3$-H$_2$O tracer-tracer plot for StratoClim 2016 which was not used in the main text.  The calculation of the sedimentation velocity is explained in S5. An example of sampling line loss calculation is presented in S6. Section  S7 shows the results of Mie calculations used to convert the calibration data to real stratospheric refractive indices. More information on the detection of meteoric material in tropospheric particles at the Jungfraujoch is presented in S8. Finally, Section S10  explains  how changing the threshold defining at what times stratospheric air was sampled affects the stratospheric proportions presented.

**S1 Clustering Parameters**

**Clustering algorithm and parameters**

The individual bipolar mass spectra were sorted using the fuzzy c-means algorithm (Bezdek et al., 1984; Hinz et al., 1999), using the software CRISP that was written at MPIC (Klimach et al., 2010; Klimach, 2012; Roth et al., 2016). The parameters are given in Table S1. Cations were used because the meteoric material is best recognized in the cation spectrum (Fe$^+$, Mg$^+$). Preprocessing is done by taking each ion signal to the power of 0.5 to reduce the influence of the signal intensity. The mass spectra were normalized to their sum to reduce the influence of the total ion count per spectrum. Linear correlation was used for the distance metric (defining "similarity" of the spectra), with  a perfect Pearson correlation  (r = 1) meaning that two spectra are identical. The number of clusters was prescribed with 20. A set of starting cluster centers was chosen from the data set with the condition that these clusters have Pearson correlation coefficient smaller than 0.9. The fuzzifier (originally introduced as "weighting exponent" by Bezdek (1981) represents the fuzziness (blurring, defocusing) of the classification. "Fuzzy abort" parameter defines the convergence criterion of the algorithm, i.e. when the differences between subsequent cluster runs change by less than the chosen value, the algorithm ends.

**Variation of clustering parameters**

To estimate the influence of the clustering parameters and the chosen sorting algorithm on the number of particles containing meteoric material, six additional different clustering runs were conducted for each UTLS

mission. The varied parameters are: number of clusters (10, 20, 30), initialization cluster difference (0.9, 0.7), fuzzyfier (1.3, 1.5), preprocessing (power = 0.5, none), and algorithm type (fuzzy c-means, k-means). Table S2 lists the different runs, the varied parameters and the resulting number of particles identified to contain ing meteoricological material.

Criteria for selecting a certain cluster as "containing meteoric material" were 1) high cation signals of $Fe^+$ and $Mg^+$ (additionally allowing $Na^+$, $K^+$, $Al^+$), 2) anion signal at $HSO_4^-$ or cation signals at $S^+$, $SO_4^+$, $H_3SO_4^+$, 3) vertical profile showing increasing fractional abundance with increasing altitude, potential temperature, or potential vorticity.

In general, the standard deviation is below 6% of the mean value, and the chosen final clustering result using the parameters given in Table S1 (printed in bold) is very close to the mean value.

**S2 Individual cluster properties.**

Figures S1 through S5 show the particle clusters identified as "containing meteoric material" for each of the five UTLS aircraft missions. The cluster were selected following the criteria given in section S1.

**S3 Theta-latitude histograms for individual aircraft missions.**

Figure S6 shows the theta-latitude histograms for each of the five UTLS aircraft missions, these. All data were merged to produce Figure 4 of the main text.

**S4 Tracer-tracer correlation for StratoClim 2016.**

The $O_3$ and $H_2O$ measurements during StratoClim 2016 did not cover the whole flight time of the three measurement flights. Thus, the data were not used in Figure 8 of the main text, but for completeness are shown here in Figure S7.

**S5 ALABAMA hit rate in stratosphere and troposphere.**

In section 3.5 of the main text we calculate the absolute number concentrations of particle containing meteoric material. For this, we use the hit rate of the mass spectrometer to estimate the number of "missed" particles. Figure S8 shows the relationship between hit rate and $O_3$ which allows for estimating the contribution of "invisible" pure $H_2SO_4$ particles in the stratosphere.

**S6 S5 Particle sedimentation in the lower stratosphere.**

The time scale for particle sedimentation was calculated as follows:

Pressure, temperature and viscosity of air were taken from the US Standard Atmosphere, using 100 m vertical resolution.

Mean free path ($\lambda$), Knudsen number ($Kn$), Cunningham correction ($C_C$) and terminal settling velocity ($V_{TS}$) were calculated using the following equations ((Hinds, 1999; Seinfeld and Pandis, 2006)):

$$\lambda = \frac{1}{\sqrt{2}\pi d_m^2 N}, \tag{S1}$$

with $d_m$ = collision diameter of air molecules ($3.7\times10^{-10}$ m), and $N$ = number density of air molecules,

$$Kn = \frac{2\lambda}{d} \tag{S2}$$

with $d$ = particle diameter,

$$C_C = 1 + Kn\left(\alpha + \beta e^{-\frac{\gamma}{Kn}}\right), \tag{S3}$$

with $\alpha = 1.155$, $\beta = 0.471$, $\gamma = 0.596$ (Allan and Raabe, 1982)

$$V_{TS} = \frac{\varrho \, d^2 \, g \, C_C}{18\,\eta}, \tag{S4}$$

with $\varrho$ = particle density, $g$ = acceleration of gravity, $\eta$ = viscosity of air.

The terminal settling velocity was calculated for pure $H_2SO_4$ particles ($\varrho = 1.83$ g cm$^{-3}$) and pure olivine particles ($\varrho = 3.30$ g cm$^{-3}$), assuming spherical particle shape. Figure  S8 shows the terminal settling velocity as a function of altitude.

**S6 Sampling line transmission efficiency.**

The sampling line transmission efficiency was calculated here as an example for the configuration of ML-CIRRUS (ALABAMA operated on the HALO aircraft). The ¼" stainless steel sampling line that connected the HALO aerosol submicrometer inlet (HASI, Wendisch et al. (2016); Andreae et al. (2018)) line had a total length of 2.9 m with several bends, horizontal and vertical sections. The calculations were done with a modified version of the Particle Loss Calculator (PLC) that was originally described in von der Weiden et al. (2009). The modified version allows for including the sampling line pressure. The results are shown in Figure S9.

**S7 Mie calculations for stratospheric aerosol for the optical particle spectrometers OPC 1.129 and UHSAS.**

The response of the optical particle spectrometers OPC 1.129 and UHSAS for stratospheric aerosol particles was calculated using an in-house written software (Vetter, 2004) following the algorithms described in Bohren and Huffmann (1983). The OPC 1.129 uses a laser wavelength of 655 nm. The scattered light is collected under 90° with an angular range of 60° (i.e. 60° – 120°). The UHSAS uses a laser wavelength of 1054 nm and collects the scattered light in an angular range between 22° and 158°. The refractive index for PSL, $m = 1.59$, was taken from Heim et al. (2008). The refractive index range ($m = 1.43 – 1.45$) for lower stratospheric aerosol was taken from Yue et al. (1994).

The results (Figure S10) show that the lower size cut-off of the OPC shifts from 250 nm to about 285 nm when the refractive index for stratospheric aerosol is used. For the UHSAS, the  lower cut-off of 180 nm (calibrated by PSL) increases  to 200 nm .

**S9 Jungfraujoch backtrajectories, ozone mixing ratio, and meteoric particle fraction.**

We inspected backtrajectories to obtain more information on the origin of the meteoric particles detected during the INUIT-2017 campaign at the Jungfraujoch station (3600 m altitude). For this, HYSPLIT (Stein et al., 2015) back trajectories were calculated for 120 – 143 hours using the GDAS 0.5 degree data set (NCEP meteorological re-analysis, Saha et al., 2010). We chose 3600 m a.s.l. as a starting point, with 27 trajectories were started per hour, using the trajectory ensemble option. All trajectory data points were stratified   altitude and latitude bins and the number of trajectory points per grid is plotted in Figure  S11 a) and b). Panel c) of Figure S11 shows the time series of the fraction of meteoric particles detected along with the $O_3$ mixing ratio. The fraction of meteoric particles is found to be highest between the 18th and 21st of February, a period during which the air mass spent more time at higher altitudes and latitudes before arriving at the Jungfraujoch. A similar but less pronounced feature is found between 15th and 17th of February. These findings support the conclusion that the origin of this particle type is at higher altitudes. The dependence on latitude can be explained by the fact that mixing between stratosphere and troposphere is stronger at mid-latitudes (see Figure 8 in the main part) than in the tropics. Thus, we would expect to see a higher meteoric particle fraction when the air masses have experienced higher latitudes and altitudes, which is confirmed by Figure S11. Furthermore, a stratospheric origin is supported by the ozone trend in Panel c). The time trend of the meteoric particles follows closely the ozone time series, and even small-scale features (e.g. Feb. 09, Feb. 11, Feb. 16, and Feb. 22) are clearly visible in both time series.

**S10 Number of particles in the stratosphere**

Table S3 shows the number of particle mass spectra measured in the stratosphere, for all particles and for meteoric particles. To test the sensitivity of the calculations, we used different definitions of the tropopause (PV > 2 PVU, PV > 4 PVU, $O_3$ > 150 ppbv, and pot. temp. > 380 K). The latter criterion could only be applied to the StratoClim data sets, because only the Geophysica reaches to 380 K. The ozone criterion could not be applied to the StratoClim 2016 data set due to low data coverage. As a final result, we decided to use PV > 4 PVU as  the threshold for Table 1 in the main text.

[Figure]

**Figure S1. Vertical profiles and mean mass spectra of all clusters from the ML-CIRRUS 2014 data set interpreted as particles containing meteoric material. Note that the negative mass spectra (anions) are noisy, because all anion spectra of each cluster type were averaged for this display, also those where no anion signal was obtained.**

[Figure]

**Figure S2. Vertical profiles and mean mass spectra of all clusters from the StratoClim 2016 data set interpreted as particles containing meteoric material.**

[Figure]

**Figure S3. Vertical profiles and mean mass spectra of all clusters from the StratoClim 2017 data set interpreted as particles containing meteoric material.**

[Figure]

**Figure S4. Vertical profiles and mean mass spectra of all clusters from the ND-MAX/ECLIF-2 data set interpreted as particles containing meteoric material.**

[Figure]

**Figure S5. Vertical profiles and mean mass spectra of all clusters from the CAFE-Africa data set interpreted as particles containing meteoric material.**

[Figure]

**Figure S6. Fraction of meteoric particles as function of potential temperature and latitude for the individual missions: a) ML-CIRRUS, b) StratoClim 2016, c) StratoClim 2017, d) ND-MAX-ECLIF-2, e) CAFE-Africa.**

[Figure]

**Figure S7. Tracer-tracer correlation for StratoClim 2016. As the instruments were not fully operative during the whole flight time of the three flights of StratoClim 2016, several gaps in the data prevent the analysis of cross-tropopause transport for this mission.**

[Figure]

**Figure S8. Hit rate (mass spectra per laser shot during time intervals of one minute) of ALABAMA during a) a flight on April 01, 2014 (ML-CIRRUS) and b) a flight on Sept 07, 2018 (CAFE-Africa) as a function of time along with ozone mixing ratio and flight altitude. c) + d): Hit rate versus ozone mixing ratio. Dividing between troposphere and stratosphere at 150 ppbv ozone (ML-CIRRUS) and 100 ppb ozone (CAFE-Africa), respectively, yields an average hit rate in the stratosphere of 0.044 ± 0.012 (ML-CIRRUS) and 0.21 ± 0.04 (CAFE-Africa). For the troposphere, we obtain 0.091 ± 0.037 (ML-CIRRUS) and 0.54 ± 0.18 (CAFE-Africa).**

[Figure]

**Figure S89. Terminal settling velocity for H₂SO₄ and olivine particles of 100, 200, and 500 nm volume equivalent diameter ($d_{ve}$).**

[Figure]

Figure S9. Transmission curves for the sampling line used in ML-CIRRUS that connected the HALO aerosol submicrometer inlet (HASI) to the OPC 1.129 in the ALABAMA rack.

[Figure]

**Figure S10.** Relative intensity at the detector calculated for the OPC (Grimm 1.129, "Sky-OPC") and for the UHSAS, for PSL particles and for stratospheric particles.

[Figure]

**Figure S11. Backtrajectory information for the INUIT-JFJ 2017 field campaign at Jungfraujoch. For each hour, 27 backtrajectories were started using the trajectory ensemble option and were followed for 120 – 143 h using HYSPLIT [Stein et al., 2015] with the GADS 0.5 degree data set. Panel a) shows the number of points during which the trajectories resided in the respective altitude and time grid, Panel b) the same but for latitude. Panel c) shows the number fraction of the meteoric particles along with ozone mixing ratio.**

| Ion type | Cations |
|---|---|
| Preprocessing | Power each m/z by 0.5 |
| Normalization | Sum |
| Distance metric | Correlation |
| Initialization | Different startclusters: |
| Number of clusters | 20 |
| Cluster difference | 0.9 |
| Fuzzifier | 1.3 |
| Fuzzy abort | 1e-5 |

Table S1. Clustering parameters used for the final analysis.

| Project | Cluster algorithm | Number of clusters | Cluster diff | Fuzzifier | Pre-processing | Number of "meteoric" particles |
|---|---|---|---|---|---|---|
| ML-CIRRUS | Fuzzy c-means | 10 | 0.9 | 1.3 | $(m/z)^{0.5}$ | 3080 |
| | **Fuzzy c-means** | **20** | **0.9** | **1.3** | **$(m/z)^{0.5}$** | **3140** |
| | Fuzzy c-means | 30 | 0.9 | 1.3 | $(m/z)^{0.5}$ | 3136 |
| | Fuzzy c-means | 20 | 0.7 | 1.3 | $(m/z)^{0.5}$ | 2931 |
| | Fuzzy c-mean | 20 | 0.9 | 1.5 | $(m/z)^{0.5}$ | 3136 |
| | Fuzzy c-means | 20 | 0.9 | 1.3 | none | 3051 |
| | k-means | 20 | 0.9 | N/A | $(m/z)^{0.5}$ | 3247 |
| Mean ± StdDev | | | | | | 3103 ± 90 |
| | | | | | | |
| StratoClim 2016 | Fuzzy c-means | 10 | 0.9 | 1.3 | $(m/z)^{0.5}$ | 2357 |
| | **Fuzzy c-means** | **20** | **0.9** | **1.3** | **$(m/z)^{0.5}$** | **2412** |
| | Fuzzy c-means | 30 | 0.9 | 1.3 | $(m/z)^{0.5}$ | 2679 |
| | Fuzzy c-means | 20 | 0.7 | 1.3 | $(m/z)^{0.5}$ | 2376 |
| | Fuzzy c-mean | 20 | 0.9 | 1.5 | $(m/z)^{0.5}$ | 2567 |
| | Fuzzy c-means | 20 | 0.9 | 1.3 | none | 2618 |
| | k-means | 20 | 0.9 | N/A | $(m/z)^{0.5}$ | 2570 |
| Mean ± StdDev | | | | | | 2511 ± 118 |
| | | | | | | |
| StratoClim 2017 | Fuzzy c-means | 10 | 0.9 | 1.3 | $(m/z)^{0.5}$ | 18355 |
| | **Fuzzy c-means** | **20** | **0.9** | **1.3** | **$(m/z)^{0.5}$** | **18688** |
| | Fuzzy c-means | 30 | 0.9 | 1.3 | $(m/z)^{0.5}$ | 19700 |
| | Fuzzy c-means | 20 | 0.7 | 1.3 | $(m/z)^{0.5}$ | 18688 |
| | Fuzzy c-mean | 20 | 0.9 | 1.5 | $(m/z)^{0.5}$ | 18459 |
| | Fuzzy c-means | 20 | 0.9 | 1.3 | none | 21235 |
| | k-means | 20 | 0.9 | N/A | $(m/z)^{0.5}$ | 20215 |
| Mean ± StdDev | | | | | | 19334 ± 1006 |
| | | | | | | |
| ND-MAX/ ECLIF-2 | Fuzzy c-means | 10 | 0.9 | 1.3 | $(m/z)^{0.5}$ | 20141 |
| | **Fuzzy c-means** | **20** | **0.9** | **1.3** | **$(m/z)^{0.5}$** | **23138** |
| | Fuzzy c-means | 30 | 0.9 | 1.3 | $(m/z)^{0.5}$ | 21883 |
| | Fuzzy c-means | 20 | 0.7 | 1.3 | $(m/z)^{0.5}$ | 21681 |
| | Fuzzy c-mean | 20 | 0.9 | 1.5 | $(m/z)^{0.5}$ | 21126 |
| | Fuzzy c-means | 20 | 0.9 | 1.3 | none | 21752 |
| | k-means | 20 | 0.9 | N/A | $(m/z)^{0.5}$ | 18998 |
| Mean ± StdDev | | | | | | 21245 ± 1237 |
| | | | | | | |
| CAFE-Africa | Fuzzy c-means | 10 | 0.9 | 1.3 | $(m/z)^{0.5}$ | 3325 |
| | **Fuzzy c-means** | **20** | **0.9** | **1.3** | **$(m/z)^{0.5}$** | **3310** |
| | Fuzzy c-means | 30 | 0.9 | 1.3 | $(m/z)^{0.5}$ | 3290 |
| | Fuzzy c-means | 20 | 0.7 | 1.3 | $(m/z)^{0.5}$ | 3194 |
| | Fuzzy c-mean | 20 | 0.9 | 1.5 | $(m/z)^{0.5}$ | 3281 |
| | Fuzzy c-means | 20 | 0.9 | 1.3 | none | 3287 |
| | k-means | 20 | 0.9 | N/A | $(m/z)^{0.5}$ | 3515 |
| Mean ± StdDev | | | | | | 3314 ± 90 |

Table S2. Variations of clustering parameters. The inferred number of particles containing meteoric material is given in the last column. Other parameters were kept as in Table S1.

a) All particles

|  | PV > 2 | PV > 4 | O3 > 150 | Theta > 380 |
|---|---|---|---|---|
| ML-CIRRUS | 13029 | 6509 | 6174 | N/A |
| SC16 | 6662 | 5092 | N/A | 4874 |
| SC17 | 76856 | 51599 | 41146 | 57109 |
| NDMAX | 78454 | 73367 | 72923 | N/A |
| CAFE | 12161 | 10771 | 9441 | N/A |

b) Meteoric particles

|  | PV > 2 | PV > 4 | O3 > 150 | Theta > 380 |
|---|---|---|---|---|
| ML-CIRRUS | 3063 | 2986 | 2477 | N/A |
| SC16 | 2363 | 2271 | N/A | 2238 |
| SC17 | 18487 | 18421 | 18016 | 18450 |
| NDMAX | 22626 | 22050 | 22104 | N/A |
| CAFE | 2946 | 2882 | 2789 | N/A |

c) Proportion

|  | PV > 2 | PV > 4 | O3 > 150 | Theta > 380 |
|---|---|---|---|---|
| ML-CIRRUS | 0.235 | 0.459 | 0.401 | N/A |
| SC16 | 0.355 | 0.446 | N/A | 0.459 |
| SC17 | 0.241 | 0.357 | 0.438 | 0.323 |
| NDMAX | 0.228 | 0.301 | 0.303 | N/A |
| CAFE | 0.242 | 0.268 | 0.295 | N/A |

Table S3. a) Number of analyzed particles in the stratosphere; b) number of meteoric particles in the stratosphere, c) proportion (numbers in b) divided by numbers in a)). Different criteria were used to define the tropopause. For Table 1 and Figure 6 in the main text, PV > 4 was selected for the following reasons: to avoid the mixing regime at mid-latitudes and to match the tropical 380 K tropopause definition (see also (Ploeger et al., 2015)).